# NeSyC: A Neuro-symbolic Continual Learner For Complex Embodied Tasks In Open Domains

**Wonje Choi**[1][*] **Jinwoo Park**[2][*] **Sanghyun Ahn**[1]**, Daehee Lee**[1][†] **Honguk Woo**[1,2][‡]
[1]Department of Computer Science and Engineering, Sungkyunkwan University
[2]Department of Artificial Intelligence, Sungkyunkwan University
{wjchoi1995, pjw971022, shyuni5, dulgi7245, hwoo}@skku.edu

## Abstract

We explore neuro-symbolic approaches to generalize actionable knowledge, enabling embodied agents to tackle complex tasks more effectively in open-domain environments. A key challenge for embodied agents is the generalization of knowledge across diverse environments and situations, as limited experiences often confine them to their prior knowledge. To address this issue, we introduce a novel framework, NeSyC, a neuro-symbolic continual learner that emulates the hypothetico-deductive model by continually formulating and validating knowledge from limited experiences through the combined use of Large Language Models (LLMs) and symbolic tools. Specifically, we devise a contrastive generality improvement scheme within NeSyC, which iteratively generates hypotheses using LLMs and conducts contrastive validation via symbolic tools. This scheme reinforces the justification for admissible actions while minimizing the inference of inadmissible ones. Additionally, we incorporate a memory-based monitoring scheme that efficiently detects action errors and triggers the knowledge refinement process across domains. Experiments conducted on diverse embodied task benchmarks—including ALFWorld, VirtualHome, Minecraft, RLBench, and a real-world robotic scenario—demonstrate that NeSyC is highly effective in solving complex embodied tasks across a range of open-domain environments.

## 1 Introduction

Recent advances in neuro-symbolic systems—integrating Large Language Models (LLMs) with symbolic tools (Frederiksen, 2008; Gebser et al., 2019)-have gained much attention for embodied task planning (Lin et al., 2024; Liu et al., 2023). These systems decouple contextual understanding—such as observation and instruction translation—from actionable knowledge including action preconditions and effects. Yet, these systems have not been thoroughly explored in open-domains, where the environment is not restricted to pre-defined tasks or knowledge and embodied agents must manage diverse scenarios. Conventional approaches rely on symbolic representations of expert-level actionable knowledge, which limits their applicability and effectiveness in real-world situations. The unpredictable and dynamic nature of open-domains often leads to incompleteness and inconsistency in knowledge, thus complicating the decision-making process of embodied agents.

In neuro-symbolic systems, generalizing prior actionable knowledge in open-domain environments presents practical challenges: (1) inherent lack of flexibility in symbolic systems to apply knowledge to unfamiliar environments, (2) limited methods to bridge the gap between the prior knowledge and new environments, leading to repeated action errors in complex situations, and (3) mislabeling of action affordances or insufficient feedback, caused by the inability to retain labeled experiences, which hinders the agent's ability to generalize knowledge and improve decision-making.

To address the challenges posed by adopting neuro-symbolic approaches in open-domains, we draw inspiration from the hypothetico-deductive model (Smokler, 1966). This model emphasizes falsification through experiences and emulates the scientific inquiry process by continually forming hypothe-

---

[*]Equal contribution     [†]Work done while a visiting scholar at CMU     [‡]Corresponding author

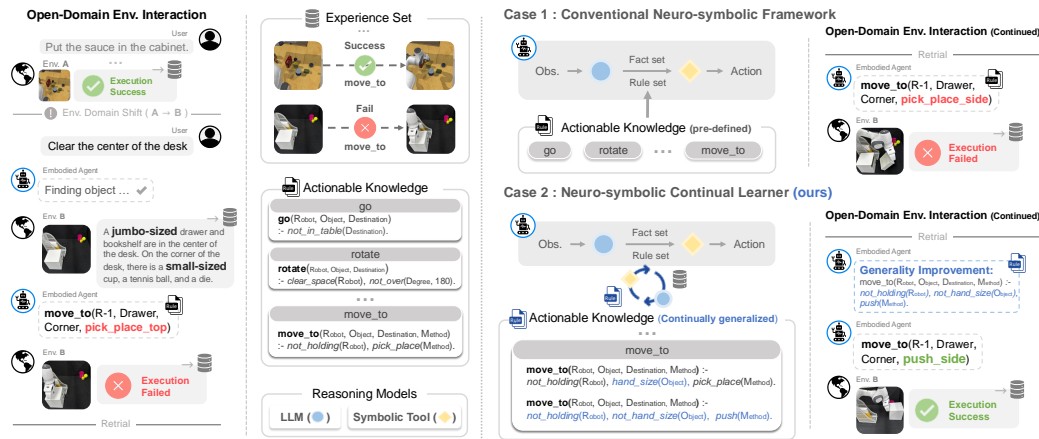

Figure 1: The concept of NESYC. In the leftmost part, domain shift leads the agent to fail by trying to grasp an oversized drawer's broad surface, which is infeasible. The remaining parts contrast two approaches: Case 1 treats LLMs and symbolic tools as separate functions for semantic parsing and logical reasoning, while Case 2 integrates them into a collaborative process, enabling NESYC to generalize actionable knowledge and compute logically valid actions for open-domain environment.

ses, rigorously testing them against available observations, and iteratively revising them. Guided by this model, we explore knowledge generalization strategies that interleave inductive and deductive reasoning, aiming to generalize knowledge applicable in open-domains and enable embodied agents to adapt more effectively to unpredictable situations. We introduce a novel framework, NESYC, a neuro-symbolic continual learner that combines the strengths of LLMs and symbolic tools to effectively formulate and apply knowledge in open-domains. In Figure 1, unlike conventional neuro-symbolic approaches relying on pre-defined actionable knowledge, NESYC uses accumulated experiences to generalize knowledge. This enables NESYC to execute actions effectively in the given environment while continually generalizing actionable knowledge to adapt across open-domains.

Specifically, we devise two key components in NESYC. First, we employ a **contrastive generality improvement** scheme that iteratively generates hypotheses using LLMs and conducts contrastive validation through symbolic tools. This scheme reinforces the validity of admissible actions while minimizing the inference of inadmissible ones, combining the generalization capabilities of LLMs with the logical rigor of symbolic tools to generalize actionable knowledge. Second, we implement a **memory-based monitoring** scheme that efficiently detects action errors and triggers the knowledge refinement process, continually expanding the agent's coverage of actionable knowledge.

To evaluate NESYC, we conduct experiments on ALFWorld (Shridhar et al., 2020c), Virtual-Home (Puig et al., 2018), Minecraft in Silver & Chitnis (2020), RLBench (James et al., 2020), and a real-world robotic scenario, demonstrating its applicability in open domains. Compared to the advanced baseline **AutoGen** (Wu et al., 2023a), NESYC achieved task success rate improvements of 33.6% on ALFWorld, 43.9% on VirtualHome, 53.7% on Minecraft, and 52.6% on RLBench.

The contributions of our work are as follows: (1) We present the neuro-symbolic continual learner NESYC based on the hypothetico-deductive model to enable generalization of actionable knowledge in open-domain environments. (2) We devise two schemes tailored for actionable knowledge generalization in NESYC: contrastive generality improvement and memory-based monitoring. (3) We validate NESYC through experiments on diverse benchmarks and real-world scenarios, demonstrating its effectiveness and significant performance improvements in open-domain environments.

## 2 BACKGROUND AND PROBLEM FORMULATION

### 2.1 INDUCTIVE LOGIC PROGRAMMING (ILP)

ILP (Muggleton & De Raedt, 1994) is a machine learning technique where the learned model is represented as a logic program, or hypothesis (i.e., a set of rules), derived from a combination of

examples and background knowledge. A common setting in ILP is Learning from Interpretations (LFI), where each example is an interpretation represented as a set of facts (Cropper & Dumančić, 2022). Given a program $BK$ denoting the background knowledge, along with sets of positive examples $E^+$ and negative examples $E^-$, the goal is to find an optimal hypothesis $H$ satisfying:

$$\begin{cases} \forall e \in E^+, e \text{ is an interpretation of } H \cup BK. \\ \forall e \in E^-, e \text{ is not an interpretation of } H \cup BK. \end{cases} \tag{1}$$

Here, $BK$ functions similarly to features in traditional machine learning, but it is more expressive, as it can include relations and information associated with examples. During LFI, $\theta$-subsumption (Sakama, 2001) is key in determining whether a hypothesis subsumes examples, checking if the hypothesis can be interpreted as the examples through variable substitution. Further details can be found in Cropper & Dumančić (2022), which offers an in-depth overview of ILP.

## 2.2 ANSWER SET PROGRAMMING (ASP)

ASP (Lifschitz, 2019) is a declarative programming paradigm well-suited for solving complex combinatorial problems like planning, particularly in non-monotonic domains where the dynamics and actions of embodied environments can alter future states. An ASP solver (Gebser et al., 2019) computes one or more *answer sets*, representing valid solutions by encoding problems as logic programs composed of rules in the following form:

$$A \text{ :- } B_1, \ldots, B_m, \text{ } not \text{ } B_{m+1}, \ldots, \text{ } not \text{ } B_n. \tag{2}$$

In this general form, a rule consists of a *head* ($A$) and a *body* ($B_1, \ldots, not \text{ } B_n$), where each $A$ and $B_i$ ($1 \leq i \leq n$) is an atom. The *head* represents the conclusion, and the *body* specifies the conditions, which include both positive conditions ($B_1, \ldots, B_m$), that must hold true, and negated conditions ($not \text{ } B_{m+1}, \ldots, not \text{ } B_n$), where *not* denotes negation as failure (NAF). NAF assumes the negated conditions to be false unless evidence to the contrary is provided. A rule with an empty *body* is a *fact* (e.g., $A$), while a rule with an empty *head* is a *constraint*, representing conditions that must not be satisfied. ASP is well-suited for evaluating the coverage of hypotheses in ILP by identifying which examples are satisfied (Law et al., 2020), and it also proves effective for planning in complex, dynamic environments (Cabalar et al., 2019). These capabilities are essential to our framework, enabling knowledge generalization through the interplay of induction and deduction.

## 2.3 PROBLEM FORMULATION

The open-domain embodied task planning problem is formulated as a tuple $(\mathcal{D}, \mathcal{S}, \mathcal{A}, \mathcal{F})$. Here, $\mathcal{D}$ represents the domain space for open-domain environments, while $\mathcal{S}$ denotes the state space. Due to partial observability (Sutton & Barto, 2018), the agent perceives observations $o_t \in \Omega$ at each timestep, which provide partial information about state $s \in \mathcal{S}$. $\mathcal{A}$ is the action space. The function $\mathcal{F}$ maps a domain $d \in \mathcal{D}$ to its specific goal states and dynamics $\mathcal{F}(d) = \{\mathcal{G}_d, T_d\}$ (Hallak et al., 2015). For a given domain $d$, $\mathcal{G}_d \subset \mathcal{S}$ represents the goal states derived from the instruction set $\mathcal{I}_d$, while $T_d : \mathcal{S} \times \mathcal{A} \rightarrow \mathcal{S}$ models how actions affect state transitions within the domain, defining the environment dynamics. In open-domain environments, the agent may not have full knowledge of $T_d$, making it essential to adapt to the environment. The objective of NESYC is formulated as:

$$\pi^* = \arg\max_{\pi} \mathbb{E}_{d \sim \mathcal{D}} \left[ \sum_t \text{SR}(s_t, \pi(\cdot \mid o_t, i_d)) \right] \tag{3}$$

where $i_d \sim \mathcal{I}_d$ corresponds to $g_d \in \mathcal{G}_d$, and $\text{SR} : \mathcal{S} \times \mathcal{A} \rightarrow \{0,1\}$ indicates whether the agent successfully completes the task given current states. Policy $\pi$ selects action $a_t$ based on observation $o_t$ and instruction $i_d$ (Yoo et al., 2024; Brohan et al., 2023; Huang et al., 2024).

## 3 NESYC: A NEURO-SYMBOLIC CONTINUAL LEARNER

## 3.1 OVERALL FRAMEWORK

We propose NESYC, a neuro-symbolic continual learner designed to generalize actionable knowledge for embodied agents in open-domain environments. To effectively utilize the limited experiences of agents, this framework integrates the capabilities of LLMs and symbolic tools. It maximizes

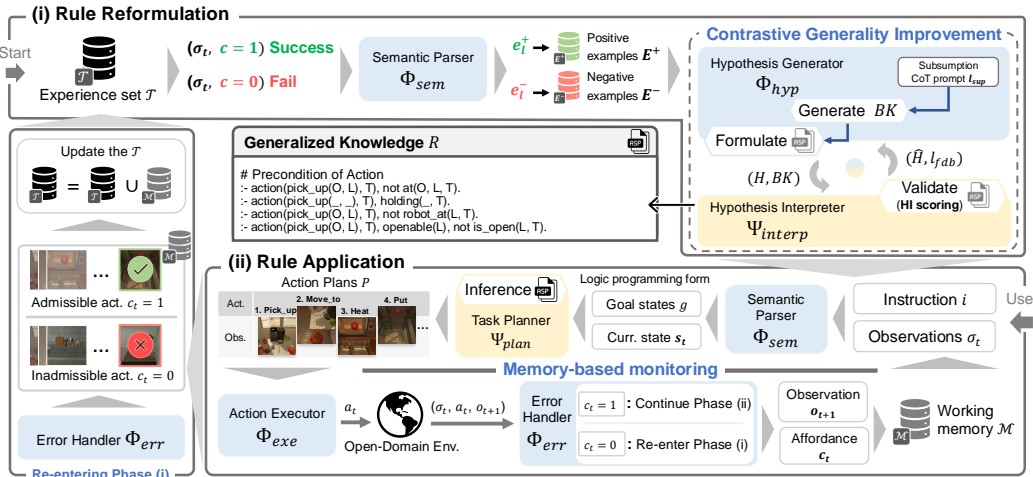

Figure 2: The structure of NESYC. NESYC iterates (i) Rule Reformulation and (ii) Rule Application phases. In (i), generalized knowledge $R$ is reformulated via contrastive generality improvement. In (ii), $R$ is applied and continually adapted to the environment via memory-based monitoring.

the consistencies of experiences with admissible actions via common-sense reasoning and minimizes contradictions from experiences with inadmissible actions via symbolic reasoning. To achieve this, our framework operates in two phases: (i) Rule Reformulation and (ii) Rule Application, as illustrated in Figure 2. Each phase is enabled by contrastive generality improvement and memory-based monitoring schemes, respectively, both of which are implemented via the interleaved collaboration of LLMs and symbolic tools. In the **Rule Reformulation** phase, NESYC employs a contrastive generality improvement scheme based on ILP to formulate generalized knowledge from accumulated experiences. An LLM leverages common-sense reasoning to generate hypotheses, while a symbolic tool ensures their logical consistency through systematic validation. The iterative and reflective nature of this scheme allows the resulting knowledge to achieve both accuracy and broad applicability across domains. In the **Rule Application** phase, NESYC employs a memory-based monitoring scheme based on ASP for embodied task planning, where experiences are collected and categorized based on admissible and inadmissible actions. The LLM offers a contextual understanding of observations, while the symbolic tool computes precise actions using the current actionable knowledge. If an action failure is detected during task execution based on observations, NESYC then re-enters the phase (i), triggering the knowledge refinement to better adapt to the environment.

## 3.2 RULE REFORMULATION

As shown in Figure 2, NESYC reformulates generalized knowledge $R$, which represents causal rules for action preconditions and effects derived from the experience set $\mathcal{T}$. To achieve this, NESYC employs the contrastive generality improvement scheme based on ILP. The experiences in $\mathcal{T}$ are translated into an example set $E$, comprising positive examples $E^+$ and negative examples $E^-$, based on action affordances. The generality improvement process is both iterative and reflective, driven by the hypothesis generator $\Phi_{\text{hyp}}$ and the hypothesis interpreter $\Psi_{\text{interp}}$, which collaboratively refine the hypotheses. Through this process, the interpretability of the hypotheses $\mathcal{H}$ is progressively enhanced, reinforcing the logical justification and reasoning with respect to $E^+$ and $E^-$, until $R$ is obtained. Algorithm 1 lists the rule reformulation phase of NESYC.

**Semantic parser.** To extract ground rules (i.e., rules without variables) from the experience set $\mathcal{T}$, the semantic parser $\Phi_{\text{sem}}$, utilizes in-context learning with an LLM, following neuro-symbolic approaches (Olausson et al., 2023; Pan et al., 2023). Focusing on action preconditions and effects, we prompt the LLM to translate trajectory $\sigma_t = (o_1, a_1, \ldots, o_t)$ into ground rules that represent a transition. The parser $\Phi_{\text{sem}}$ and the example set $E$ are formulated as:

$$E = \{(\Phi_{\text{sem}}(\sigma_t), c_{t-1}) \mid (\sigma_t, c_{t-1}) \in \mathcal{T}\} \quad \text{where} \quad \Phi_{\text{sem}} : \sigma_t \mapsto (s_{t-1}, a_{t-1}, s_t). \quad (4)$$

Examples are partitioned into positive set $E^+$ and negative set $E^-$ based on action affordance $c_{t-1}$.

**Hypothesis generator.** To induce hypotheses $\mathcal{H}$ that satisfy both positive and negative examples, we guide the LLM to extract background knowledge $BK$, which enhances context and facilitates the alignment of hypotheses with the examples. We then use $BK$ to generate $\mathcal{H}$ by employing a structured prompt that incorporates the $\theta$-subsumption technique from ILP, combined with a batch sampling strategy. The $\theta$-subsumption technique allows us to determine if one clause is more general than another by finding a substitution $\theta$ that makes one clause imply the other. To simplify this process, we leverage the LLMs' multi-step reasoning capabilities via a subsumption Chain-of-Thought (CoT) prompt, denoted as $l_{\text{sub}}$. The hypothesis generator $\Phi_{\text{hyp}}$ is then defined as:

$$\Phi_{\text{hyp}} : (\mathcal{B}, \ \mathcal{H}_{b-1}^i, \ l_{\text{sub}}, \ l_{\text{fdb}}^{i-1}) \mapsto (\mathcal{H}_b^i, \ BK) \quad \text{where} \quad \mathcal{B} \overset{k}{\sim} E. \tag{5}$$

Here, $\mathcal{B}$ is a batch of $k$ randomized examples, and $\mathcal{H}_b^i$ is the hypotheses at batch iteration $b$. Feedback $l_{\text{fdb}}^{i-1}$ from the previous interpretation step $i-1$, provided by the hypothesis interpreter $\Psi_{\text{interp}}$, guides the update of $\mathcal{H}^i$. The $l_{\text{sub}}$ explicitly derives $BK$, which serves as intermediate rationales chaining the $E$ to $\mathcal{H}$. The generated $BK$ is then reused by $\Psi_{\text{interp}}$ to validate $\mathcal{H}_b^i$ as input for symbolic tool.

**Hypothesis interpreter.** To validate the hypotheses $\mathcal{H}^i$, we employ a symbolic tool (i.e., ASP solver) to assess the interpretability of each hypothesis $H$ for including the positive examples $E^+$ and excluding the negative examples $E^-$. We define the hypothesis interpreter $\Psi_{\text{interp}} : (E, \ \mathcal{H}^i, \ BK) \mapsto (\hat{H}, \ l_{\text{fdb}})$, where $\hat{H}$ is the optimized hypothesis, and $l_{\text{fdb}}$ is a feedback for the hypothesis generator $\Phi_{\text{hyp}}$. The $\hat{H}$ and $l_{\text{fdb}}$ are determined by:

$$l_{\text{fdb}} = \begin{cases} \text{"satisfy"} & \text{if } i = \text{itermax} \\ \text{feedback with HI}(\hat{H}) & \text{otherwise} \end{cases} \tag{6}$$

where $\hat{H} = \arg\max_{H \in \mathcal{H}^i} \text{HI}(H)$ and the scoring function HI is defined as:

$$\text{HI}(H; E, BK) = f_{\text{TPR}}(H, E^+, BK) - f_{\text{FPR}}(H, E^-, BK). \tag{7}$$

If an interpretation step $i$ reaches its maximum, $\hat{H}$ is accepted as generalized knowledge $R$. The generalizability of a hypothesis across the entire $E$ is assessed using HI, following an approach similar to the contrastive learning objective (Oord et al., 2018). We define $f_{\text{FPR}}$ and $f_{\text{TPR}}$ as metric functions that evaluate the False Positive Rate (FPR) and True Positive Rate (TPR) for a given $H$, with respect to the $E$ and $BK$. In the formulation of $f_{\text{FPR}}$ and $f_{\text{TPR}}$, we prioritize examples from the current environment by assigning them higher weights than existing examples, thereby ensuring that knowledge improvement is aligned to the current environment. Details are provided in Appendix D.

### 3.3 RULE APPLICATION

As shown in Figure 2, generalized knowledge $R$ is used to complete embodied tasks specified by the user instruction $i$. Specifically, NESYC employs a symbolic tool and a memory-based monitoring scheme, utilizing ASP for action planning. During task execution, the error handler $\Phi_{\text{err}}$ manages interaction experiences from the environment via the action executor $\Phi_{\text{exe}}$, storing them in the working memory $\mathcal{M}$. If an inadmissible action is detected, $\Phi_{\text{err}}$ triggers the refinement of $R$ by re-entering the phase (i), where $\mathcal{M}$ is integrated into the experience set $\mathcal{T}$. With the refined $R$, NESYC effectively adapts to unpredictable situations. Algorithm 2 lists the rule application phase.

**Task planner.** In computing action plans, a symbolic tool that takes $(R, s_t, g)$ as input programs is used, following Tran et al. (2023); Aeronautiques et al. (1998), where $R$ is the generalized knowledge for action preconditions and effects, $s_t$ is a current state, and $g$ is goal state. Since the current and goal state is often not clearly specified in the environment, the semantic parser $\Phi_{\text{sem}}$ in Eq.(4) translates the trajectory $\sigma_t$ into a programmatic form of $s_t$, and the instruction $i$ into $g$. Based on these inputs, the task planner $\Psi_{\text{plan}}$ computes action plans $P$ to transition from $s_t$ to $g$, utilizing $R$. Formally, the $\Psi_{\text{plan}}$ is defined as $\Psi_{\text{plan}} : (R, s_t, g) \mapsto P$.

**Action executor.** From action plans $P$ deduced by task planner $\Psi_{\text{plan}}$, an individual plan can be chosen by action executor $\Phi_{\text{exe}}$ to perform relevant actions in the environment, starting from the current step $t$; i.e., $\Phi_{\text{exe}} : (P, s_t, g) \mapsto a_t$. Note that $\Phi_{\text{exe}}$ performs action $a_t$, sending observation $o_{t+1}$ from the environment along with the result of the action taken, to the error handler $\Phi_{\text{err}}$.

**Error handler.** To maintain consistency between the predicted state changes and actual observations, the error handler $\Phi_{\text{err}}$ monitors task execution using the memory-based retention of trajectory

samples. Due to the dynamic nature of embodied environments, planning based on $\Psi_{\text{plan}}$ often falls short of task completion. Based on the execution results, $\Phi_{\text{err}}$ measures action affordance $c_t$ and rewrites the next observation $o_{t+1}$ to provide a more attentive representation of the environment. We define $\Phi_{\text{err}}$ to trigger the refinement of generalized knowledge $R$ based on action affordance $c_t$.

$$\text{Next phase} = \begin{cases} \text{Phase (ii)}, & \text{if } c_t = 1 \\ \text{Phase (i)}, & \text{if } c_t = 0 \end{cases} \quad \text{where} \quad \Phi_{\text{err}} : (\sigma_t, a_t, o_{t+1}) \mapsto c_t, o_{t+1} \tag{8}$$

When $c_t = 1$, even though the action is successfully executed, the changes in the observations might invalidate the preconditions for the next action. To resolve this, $\Phi_{\text{err}}$ updates the current observation via $\Phi_{\text{sem}}$, and $\Psi_{\text{plan}}$ re-plans accordingly. When $c_t = 0$, the action fails, necessitating the refinement of generalized knowledge $R$. $\Phi_{\text{err}}$ appends all pairs of $(\sigma_{t+1}, c_t)$ to working memory $\mathcal{M}$, including those for $c_t = 0$. This robustly refines $R$ by re-entering the phase (i) with the updated experience set $\mathcal{T} = \mathcal{T} \cup \mathcal{M}$, continually improving the understanding on the environment.

---

**Algorithm 1** Rule Reformulation

**Agent**: $\Phi_{\text{sem}}, \Phi_{\text{hyp}}, \Psi_{\text{interp}}$
Experience set $\mathcal{T}$
Example set $E \leftarrow \emptyset$
Hypotheses $\mathcal{H} \leftarrow \emptyset$
Optimized hypothesis $\hat{H} \leftarrow \emptyset$
Generalized knowledge $R \leftarrow \emptyset$
Subsumption CoT prompt $l_{\text{sub}}$
Feedback prompt $l_{\text{fdb}} = $ ""

1: **for all** $\sigma, c \in \mathcal{T}$ **do**
2: $\quad E \leftarrow E \cup \Phi_{\text{sem}}(\sigma)$
3: **end for**
4: **while** $\hat{H} = \emptyset$ **do**
5: $\quad$ **for all** batch $\mathcal{B} \in E$ **do**
6: $\quad\quad \mathcal{H}, BK \leftarrow \Phi_{\text{hyp}}(\mathcal{B}, \mathcal{H}, l_{\text{sub}}, l_{\text{fdb}})$
7: $\quad$ **end for**
8: $\quad \hat{H}, l_{\text{fdb}} \leftarrow \Psi_{\text{interp}}(E, \mathcal{H}, BK)$
9: $\quad$ **if** $l_{\text{fdb}}$ is not "satisfy" **then**
10: $\quad\quad \mathcal{H} \leftarrow \hat{H}, \hat{H} \leftarrow \emptyset$
11: $\quad$ **end if**
12: **end while**
13: **return** $R \leftarrow \hat{H}$

---

**Algorithm 2** Rule Application

**Agent**: $\Phi_{\text{sem}}, \Psi_{\text{plan}}, \Phi_{\text{exe}}, \Phi_{\text{err}}$
Experience set $\mathcal{T}$, Generalized knowledge $R$
Working memory $\mathcal{M} \leftarrow \emptyset$
trajectory $\sigma \leftarrow [\,]$

1: $t \leftarrow 0, (o_t, i) \leftarrow env.\text{reset}()$
2: $\sigma \leftarrow \sigma.\text{append}(o_t)$
3: **for** $1 \leq t \leq \text{itermax}$ **do**
4: $\quad (s_{t-1}, a_{t-1}, s_t) \leftarrow \Phi_{\text{sem}}(\sigma)$
5: $\quad g \leftarrow \Phi_{\text{sem}}(i)$
6: $\quad P \leftarrow \Psi_{\text{plan}}(R, s_t, g)$
7: $\quad a_t \leftarrow \Phi_{\text{exe}}(P)$
8: $\quad o_{t+1} \leftarrow env.\text{step}(a_t)$
9: $\quad c_t, o_{t+1} \leftarrow \Phi_{\text{err}}(\sigma, a_t, o_{t+1})$
10: $\quad \sigma \leftarrow \sigma.\text{concat}([a_t, o_{t+1}])$
11: $\quad \mathcal{M} \leftarrow \mathcal{M}.\text{append}((\sigma, c_t))$
12: $\quad$ **if** $c_t = 0$ **then**
13: $\quad\quad \mathcal{T} \leftarrow \mathcal{T} \cup \mathcal{M}, \mathcal{H} \leftarrow R$
14: $\quad\quad R \leftarrow$ Re-entering phase (i)
15: $\quad\quad \mathcal{M} \leftarrow \emptyset, \sigma \leftarrow [o_{t+1}]$
16: $\quad$ **end if**
17: **end for**

---

## 4 EVALUATION

### 4.1 EXPERIMENT SETTING

**Environments.** For evaluation, we utilize several embodied benchmarks such as ALFWorld, VirtualHome, Minecraft, and RLBench. Additionally, we conduct experiments with a real-world robot to demonstrate NESYC's effectiveness and applicability in real-world complex tasks. For open-domain evaluation, we use three environment settings, categorized by their level of dynamics, which result in significant state changes. In a *Static* setting, object states, goal conditions and action effects are consistent across episodes. In a *Low Dynamic* setting, object states change unpredictably within an episode, though goal conditions and action preconditions remain consistent. In a *High Dynamic* setting, both object states, goal conditions, and even the preconditions of actions change unpredictably within an episode. For each task, we generate rephrased instructions using ChatGPT (Ouyang et al., 2022) based on the templated instructions from each benchmark, similar to Szot et al. (2023).

**Dataset.** We utilize a few expert-level episodic experience data from environments with conditions identical to the *Static* setting. Each experience captures state transitions, including an initial observation, action taken, execution result, and resulting next observation. Note that this experience dataset represents about 7% of the evaluation task episodes used in our experiments.

**Evaluation metrics.** We use several evaluation metrics, consistent with prior works (Shridhar et al., 2020b;c). SR (%) measures the percentage of tasks successfully completed, defined as meeting all

Table 1: Performance comparison of open-domain embodied task planning for static and two dynamic environment configurations. Variations for each metric are reported with three seeds.

| ALFWorld | Static | | | Low Dynamic | | | High Dynamic | | |
|---|---|---|---|---|---|---|---|---|---|
| Methods | SR | GC | Step | SR | GC | Step | SR | GC | Step |
| LLM-planner | 10.6±2.8 | 17.7±3.4 | 20.9±3.7 | 9.8±2.7 | 22.2±3.8 | 26.8±4.0 | 7.3±2.4 | 17.1±3.4 | 21.1±3.7 |
| ReAct | 35.8±3.5 | 48.2±4.1 | 51.7±4.3 | 34.1±4.3 | 45.2±4.5 | 50.6±4.5 | 18.7±3.5 | 28.1±4.1 | 33.5±4.3 |
| Reflexion | 39.0±3.7 | 63.5±4.4 | 67.0±4.5 | 37.4±4.4 | 64.8±4.3 | 70.6±4.1 | 21.1±4.4 | 41.5±4.3 | 43.6±4.2 |
| AutoGen | 58.5±4.4 | 77.8±3.7 | 81.1±3.5 | 51.2±4.5 | 69.3±3.9 | 75.6±4.2 | 30.9±4.2 | 50.6±4.5 | 58.8±4.4 |
| CLMASP | **88.5±2.9** | **89.1±2.8** | **89.1±2.8** | 58.8±4.4 | 63.3±4.4 | 71.8±4.1 | 23.8±3.9 | 36.5±4.4 | 45.8±4.5 |
| NESYC | 82.9±3.4 | 83.5±3.4 | 83.6±3.3 | **78.9±3.7** | **79.4±3.7** | **80.0±3.6** | **70.7±4.1** | **75.5±3.9** | **76.4±3.8** |

| VirtualHome | Static | | | Low Dynamic | | | High Dynamic | | |
|---|---|---|---|---|---|---|---|---|---|
| Methods | SR | GC | Step | SR | GC | Step | SR | GC | Step |
| LLM-planner | 21.5±0.5 | 33.2±0.4 | 33.0±9.7 | 20.5±0.6 | 32.7±0.8 | 29.8±2.7 | 14.8±3.4 | 27.7±2.9 | 21.5±2.2 |
| ReAct | 40.0±5.0 | 51.9±4.5 | 44.8±0.8 | 34.6±3.6 | 46.8±3.4 | 35.9±3.7 | 17.2±2.1 | 32.4±1.2 | 18.8±2.0 |
| Reflexion | 36.2±1.7 | 47.5±2.0 | 16.4±1.1 | 35.4±1.9 | 46.6±1.0 | 36.5±1.8 | 15.5±0.9 | 29.7±1.6 | 16.4±1.1 |
| AutoGen | 44.3±2.2 | 54.8±2.7 | 45.8±2.2 | 43.2±1.2 | 54.4±1.4 | 44.9±1.2 | 18.9±0.9 | 33.0±1.7 | 20.6±1.0 |
| CLMASP | 76.4±0.8 | **89.1±0.8** | 84.1±0.0 | 28.9±0.4 | 42.5±0.2 | 28.9±0.4 | 0.0±0.0 | 13.1±0.0 | 0.0±0.0 |
| NESYC | **82.3±0.4** | 87.4±0.6 | **84.2±0.6** | **79.6±1.9** | **85.8±1.1** | **80.8±1.9** | **77.5±1.3** | **84.1±0.6** | **79.0±1.2** |

| Minecraft | Static | | | Low Dynamic | | | High Dynamic | | |
|---|---|---|---|---|---|---|---|---|---|
| Methods | SR | GC | Step | SR | GC | Step | SR | GC | Step |
| LLM-planner | 31.1±1.5 | 41.2±1.4 | 42.5±2.4 | 28.9±4.2 | 31.9±1.6 | 37.0±1.6 | 23.3±2.7 | 25.8±1.3 | 28.9±0.4 |
| ReAct | 34.4±1.6 | 38.3±1.4 | 44.4±2.7 | 27.8±1.6 | 31.6±1.7 | 40.5±4.0 | 21.1±1.6 | 24.7±2.8 | 30.6±3.6 |
| Reflexion | 41.1±1.6 | 47.2±1.3 | 49.0±1.6 | 30.0±2.7 | 34.0±2.7 | 39.1±2.3 | 21.1±1.6 | 24.0±2.4 | 30.9±4.0 |
| AutoGen | 51.1±4.2 | 52.2±3.4 | 53.9±2.8 | 33.3±2.7 | 36.6±2.4 | 38.2±2.5 | 25.6±3.1 | 28.3±3.6 | 32.7±4.2 |
| CLMASP | **94.4±3.1** | **95.4±1.7** | **95.8±1.3** | 52.2±5.7 | 55.7±6.4 | 59.1±5.5 | 48.9±3.1 | 50.9±3.6 | 52.8±2.8 |
| NESYC | 92.2±1.6 | 94.3±0.9 | 95.3±1.0 | **91.1±1.6** | **93.2±1.4** | **94.1±1.4** | **87.8±5.7** | **89.9±5.9** | **90.9±5.9** |

| RLbench | Static | | | Low Dynamic | | | High Dynamic | | |
|---|---|---|---|---|---|---|---|---|---|
| Methods | SR | GC | Step | SR | GC | Step | SR | GC | Step |
| LLM-planner | 16.7±5.7 | 23.3±2.9 | 35.5±1.9 | 16.7±2.9 | 20.8±1.4 | 27.4±1.0 | 18.3±2.9 | 21.7±1.4 | 27.2±1.9 |
| ReAct | 23.3±2.9 | 25.8±1.4 | 36.5±1.7 | 21.7±2.8 | 23.3±1.4 | 30.8±1.6 | 18.3±2.9 | 20.0±2.5 | 26.8±2.5 |
| Reflexion | 33.3±2.8 | 41.4±5.1 | 47.5±3.3 | 21.7±2.9 | 24.4±2.1 | 32.1±2.5 | 23.3±2.8 | 23.3±2.8 | 29.6±2.8 |
| AutoGen | 43.3±8.6 | 54.2±4.6 | 57.9±3.3 | 23.3±2.9 | 28.6±2.7 | 32.1±2.3 | 21.7±5.8 | 23.3±2.9 | 28.9±1.9 |
| CLMASP | **94.5±4.2** | **95.8±2.8** | **96.0±2.7** | 0.0±0.0 | 6.0±0.8 | 25.0±2.0 | 0.0±0.0 | 3.7±0.7 | 12.3±0.9 |
| NESYC | 85.5±2.7 | 88.5±0.9 | 91.9±0.7 | **81.5±4.5** | **84.8±4.5** | **88.7±3.6** | **79.0±6.6** | **81.8±7.0** | **86.2±6.2** |

goal conditions. GC (%) measures the success rate of individual goal conditions. Step (%) measures the percentage of the action sequences that align with the ground-truth sequence from the start.

**Baselines.** We implement several baselines, categorized into three groups: i) an LLM-based planning method **LLM-planner** (Song et al., 2023). ii) Multi-agent frameworks including **ReAct** (Yao et al., 2023), **Reflexion** (Shinn et al., 2024), and **AutoGen** (Wu et al., 2023a). iii) neuro-symbolic approaches such as **ProgPrompt** (Singh et al., 2023) and **CLMASP** (Lin et al., 2024).

**NESYC implementation.** For the symbolic tool, we use the ASP solver *clingo* (Lifschitz, 2019) (version 5.7.1). The LLM mainly used is GPT-4o (version gpt-4o-2024-08-06) with temperature 0. For fair comparisons, the same LLM configuration is applied across all baselines.

Detailed explanations of the experiment settings are in Appendix B.

## 4.2 MAIN RESULTS

In Table 1, we evaluate the performance of embodied task planning in open-domain settings, comparing each method's action plans based on their use of experiences, primarily through in-context learning. NESYC consistently outperforms the most competitive baseline, **AutoGen**, across all test settings (*Static*, *Low Dynamic*, and *High Dynamic*) and on evaluation metrics (SR, GC, and Step). Specifically, NESYC achieves an average improvement of 45.2% in SR, 38.7% in GC, and 38.4% in Step across the evaluated benchmarks.

In the *Static* setting, the conventional neuro-symbolic approach, **CLMASP**, outperforms NESYC due to its use of additional expert-level knowledge tailored to the given environment. However, NESYC achieves comparable performance by generalizing knowledge solely from the provided experience data. As the dynamicity of the environment increases, **CLMASP**, which lacks the ability to reformulate its knowledge, exhibits a noticeable decline in performance, even falling behind LLM-based approaches. In contrast, NESYC remains robust across a range of open-domain settings, including both *Low Dynamic* and *High Dynamic* environments. By continually refining generalized

knowledge from accumulated experiences, NESYC maintains consistent performance across the benchmarks that involve varying action types and environmental dynamics. In RLBench, specifically, the focus is on fine-grained physical control and interaction, constrained by factors such as actuator range, grip force, and balance. These precise physical constraints, which are critical for task success, pose significant challenges for LLM-based approaches and can lead to substantial performance drops even with subtle environmental changes for neuro-symbolic agents. In contrast, NESYC controls a robotic arm with precision and stability by continually refining its knowledge, resulting in robust performance across all open-domain settings.

## 4.3 ANALYSIS

Table 2: Performance evaluation on robustness to experience incompleteness. 'Logic Exp.' denotes the logic expression (i.e., Natural Language, Imperative Programming, or Declarative Programming). 'Refine' indicates if the logic is refined, with ✖ for no refinement and ✔ for refinement.

| ALFWorld | | *Complete* Experience Set | | | *Noisy* Experience Set | | | *Imperfect* Experience Set | | |
|---|---|---|---|---|---|---|---|---|---|---|
| Logic Exp. | Method | SR | GC | Refine | SR | GC | Refine | SR | GC | Refine |
| NL | Autogen | 54.6±8.7 | 73.7±7.7 | ✖ | 54.6±8.7 | 63.4±8.4 | ✖ | 57.6±8.6 | 76.0±7.4 | ✖ |
| Imperative | ProgPrompt | 72.7±7.8 | **98.5±2.1** | ✖ | 48.5±8.7 | 61.1±8.5 | ✖ | 48.5±8.2 | 67.4±8.2 | ✖ |
| Declarative | CLMASP | **97.0±3.0** | 98.0±2.5 | ✖ | 69.7±8.0 | 78.8±7.1 | ✖ | 54.6±8.7 | 68.2±8.1 | ✖ |
| | NESYC | 90.9±5.0 | 96.0±3.4 | ✔ | **90.9±5.0** | **91.9±4.7** | ✔ | **84.9±6.2** | **89.9±5.3** | ✔ |

**Robustness to experience incompleteness.** In Table 2, we evaluate the robustness of NESYC upon varied experience quality. The experience sets are categorized by their incompleteness: *Complete* includes sufficient actionable knowledge, *Noisy* contains mislabeled action affordances, and *Imperfect* omits some actionable knowledge. Although the baselines use expert knowledge in different forms, they similarly assume that provided knowledge is either noisy or imperfect. In the *Complete* case, with full expert knowledge, **ProgPrompt** and **CLMASP** perform best in GC and SR, respectively. However, NESYC not only achieves task planning performance comparable to **CLMASP** with complete experiences but also demonstrates robust performance with incomplete experiences through its knowledge generalization strategy.

| LLM | SR | ➜ SR | GC | ➜ GC | HI | ➜ HI |
|---|---|---|---|---|---|---|
| ▬ Llama-3-8B | 43.9 | → 40.2 | 43.9 | → 40.2 | 0.344 | → 0.378 |
| ▬ GPT-4o-mini | 41.9 | → 78.7 | 44.7 | → 79.3 | 0.634 | → 0.697 |
| ▬ Claude-3.0-Opus | 50.7 | → 76.7 | 53.6 | → 78.6 | 0.516 | → 0.567 |
| ▬ Claude-3.5-Sonet | 51.4 | → 78.4 | 54.2 | → 80.2 | 0.559 | → 0.615 |
| ▬ Llama-3-70B | 58.8 | → 85.1 | 60.4 | → 86.6 | 0.709 | → 0.762 |
| ▬ GPT-4o | 64.2 | → 90.2 | 67.0 | → 90.5 | 0.752 | → 0.786 |
| ▬ GPT-4 | 69.6 | → 89.2 | 73.3 | → 89.8 | 0.784 | → 0.804 |

Table 3: Contrastive generality improvement scheme evaluation on environments. SR, GC, and HI measure the performance of the generalized knowledge $R$ on the first interpretation step. ➜ **SR**, ➜ **GC** and ➜ **HI** report scores after iterative adjustment.

Figure 3: HI score evaluation.

**Impact by different LLMs.** We examine the dependency of the contrastive generality improvement scheme on various LLMs in the phase (i). Table 3 specifies the impact of different LLMs on the planning performance, measured in SR, GC, and HI between each initial hypothesis (i.e., SR, GC, HI in the table) and its corresponding updated one (i.e., ➜ **SR**, ➜ **GC** and ➜ **HI**) after the improvement process. As shown, updated hypotheses consistently achieve performance gains in planning for all tested LLMs with the exception of smaller Llama-3-8B. These results indicate the robustness of our contrastive generality improvement across capable LLMs. However, smaller models like Llama-3-8B reveal certain limitations in the scheme's effectiveness. In Figure 3, we further illustrate the consistency of these results, emphasizing the alignment between the HI score of the updated hypotheses and other traditional metrics.

**Comparison on different dynamics predicates.** Figure 4 shows differences in performance with respect to different dynamics predicates in Table 4. The performance is based on the comparison between expert-level actionable knowledge, initial hypotheses, and updated hypotheses. In Figure 4, the blue arrows indicate the improved performance of updated hypotheses from their respective initial ones. As shown, the *Attribute* dynamics predicates, which pertain to relatively static features,

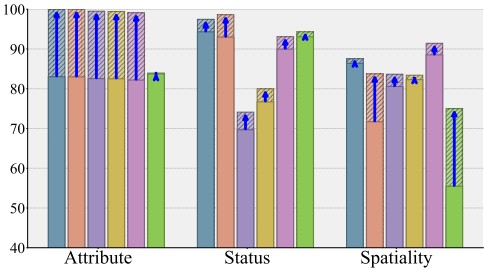

Figure 4: F1 score evaluation on predicate categories. Colors indicating LLMs are consistent with those used for different LLMs in Table 3

| Category | Num. | Predicates |
|---|---|---|
| Attribute | 20 | grabbable, cuttable, can_open, readable, has_paper, movable, pourable, cream, has_switch, has_plug, drinkable,lookable, body_part, surfaces, sittable, lieable, person, hangable, clothes, eatable |
| Status | 10 | closed, open, plugged_out, plugged_in, on, off, sitting, lying, clean, dirty |
| Spatiality | 9 | obj_ontop, ontop, inside_room, obj_inside, inside, on_char, obj_next_to, next_to, between |

Table 4: Dynamics Predicates. The predicates are categorized based on their dynamics.

consistently achieve high F1 scores; conversely, *Status* and *Spatiality* dynamics predicates show variability due to their dynamic natures on physical states and spatial relations. While the improvement varies across different predicates and LLMs, predicates related to common-sense knowledge tend to show better improvement overall.

Table 5: Ablation on two reasoning components of NESYC. 'w/o. $l_{sub}$' refers to replacing $l_{sub}$ with a simple LLM prompt, and 'w/o. ASP sol.' indicates using LLM prompting for planning instead of the ASP solver. The symbol '→' denotes replacement with another LLM prompting technique.

| **ALFWorld** | *Static* | | | *Low Dynamic* | | | *High Dynamic* | | |
|---|---|---|---|---|---|---|---|---|---|
| Method | SR | GC | Step | SR | GC | Step | SR | GC | Step |
| NESYC | **82.9±3.4** | 83.5±3.4 | 83.6±3.3 | **78.9±3.7** | 79.4±3.7 | 80.0±3.6 | **70.7±4.1** | **75.5±3.9** | **76.4±3.8** |
| w/o. $l_{sub}$ | 41.5±4.4 | 45.7±4.5 | 51.7±4.5 | 39.0±4.4 | 42.1±4.5 | 48.4±4.5 | 36.6±4.3 | 43.3±4.5 | 49.3±4.5 |
| ASP sol. → SymbCoT | 71.5±4.1 | **89.8±2.7** | **92.3±2.4** | 60.2±4.4 | **80.0±3.6** | **85.3±3.2** | 27.6±4.0 | 48.0±4.5 | 58.9±4.4 |
| ASP sol. → CoT | 69.1±4.2 | 87.6±3.0 | 89.2±2.8 | 58.5±4.4 | 77.2±3.8 | 81.5±3.5 | 24.4±3.9 | 49.9±4.5 | 59.3±4.4 |
| w/o. ASP sol. | 48.8±4.5 | 77.7±3.8 | 81.7±3.5 | 35.8±4.3 | 59.1±4.4 | 70.0±4.1 | 17.1±3.4 | 41.5±4.4 | 51.4±4.5 |
| w/o. $l_{sub}$ & ASP sol. | 25.2±3.9 | 57.9±4.5 | 66.3±4.3 | 9.8±2.7 | 30.6±4.2 | 41.2±4.4 | 3.3±1.6 | 19.8±3.6 | 27.3±4.0 |

**Ablation study.** In Table 5, we conduct an ablation study assessing the impact of reasoning components in NESYC, specifically the subsumption CoT prompt $l_{sub}$ and the ASP solver used for planning. The w/o. $l_{sub}$, NESYC experiences an average 38.5% decrease in SR, underscoring the importance of structured prompting for generating background knowledge and hypotheses. The next three rows present the results when we replace the ASP solver in the task planner $\Psi_{plan}$ with different LLM-based reasoning methods: 'ASP sol. → SymbCoT' employs an LLM as a symbolic reasoning tool, following the symbolic CoT (Xu et al., 2024), 'ASP sol. → CoT' refers to the use of standard CoT (Wei et al., 2022), and 'w/o. ASP sol.' represents a naive prompting without an ASP solver. Using an LLM in place of the ASP solver can simplify planning, often improving GC and Step performance for shorter sequences by reducing the strictness required by symbolic tools. However, they exhibit lower reliability in task completion, as indicated by reduced SR. Furthermore, replacing both $l_{sub}$ and the ASP solver with simple LLM prompting, denoted as 'w/o. $l_{sub}$ & ASP sol.', leads to a significant degradation in task performance.

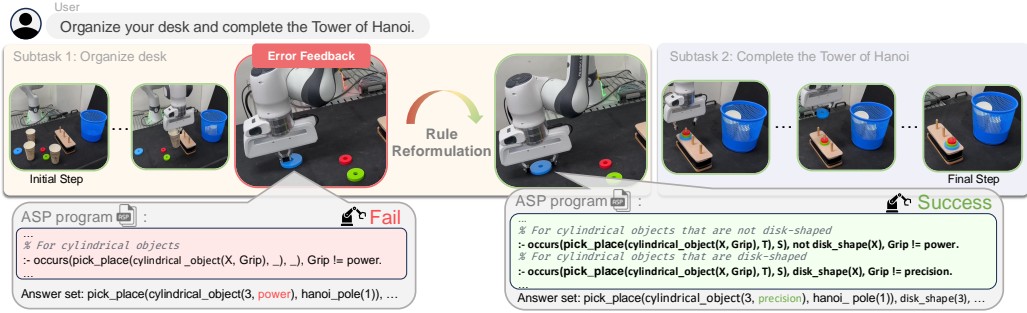

Figure 5: Real-world desk rearrangement tasks. Initially, NESYC does not include knowledge for picking up Hanoi blocks from experiences. After failures, NESYC refines to enhance grasping capabilities, enabling the robot to successfully complete the desk rearrangement task.

**Real-word scenario.** In Figure 5, we illustrate the real-world experimental setup for demonstrating the practical applicability of NESYC. The task involves rearranging a desk, with scattered blocks of a Hanoi Tower and other objects. Using the same experience set of the RLBench experiments, NESYC restructures actionable knowledge for the real-world robot and refines the knowledge to handle unfamiliar objects, such as the Hanoi blocks. The robot successfully completes the instruction, highlighting the practical applicability of NESYC.

Table 6: Comparison of LLM feedback and Human feedback. In *Binary* case, the LLM solely calculates action affordances but does not rewrite the next observation. In *Cause* case, the Human and LLM, respectively, calculate action affordances and rewrite the next observation. In *Guidance* case, the Human and LLM additionally incorporate corrected experiences.

| **Real-world** | *Static* | | *Dynamic* | | **Real-world** | *Static* | | *Dynamic* | | **Real-world** | *Static* | | *Dynamic* | |
|---|---|---|---|---|---|---|---|---|---|---|---|---|---|---|
| *Binary* | SR | GC | SR | GC | *Cause* | SR | GC | SR | GC | *Guidance* | SR | GC | SR | GC |
| LLM | 22.2 | 59.2 | 11.1 | 42.4 | LLM | 66.6 | 87.0 | 44.4 | 79.7 | LLM | 55.6 | 81.4 | 33.3 | 72.1 |
| | | | | | Human | 77.8 | 92.6 | 55.6 | 85.2 | Human | 88.9 | 98.1 | 66.7 | 94.3 |

In Table 6, we compare different types of feedback integrated into experiences via memory-based monitoring. In both the *Binary* and *Cause* cases, we observe that the error handler $\Phi_{\text{err}}$ effectively refines actionable knowledge, comparable to high-quality human feedback, by directly measuring action affordances and rewriting the next observations based on the action taken. However, in the *Guidance* case, while accurate guidance from humans is effective, LLM guidance often adds errors in experience representations, resulting in performance degradation.

## 5 RELATED WORK

LLM-based task planning approaches have opened new avenues for leveraging linguistic knowledge to guide agent behaviors in embodied environments (Brohan et al., 2023; Huang et al., 2023b; Song et al., 2023; Driess et al., 2023; Zhao et al., 2024; Singh et al., 2023; Wu et al., 2023b; Wang et al., 2023). Meanwhile, neuro-symbolic systems combine the capabilities of neural networks with symbolic reasoning tools to enhance explainability, reliability, and flexibility (Olausson et al., 2023; Pan et al., 2023; Fang et al., 2024; Yang et al., 2023; Ishay et al., 2023). These systems often rely on fully defined symbolic knowledge for embodied control (Lin et al., 2024; Liu et al., 2023; Agarwal et al., 2024; Cornelio & Diab, 2024), which constrains their applicability and effectiveness in open domains. Recent advancements in LLM-based multi-agent frameworks have enhanced problem-solving capabilities by fostering the collaborative interaction between agents, external tools, and environments (Yao et al., 2024; Hao et al., 2023; Shinn et al., 2024; Yao et al., 2023; Wu et al., 2023a). Further details on related work are in Appendix C.

## 6 CONCLUSION

We presented the NESYC framework to enable effective embodied task planning in open domains by continually generalizing actionable knowledge from experiences. The framework adapts neuro-symbolic approaches via two schemes, contrastive generality improvement, and memory-based monitoring, which enable the interleaving of inductive and deductive knowledge refinement in a continual learning manner. Experiments on ALFWorld, VirtualHome, Minecraft, RLBench, and real-world robotic scenarios demonstrate the robustness and applicability of NESYC across diverse open domains. In static settings, although using pre-defined expert knowledge involves trade-offs, our model still shows clear advantages over other LLM-based and neuro-symbolic approaches.

**Limitation and future work.** As reported in Figure 3 and Table 3, NESYC encounters difficulties when applied to smaller LLMs, such as Llama-3-8B; performance improvements are rarely achieved due to persistent rule conflicts and errors during the rule reformulation phase. We plan to explore neuro-symbolic knowledge distillation for resource-efficient embodied control with smaller LLMs.

**Ethical concerns.** LLMs operating in environments with hazardous tools (e.g., knives and forks) can lead to unsafe outcomes if errors occur. Therefore, strict and transparent safety guidelines must be implemented to verify all outputs, and we are committed to providing them.

ACKNOWLEDGEMENT

This work was supported by Institute of Information & communications Technology Planning & Evaluation (IITP) grant funded by the Korea government (MSIT), (RS-2022-II220043 (2022-0-00043), Adaptive Personality for Intelligent Agents, RS-2022-II221045 (2022-0-01045), Self-directed multi-modal Intelligence for solving unknown, open domain problems, RS-2025-02218768, Accelerated Insight Reasoning via Continual Learning, and RS-2019-II190421, Artificial Intelligence Graduate School Program (Sungkyunkwan University)), the National Research Foundation of Korea (NRF) grant funded by the Korea government (MSIT) (No. RS-2023-00213118), IITP-ITRC (Information Technology Research Center) grant funded by the Korea government (MIST) (IITP-2025-RS-2024-00437633, 10%), IITP-ICT Creative Consilience Program grant funded by the Korea government (MSIT) (IITP-2025-RS-2020-II201821, 10%), and by Samsung Electronics.

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

## A    RELATED WORK

**Embodied Control with Large-Language Models.** Integrating LLMs into embodied control systems has opened new opportunities to leverage extensive linguistic knowledge to guide agent behavior. LLMs offer flexible and generalizable control of embodied agents in realistic environments, including real-world scenarios (Brohan et al., 2023; Huang et al., 2023b; Song et al., 2023; Driess et al., 2023; Zhao et al., 2024; Singh et al., 2023; Wu et al., 2023b; Wang et al., 2023). Our work aims to achieve robust performance in LLM-based embodied control within open-domain environments by integrating the strengths of LLM common-sense reasoning and symbolic reasoning tools.

**Neuro-Symbolic Approaches.** Neuro-symbolic approaches combine the strengths of neural networks with symbolic systems, enhancing explainability, generalizability, and flexibility. Recently, systems augmented with LLMs have significantly advanced in solving traditional logic problems within Natural Language Processing domains (Olausson et al., 2023; Pan et al., 2023; Fang et al., 2024; Yang et al., 2023; Ishay et al., 2023). Research interest is growing in adapting neuro-symbolic approaches to embodied domains (Lin et al., 2024; Liu et al., 2023; Agarwal et al., 2024; Cornelio & Diab, 2024). While existing approaches depend on the complete provision of expert-level symbolic knowledge for embodied control, our work addresses scenarios where this knowledge is insufficient or inapplicable due to the unpredictable nature of open-domain environments.

**Large Language Model-based multi-agent framework.** LLM-based multi-agent frameworks have recently garnered significant attention, with many works improving their problem-solving abilities through collaboration among autonomous agents. Recent advancements in multi-agent architecture show a trend toward integrating diverse techniques, where agents interact with external tools and environments to improve planning, execution, and iteration (Yao et al., 2024; Hao et al., 2023; Shinn et al., 2024; Yao et al., 2023; Wu et al., 2023a). Our work introduces a novel multi-agent framework that emulates the hypothetico-deductive model, with a focus on generalizing actionable knowledge in dynamic environments.

## B    ENVIRONMENT SETTINGS

### B.1    ALFWORLD

We use ALFWorld (Shridhar et al., 2020c), an advanced simulator that integrates the text-based interactive environment of TextWorld (Côté et al., 2018) with the visual and physical interaction capabilities of the ALFRED benchmark (Shridhar et al., 2020a). This integration bridges the gap between abstract reasoning and physical action. ALFWorld includes 108 different object types (e.g., bread) and 37 receptacle types (e.g., plate) spread across 120 diverse indoor scenes (e.g., kitchen). The platform supports 3554 unique tasks, each crafted by combining these elements with one of six instruction types (e.g., pick & place), such as "Put a keychain in a plate and then put them on a shelf." Details of the instructions and executable plans are provided in Table 7, and visualizations of various indoor scenes and observations in ALFWorld are depicted in Figure 6.

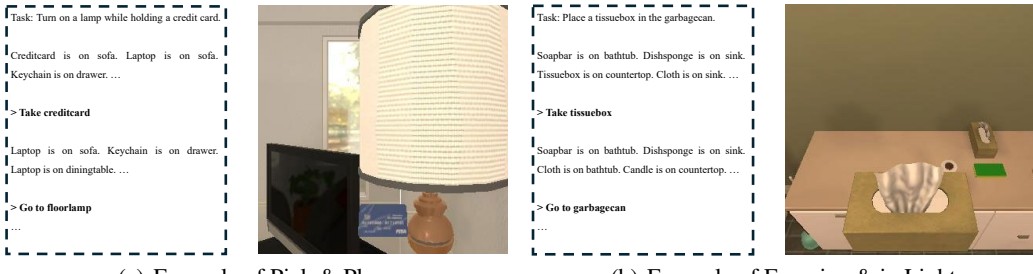

(a) Example of Pick & Place                    (b) Example of Examine & in Light

Figure 6: Task examples set of ALFWorld

Table 7: Instructions and executable plans in ALFWorld

|  | Type | Example |
|---|---|---|
| Instructions | Pick & Place | Apply spray bottle to toilet. |
|  | Pick Two & Place | Locate two glass bottles and place them on the shelf. |
|  | Clean & Place | Wash a mug and place it in the coffee machine. |
|  | Heat & Place | Refrigerate the heated tomato. |
|  | Cool & Place | Chill the wine bottle and place it on the dining table. |
|  | Examine & in Light | Examine the compact disc beneath the desk lamp. |
| Plans | Goto [Receptacle Object] | Goto countertop |
|  | Open [Receptacle Object] | Open garbagecan |
|  | Close [Receptacle Object] | Close garbagecan |
|  | Pickup [Object] [Receptacle Object] | Take cloth from countertop |
|  | Put [Object] [Receptacle Object] | Put plate in/on diningtable |
|  | Heat [Object] [Receptacle Object] | Heat mug with microwave |
|  | Cool [Object] [Receptacle Object] | Cool apple with fridge |
|  | Clean [Object] [Receptacle Object] | Clean tomato with sinkbassin |
|  | Slice [Object] [Instrument Object] | Slice tomato with knife |
|  | Examine [Object] | Examine cloth |
|  | Examine [Receptacle Object] | Examine countertop |
|  | End | Finish |

## B.2 VIRTUALHOME

We also utilize VirtualHome (Puig et al., 2018), a simulation platform designed for modeling every-day household activities in a 3D environment. This platform aids in the training and evaluation of agents who understand and execute complex task sequences based on language instructions. Virtual-Home includes 188 different object types (e.g., Fridge) across 7 unique environment IDs. It supports 2821 distinct tasks, each created by combining these elements with various instruction types, such as "Throw away newspaper". Details of the executable actions are provided in Table 8, and visualizations of various indoor scenes and observations are depicted in Figure 7.

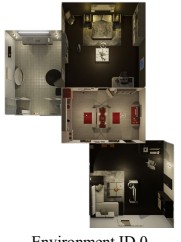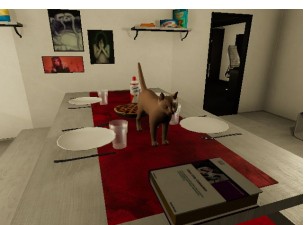 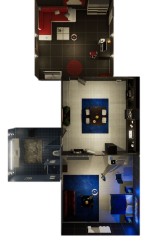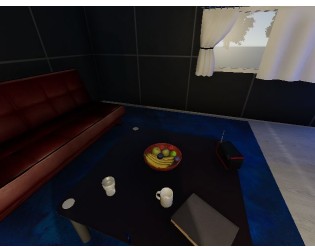

Environment ID 0     Observation        Environment ID 1     Observation

(a) Example of environment & observation      (b) Example of environment & observation

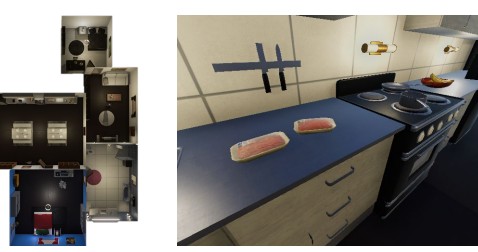

Environment ID 2     Observation

(c) Example of environment & observation

Figure 7: Environment examples set of VirtualHome

Table 8: Instructions and executable plans in VirtualHome

|  | Type | Example |
|---|---|---|
| Instructions | Pick & Place | Locate an apple on the kitchen table. |
|  | Pick & Place | Detect an apple and convey it onto the kitchen table. |
|  | Pick & Place | Can you place apple upon the kitchen table? |
|  | pick & Place | Undertake the endeavor to scout for the apple, hold it, move position the apple on the kitchen table. |
|  | Pick & Sit | Grab a book and sit on the bed. |
|  | Pick & Sit | Scour the room for the book, firmly grab it, seek the bed, and ease yourself onto the bed. |
|  | Pick & Sit | Begin the mission to fetch the book, seize the book, move to the bed, and relax into a seated position on the bed. |
|  | Pick & Sit | Embark upon the quest to get the book, pick the book, find the bed, and calmly take a seat on the bed. |
| Plans | TurnLeft | Turnleft |
|  | TurnRight | Turnright |
|  | StandUp | Standup |
|  | Walk [Object1] | Walk kitchen |
|  | Run [Object1] | Run kitchen |
|  | Walkforward | Walkforward |
|  | Walktowards [Object1] | Walktowards kitchen |
|  | Sit [Object1] | Sit chair |
|  | Grab [Object1] | Grab apple |
|  | Open [Object1] | Open fridge |
|  | Close [Object1] | Close fridge |
|  | SwitchOn [Object1] | Switchon stove |
|  | SwitchOff [Object1] | Switchoff stove |
|  | Drink [Object1] | Drink waterglass |
|  | Touch [Object1] | Touch stove |
|  | LookAt [Object1] | Lookat stove |
|  | Put [Object1] [Object2] | Putback apple table |
|  | PutIn [Object1] [Object2] | Putin apple fridge |

## B.3 MINECRAFT

In this study, we also utilize the Minecraft environment from PDDLGym (Silver & Chitnis, 2020) as a benchmark for testing task planning in complex outdoor scenarios. This environment is one of the seven classical planning domains written in PDDL that we empirically tested our approach on, as mentioned in the main text. Minecraft provides a unique setting for evaluating planning algorithms in a dynamic, open-world context. Key Features:

- **Outdoor Environment Simulation:** Unlike many indoor-based benchmarks, Minecraft simulates an outdoor environment, presenting challenges more akin to real-world scenarios. The environment includes fundamental materials such as wood, stone, and grass, with which agents must interact by moving, processing, and storing them.

- **Distinct Skill Requirements:** Tasks in Minecraft demand a different set of skills compared to indoor environments, including resource gathering, crafting, and construction. Additionally, agents need to continuously assess the current state of their tasks and adapt their strategies as the environment changes, requiring dynamic task management skills. The environment also provides sequentially complex task instructions that agents must interpret and execute. These instructions involve multiple steps that must be performed in a specific order.

These characteristics make the Minecraft environment particularly suitable for testing egocentric planning approaches. It challenges agents to operate effectively in a complex, dynamic world where the ability to adapt to changing circumstances, manage resources efficiently, and execute multi-step instructions is crucial. The open-world nature of Minecraft, coupled with the complexity of the provided commands, allows for the creation of diverse and challenging scenarios. This provides a

robust testbed for evaluating the flexibility and effectiveness of planning algorithms in environments that more closely resemble real-world outdoor settings and task complexities.

Details of the executable actions are provided in Table 9, and visualizations of various grid worlds and observations are depicted in Figure 8.

Table 9: Instructions and executable plans in Minecraft

|  | Type | Example |
|---|---|---|
| Instructions | Move & Equip | Move to location 2-4 and equip grass-0. |
|  | Collect & Move | Collect grass-1 and move to location 1-1. |
|  | Craft & Equip | Craft planks from new-1 and equip them. |
|  | Move & Inventory | Move to location 0-3 and store new-0 in the inventory. |
|  | Equip & Inventory | Equip grass-2 and store log-3 in the inventory. |
|  | Craft & Inventory | Craft planks from new-0 and store them in the inventory. |
| Plans | Recall [Object] | Recall log-1. |
|  | Move [Location] | Move loc-2-4. |
|  | CraftPlank [Object] [Object] | Craftplank new-0 log-1. |
|  | Equip [Object] | Equip grass-0. |
|  | Pick [Object] [Location] | Pick grass-1 from loc-1-1. |

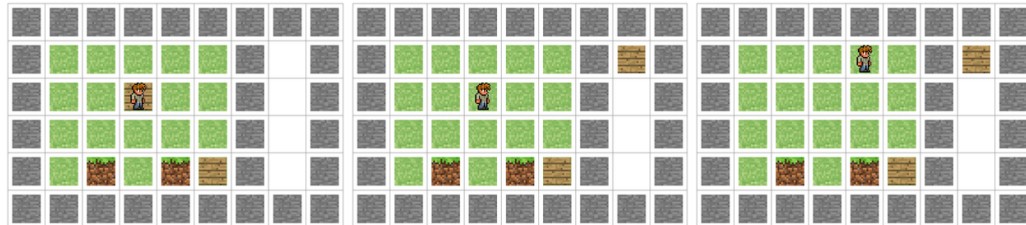

(a) Example of "Get new-0 item and go to loc-0-3."

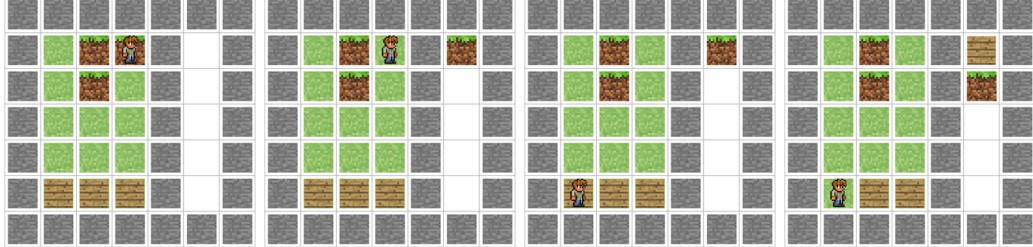

(b) Example of "Equip grass-2 and get log-3."

Figure 8: Environment examples set of Minecraft

### B.4 RLBENCH

We implement a tabletop manipulation environment using the RLBench (James et al., 2020) with a Franka Emika Panda 7-DoF robotic arm. This setup integrates precise robotic control with a rich, interactive environment, bridging the gap between abstract reasoning and physical manipulation. Our environment consists of 12 diverse objects arranged within a customizable workspace on a wooden table. The objects include everyday items and structural elements: a charger, 2 walls, steak and chicken, a grill, a wine bottle, a cap, 2 windows (right and left), and handles for each window. The position of each object is randomly assigned for each instance of the environment, creating unique configurations every time. The scene is illuminated by 3 directional lights to ensure consistent visual input. For perception, we employ a stereo camera system and a monocular wrist camera, providing RGB, depth, and segmentation mask data for each frame. Additionally, the system can

retrieve robot proprioceptive data, including joint angles, velocities, torques, and end-effector pose. Additionally, we used VoxPoser (Huang et al., 2023a) as a skill decoder to execute actions. By using this skill decoder, it became possible to execute open-set actions.

Details of the executable actions are provided in Table 10, and visualizations of various tabletop scenes and observations are depicted in Figure 9.

Table 10: Instructions and executable plans in RLBench

|  | **Type** | **Example** |
|---|---|---|
| Instructions | Pick & Place | Pick the chicken from the grill and place it in the goal place. |
|  | Pick & Move | Unplug the charger |
|  | Pick & Rotate | Open wine bottle. |
|  | Pick & Rotate & Move | Open the left window through the handle. |
| Plans | Grasp [Object] | Grasp cube. |
|  | Rotate [Direction] [Degree] | Rotate clockwise 100 degrees. |
|  | Move [Direction] [Distance] | Move forward 10cm. |
|  | Move To [Position] [Object] | Move to top of sphere. |
|  | Open Gripper | Open gripper. |
|  | Back To Default Pose | Back to default pose. |

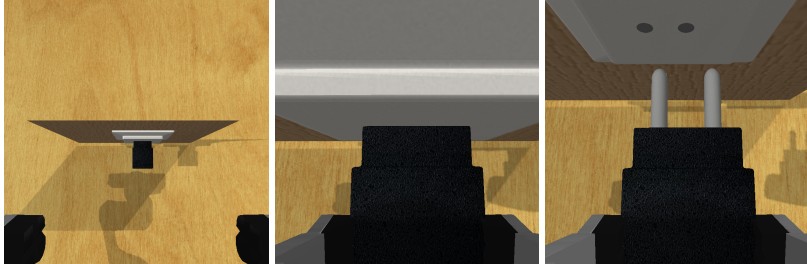

(a) Example of "Unplug charger" with gripper view

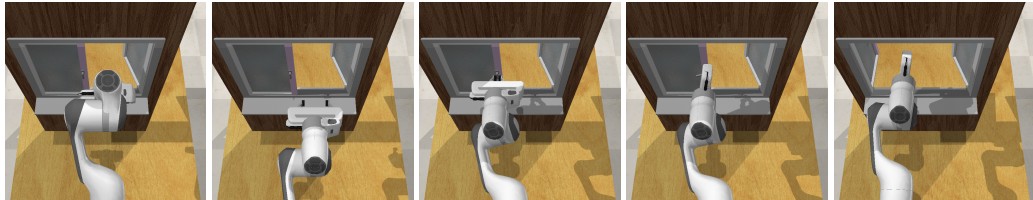

(b) Example of "Open the left window through the handle" with overhead view

Figure 9: Environment examples set of RLBench

### B.5 REAL-WORLD

**Setup and Implementation.** In our real-world experiments, we implemented a tabletop manipulation system using a Franka Emika Research 3 7-DoF robotic arm. The environment was equipped with an RGB-D camera(Intel RealSense D435) mounted for a top-down view, providing point cloud and RGB information for object detection and coordinate estimation.

**Environment Configuration.** Our experimental setup consisted of 10 common tabletop objects: three Hanoi tower blocks of varying sizes, three Hanoi tower poles, three paper cups, one plastic cup, one cardboard box, and a trash can. To ensure robustness, object positions were randomly assigned for each experimental instance, creating unique environmental configurations.

**Control System Implementation.** Unlike the RLBench experimental setting where VoxPoser was adapted, our real-world implementation utilized distinct methods for both perception and control:

- For perception, we integrated an RGB-D camera with an object detection module (Minderer et al., 2024) for accurate object coordinate identification:
  - A top-view image is captured using a camera positioned above the table.
  - The object detection module (Minderer et al., 2024) is used to identify the categories and bounding box coordinates of one or more target objects from the top-view image, with their heights calculated using a depth camera.
  - The bounding box and height coordinates of the target object(s) are converted into the robot arm's coordinate system.
- For low-level control, we employed MoveIt (Coleman et al., 2014), an open-source robot motion-planning framework:
  - Model Predictive Control (MPC) was implemented for trajectory optimization, which is highly effective in handling complex environments and constraints while ensuring real-time execution performance (Yu et al., 2023).
  - When one or more transformed target object coordinates are provided, they are input into predefined heuristic skill functions (e.g., 'move A to B', 'grasp A') to prioritize which target coordinates to approach first and assign waypoints for these target coordinates accordingly.
  - The inferred waypoints are fed into the motion planning algorithm (i.e., MPC) to compute the low-level action sequence.
- The robot executed movements using pre-defined primitive skills based on the generated plans.

While RLBench experiments used VoxPoser for low-level control with parameterized actions (e.g., action(rotate(clockwise, 100), T)), our real-world implementation adapted this approach to handle physical constraints. Although continuous force control for the gripper was not implemented, we developed a discrete mapping system that translates physical parameters into appropriate grasping strategies. An example of this mapping is provided in Table 12. The detailed executable actions are provided in Table 11, and the environmental setup is shown in Figure 10. The actual robot arm actions executed according to plans generated by NESYC can be observed in Figure 13.

Table 11: Instructions and executable plans in real-world experiments

|  | Type | Example |
|---|---|---|
| Instructions | Pick N & Place N
Pick N & Place N | Clean up the table.
Complete the Tower of Hanoi. |
| Plans | Grasp [Object]
Move [Object] to [Location]
Open Gripper
Back To Default Pose | Grasp log-1.
Move cup to trashbin.
Open gripper.
Back to default pose. |

## B.6 DATASETS

Table 12: Mapping of physical parameters to discrete semantic actions

| Semantic values for parameter Grip | Grab Region (Diameter Range) |
|---|---|
| precision | Center Point + 0–10% |
| focus | Center Point + 10–20% |
| standard | Center Point + 20–30% |
| balance | Center Point + 30–40% |
| power | Center Point + 40–50% |

In the static setting, the episodic data used for each environment is as follows: ALFWorld used 10 episodic data, VirtualHome used 15 episodic data, Minecraft used 6 episodic data, and RLBench

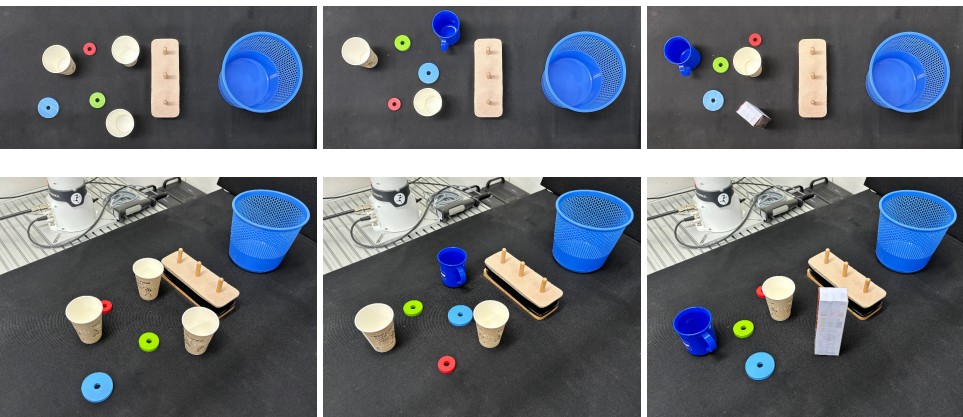

Figure 10: Environment examples set of real-world

used 5 episodic data. Episodic data comprises a series of transitions, capturing the state, the action taken, the success or failure of the action, and the resulting next state. Additionally, episodic data is expressed as text using VLM or LLM to represent visual observations or sensor data obtained from the environment. Note that this dataset is only a small portion of the overall evaluation environments used in our experiments.

For each task, we generate several rephrased instructions using ChatGPT, based on the templatized instructions provided by each benchmark, as detailed in Szot et al. (2023). These variations demonstrate how accurately NESYC can define goal conditions and execute tasks in open-domain scenarios, allowing us to evaluate its performance across a diverse range of instructions. For example, the original instructions such as "Clean some plates and put them in the fridge," "Examine the pillow with the desk lamp," and "Heat some tomatoes and put them in the fridge" are transformed into generated instructions like "Wash a few plates and refrigerate them," "Inspect the pillow using the desk lamp," and "Refrigerate the heated tomatoes," respectively.

### B.6.1 TASK-AGNOSTIC DATASET GENERATION PIPELINE

Our dataset generation follows a three-step process to create task-agnostic examples:

**Step 1: Task-agnostic Environment Information Collection.** The first step consists of two main processes. Initially, we collect environment meta-information including scene layouts, object relationships, and transition rules from the base environment. (Srivastava et al., 2022) Then, using this collected information as context, we leverage ChatGPT to generate new task-agnostic transition datasets.

**Step 2: State Set Example Generation.** This step involves creating two types of examples: positive and negative. Positive examples represent scenarios where actions are affordable and transitions are valid, documenting successful state changes with satisfied preconditions and effects. Negative examples demonstrate unaffordable actions and invalid transitions, capturing constraint violations and realistic error cases. This dual approach ensures comprehensive coverage of both successful and failed interactions.

**Step 3: Task-agnostic Episode Scenario Generation.** The final step transforms the generated examples into a task-agnostic experience dataset, where each experience is encoded in natural language form, abstracting specific objects and actions into general categories while maintaining rich textual descriptions of environmental states and interactions. We provide detailed examples of this experience dataset in Section B.6.2.

### B.6.2 EXAMPLE OF DATA STRUCTURE

```
{
    "instruction": "The robot needs to extract a fragile antique teacup
        from a high kitchen cabinet and place it safely on the dining
        table for inspection.",
    "initial_observation": "You're in a traditional kitchen with oak
        cabinets. It's mid-morning, and sunlight streams through lace
        curtains, creating intricate shadow patterns. The high cabinet's
        glass door reveals several pieces of antique china. The valuable
        teacup in question sits on the top shelf, partially hidden behind
        a larger serving plate. A stepstool is visible near the
        refrigerator, and there's a slight layer of dust on some of the
        cabinet surfaces, suggesting these items aren't frequently
        accessed.",
    "trajectory": {
        "0": {
            "observation": "The antique teacup is clearly visible through
                the glass cabinet door, sitting on the highest shelf
                approximately 7 feet from the floor. It's a delicate Bone
                China piece with hand-painted roses and gold trim,
                estimated to be from the 1920s. The cup is positioned
                behind a heavy ceramic serving plate, which partially
                blocks access. The cabinet's brass handle shows
                fingerprints from previous use, and the glass panes have
                some smudges that distort the view slightly. The overhead
                lighting reflects off the cabinet's glass, creating glare
                spots that make it difficult to see all angles of the
                teacup. A thin film of dust is visible on the shelf
                surface.",
            "action": "pick_up stepstool from floor",
            "affordance": "true",
            "next_observation": "Standing with the stepstool, you're
                facing the cabinet. The stool's rubber feet rest on the
                recently waxed hardwood floor, which shows a noticeable
                sheen. The teacup remains visible through the glass, but
                from this angle, you can now see that the serving plate in
                 front appears to be leaning slightly against the cup. The
                 cabinet's hinges look slightly loose, with one screw not
                fully tightened. The morning sunlight has shifted, causing
                 stronger glare on the cabinet glass."
        },
        "1":{
          ....

    }
}
```

### B.6.3 IMPLEMENTATION CONSIDERATIONS

Our implementation emphasizes several key aspects to ensure dataset quality:

- Consistent use of structured templates across all generated examples
- Regular validation of logical coherence and physical plausibility
- Careful balance between specific detail and general applicability
- Maintenance of realistic physical constraints and interaction patterns

### B.7 PERFORMANCE DIFFERENCES IN OPEN-DOMAIN SETTINGS

The performance differences observed in the baseline methods between the existing work and our experiments, such as in ALFWorld, are primarily attributed to changes in the environmental configuration made to accommodate the open-domain setting.

The environmental settings are designed to reflect open-domain conditions, building upon existing benchmark configurations, which are inherently complex due to their dynamic and unpredictable nature. These settings involve: (1) expanded observations, including spatial relations and object physical states, along with diverse and rephrased instructions that go beyond simple templates, requiring agents to process varied linguistic inputs, as demonstrated in Zheng et al. (2022); Chen et al. (2024); and (2) frequent environmental changes, such as unpredictable state transitions and variable object affordances, necessitating robust reasoning and adaptability, similar to Yoo et al. (2024); Cai et al. (2023).

Specifically, for observations, we configured the environment to include a wide range of object relations and states, extending beyond a simple list of object types. This setup incorporates richer details using various predicates related to object states and relations (e.g., pickupable, sliceable, can open, dirty). For instructions, instead of relying solely on about ten types of template-based instructions (e.g., "Heat some tomato and put it in the fridge.") from the original benchmarks, we utilized a diverse set of paraphrased instructions (e.g., "Refrigerate the heated tomato."). These extensions were designed to effectively capture and utilize the changes in environmental dynamics as inputs for the agent.

In terms of environment dynamics, we categorized the settings into three levels based on their dynamics, each affecting state changes:

For **Static** settings, the environment remains consistent across episodes, with stable object states, action preconditions and effects, and goal conditions.

For **Low Dynamic** settings, objects may change locations or conditions within episodes, but these changes require only minor adjustments to the agent's existing transition rules.

For **High Dynamic** settings, object states and attributes can change unpredictably within episodes, affecting goal conditions and action preconditions, significantly increasing complexity. The agent must continuously re-evaluate its plans and refine its knowledge to effectively handle frequent state shifts and varying affordances.

For this reason, baselines such as ReAct (Yao et al., 2023) and Reflexion (Shinn et al., 2024) show **performance differences** even in the static setting. Additionally, to ensure fair and meaningful comparisons under these open-domain conditions, we made the following adjustments for the baselines, as demonstrated in Xie et al. (2024).

For ReAct and Reflexion, the original papers utilize **task-specific demonstrations** as few-shot input prompts. In our setting, however, we provide task-agnostic demonstrations as input prompts without specifying task information, which increases the complexity of grounding within the given domain.

To quantitatively validate this performance difference, we conduct ablation studies with ReAct focusing on two key factors: expanded observations and task-specific few-shot prompts, which can affect model performance. We randomly selected 50 tasks and used GPT-4o-mini as the LLM for the experiments.

The results indicate that when we adjust our Static settings to replicate the original experimental settings used by ReAct—by reducing the complexity of observations and providing task-specific prompts—the performance aligns more closely with the results reported in the original paper. This demonstrates that the observed performance differences stem from the additional challenges introduced by our open-domain settings. Additionally, the 71% SR in Yao et al. (2023) corresponds to the average performance on the best-performing task category. However, when considering the overall average performance across all task categories, the reported SR in the original paper is 57%. In our experimental results in 13, when comparing the average performance across all tasks, ReAct achieved approximately 59.2% in the reproduced original setting, which aligns closely with the score originally reported in Yao et al. (2023).

## C  BASELINES

We implement several baselines for comparison. These baselines represent a diverse range of LLM-based approaches for task planning and execution in embodied environments, including methods that utilize dynamic replanning, self-reflection, grounding agents, and declarative programming. Each

Table 13: Performance comparison of ReAct on variants of ALFWorld environmental settings.

| Configuration | Static | |
| --- | --- | --- |
| | **SR** | **GC** |
| Static setting (w/o. task-specific prompt & w. expended observations) | 14.3 | 33.7 |
| Original setting (w/ task-specific prompt & w/o expended observations) | 59.2 | 64.8 |

baseline is adapted, where necessary, to operate in an open-domain setting, ensuring fair comparison with our proposed method.

## C.1 LLM-BASED PLANNING APPROACH

The hyperparameter settings for the LLM-based planning approach are summarized in Table 14. For a fair comparison, incontext samples were provided through retrieval of the top k samples. For Alfworld and Minecraft k=3, while for VirtualHome k=5 and RLBench k=10.

- **LLM-planner** (Song et al., 2023) utilizes an LLM for embodied task planning with dynamic re-planning. While originally designed to use task-specific expert knowledge, our implementation adapts it to an open-domain setting by providing task-agnostic experiences retrieved from a general dataset, aligning with our proposed method for fair comparison.

Table 14: Hyperparamer settings for LLM

| Hyperparameters | Value |
| --- | --- |
| **LLM configuration** | |
| Model | gpt-4o-2024-08-06 |
| **Text generation configuration** | |
| Temperature | 0.0 |
| Top $k$ | 1 |
| Top $p$ | 1.0 |
| Maximum new tokens | 256 |

## C.2 LLM-BASED MULTI-AGENT FRAMEWORK

The hyperparameter settings for LLM-based multi-agent frameworks are summarized in Table 14. Other settings are similar to those of the LLM-based planning approach. However, in the case of AutoGen, additional expert knowledge was provided through the grounding agent. For the remaining comparison groups in this approach, ReAct and Reflexion, this additional expert knowledge was not provided.

- **ReAct** (Yao et al., 2023), an LLM-based agent repeatedly performs reasoning and decision-making to solve a given task in the environment. Upon receiving observations, the LLM alternates between generating thoughts (reasoning) and acts (actions), autonomously making decisions to complete the task. To ensure a fair comparison, we provided task-agnostic samples for retrieval, similar to our approach with LLM-planner (Song et al., 2023), rather than using task-specific knowledge.

- **Reflexion** (Shinn et al., 2024), an LLM-based agent that, like ReAct, alternates between reasoning and acting, but uniquely incorporates a self-reflection mechanism that generates feedback by analyzing its long-term and short-term memory, using this introspective insight to iteratively refine its decision-making process.

- **AutoGen** (Wu et al., 2023a), an LLM-based system that, similar to ReAct, employs reasoning and acting cycles, but distinctively features a separate grounding agent that leverages both expert domain knowledge and implicit linguistic understanding to anchor the system's operations in the environment.

### C.3 NEURO-SYMBOLIC APPROACH

The hyperparameter settings for neuro-symbolic approaches are summarized in Table 14. Other settings are also similar to those using the LLM-based planning approach. CLMASP used an ASP solver, and in D.2, it used the same settings as ours.

- **ProgPrompt** (Singh et al., 2023) utilizes a predefined code-based system for planning, incorporating human-crafted programmatic assertion syntax to verify skill execution pre-conditions and respond to failures with predefined recovery rules; it requires expert knowledge in imperative program formats, which adds to its specificity but potentially limits its accessibility. However, it lacks a dynamic replanning mechanism, limiting its adaptability to unforeseen scenarios.

- **CLMASP** (Lin et al., 2024) is a neuro-symbolic approach that integrates LLMs with ASP, utilizing ASP's non-monotonic logic programming to represent and reason based on the robot's action knowledge; it employs declarative programming, making it the most similar approach to our research. This combination of neural networks (LLMs) and symbolic reasoning (ASP) allows CLMASP to leverage the strengths of both paradigms. However, CLMASP lacks mechanisms for dynamically generating and improving its programs, which limits its adaptability and self-improvement capabilities.

## D NESYC

### D.1 HI SCORING

$$\text{HI}(H; E, BK) = \alpha \cdot f_{\text{TPR}}(H, E^+, BK) - (1 - \alpha) \cdot f_{\text{FPR}}(H, E^-, BK)$$

where:

$$f_{\text{TPR}}(H, E^+, BK) = \lambda \cdot \frac{TP_{\mathcal{T}}}{|E_{\mathcal{T}}^+|} + (1 - \lambda) \cdot \frac{TP_{\mathcal{M}}}{|E_{\mathcal{M}}^+|} \tag{9}$$

$$f_{\text{FPR}}(H, E^-, BK) = \lambda \cdot \frac{FP_{\mathcal{T}}}{|E_{\mathcal{T}}^-|} + (1 - \lambda) \cdot \frac{FP_{\mathcal{M}}}{|E_{\mathcal{M}}^-|}$$

Eq.(9) is a rigorous and specific version of Eq.(7). Here, $\lambda \in [0, 1]$ is a hyperparameter that balances the importance between the existing experience set $\mathcal{T}$ and working memory $\mathcal{M}$. A higher $\lambda$ value gives more weight to the performance on $\mathcal{T}$, while a lower value emphasizes $\mathcal{M}$. Additionally, $\alpha \in [0, 1]$ is a parameter that adjusts the relative importance of TPR (True Positive Rate) and FPR (False Positive Rate) in the hypothesis evaluation. A higher $\alpha$ value places more emphasis on TPR, while a lower value gives more weight to minimizing FPR. For each experience set $\mathcal{X} \in \{\mathcal{T}, \mathcal{M}\}$, we define:

$$TP_{\mathcal{X}} = |\{e \in E_{\mathcal{X}}^+ : e \vDash BK \cup H\}|$$

$$FP_{\mathcal{X}} = |\{e \in E_{\mathcal{X}}^- : e \vDash BK \cup H\}|$$

$$TN_{\mathcal{X}} = |\{e \in E_{\mathcal{X}}^- : e \nvDash BK \cup H\}| \tag{10}$$

$$FN_{\mathcal{X}} = |\{e \in E_{\mathcal{X}}^+ : e \nvDash BK \cup H\}|$$

where $TP_{\mathcal{X}}$ represents the number of true positives (i.e., positive examples in $E_{\mathcal{X}}^+$ that are models of $BK \cup H$), $FP_{\mathcal{X}}$ is the number of false positives, $TN_{\mathcal{X}}$ is the number of true negatives, and $FN_{\mathcal{X}}$ is the number of false negatives. The symbol $\vDash$ denotes the satisfaction relation, indicating that an example $e$ is a model of $BK \cup H$.

This formulation allows for a comprehensive evaluation of the hypothesis $H$, considering its performance on both the existing experience set and the current environment experience set while providing flexibility in adjusting their relative importance through the $\lambda$ parameter.

## D.2 Hyperparamter settings

The configuration of the LLM follows the setup described in Table 14. For the heuristic solver in Answer Set Programming (ASP), we used clingo version 5.7.1. The clingo solver was run with specific control parameters. The parameter '–opt-mode=optN' was used to specify that the optimization mode was set to find all optimal models. The option -t '1' controlled the number of threads used during execution, which was set to 1 in this case. Lastly, the '–seed' parameter was employed to control randomness during solving, and seeds from 1 to 3 were used to ensure reproducibility across different runs. For different LLMs, we include Llama-3 (*meta-llama-3.0-8b*, *meta-llama-3.0-70b*), GPT-4 (*gpt-4o-mini, gpt-4o*), Claude (*claude-3.5-sonnet, claude-3-opus*).

## D.3 Component Details

### D.3.1 Examples of ILP

Consider the simple embodied agent scenario, where $B$:

$$B = \begin{cases} small(\text{apple}).\ small(\text{cup}).\ small(\text{glass\_vase}). \\ light(\text{apple}).\ light(\text{cup}).\ heavy(\text{table}).\ heavy(\text{shelf}). \\ fragile(\text{glass\_vase}).\ not\_fragile(x) \text{ :- } small(x), light(x). \end{cases}$$

and the examples $E = \{E^+, E^-\}$:

$$E^+ = \begin{cases} e_1^+ = pickup(\text{apple, table}). \\ e_2^+ = pickup(\text{cup, shelf}). \end{cases} \quad E^- = \begin{cases} e_1^- = pickup(\text{table, table}). \\ e_1^- = pickup(\text{glass\_vase, shelf}). \end{cases}$$

Also assume the hypothesis space $\mathcal{H}$:

$$\mathcal{H} = \begin{cases} h1 = pickup(x, y) \text{ :- } small(x), fragile(x), heavy(y). \\ h2 = pickup(x, y) \text{ :- } small(x), light(x), heavy(y). \\ h3 = pickup(x, y) \text{ :- } small(x), light(x), non\_fragile(x), heavy(y). \end{cases}$$

we need to find a hypothesis $H$ such that $e_1^+$ and $e_2^+$ are models of $H \in B$, while $e_1^-$ and $e_2^-$ are not. In this case, $e_1^-$ is not a model of $h_2$ because there exists a substitution $\theta = \{x/\text{table}, y/\text{table}\}$ such that the body does not hold, and thus the head is not valid. For the same reason, none of the examples is a model of $h_1$. This indicates that the hypothesis $H = \{h_2, h_2\}$ consists of both $h_2, h_3$.

### D.3.2 Implementation Guidelines for ASP Action Rules

ASP rules for actions are primarily divided into two categories: Precondition rules that specify when actions are allowed, and Effect rules that define how actions change the world state.

**Action Precondition Format:**

```
:- action(ActionName(Args), Time), Cond1(Args, Time),
   not Cond2(Args, Time).
```

- Uses integrity constraints (:−) to specify invalid action conditions
- When the body of constraint is satisfied, the action is forbidden
- `not` operator indicates the precondition must be satisfied
- Multiple constraints can be defined for a single action

**Action Effect Format:**

```
State(Args, Time) :- action(ActionName(Args), Time).
```

- Defines state changes resulting from actions
- State can be any predicate (e.g., `holding/2`, `at/3`)
- Direct causal relationship between action and its effects
- Can include additional conditions with multiple body literals

**Key Components:**

- `action/2`: Predicate representing actions with arguments and time
- State predicates: Represent world states (e.g., `holding/2`, `at/3`)
- Condition predicates: State properties that must hold
- Time variable: Usually denoted as `T` for temporal reasoning
- Variables: Typically capitalized (e.g., `O` for object, `L` for location)
- `_`: Anonymous variable, used when specific value is not relevant
  - Each `_` is treated as a distinct, unique variable
  - Used when we don't need to reference the value later
  - Multiple `_` in the same rule are independent variables

**Example of Precondition Rules:**

```
1. Object Location Check:
 :- action(pick_up(O, L), T), not at(O, L, T).
 % Ensures object O is at location L before pickup

2. Holding State Check:
 :- action(pick_up(O, _), T), holding(O2, T).
 % Prevents pickup when already holding something
```

### D.3.3 EXAMPLES OF ASP

Given a scenario where an agent receives environmental observation information and an instruction to "pick up a fruit", here is an example of an ASP program for this simple task.

```
1.   #program base.
2.   location(table). object(plate). object(fork). object(orange). holding
     (none).
3.   location(X) :- object(X).
4.   holds(F,0) :- init(F).
5.   init(on(fork,table)). init(on(orange,plate)). init(on(plate,table)).
6.   goal(holding(orange)).
7.   #program step(t).
8.   0 { occurs(pickup(X,Y),t) : object(X), location(Y), X != Y } 1.
9.   :- occurs(pickup(X,Y),t), holds(on(A,X),t-1).
10. holds(holding(X),t) :- occurs(pickup(X,Y),t).
11. holds(on(X,Z),t) :- holds(on(X,Z),t-1), not holding(X).
12. #program check(t).
13. :- query(t), goal(F), not holds(F,t).
```

The planning problem is defined in lines 1 to 6. This code sets up a scenario where three objects—plate, fork, and orange—are placed either on a table or on each other. The goal is for the agent to be holding the orange. The initial positions of the objects are specified using the init predicates. The action logic is defined in lines 7 to 11, which includes the preconditions and effects of the pickup action. The occurs predicate is used to specify when actions happen, while the preconditions (line 9) restrict when the pickup action can be executed. The effects (line 10) update the state of the environment based on the actions taken. In lines 12 and 13, the goal condition is checked to ensure that the agent reaches the desired goal state.

For ALFWorld agents, we define symbolic actions by combining behaviors (e.g., "pick up") with target objects (e.g., "apple") and receptacle objects (e.g., "sink"). A symbolic action can be represented as *pickup(X, Y)*, with preconditions such as *at(apple 2, diningtable 1)*, *pickupable(apple 2)*, *Object(apple 2)*, and *receptacle(diningtable 1)*. The effects of this action would be *hold(agent, apple 2)*, and *at(apple 2, agent)*. For RLBench agents, we define skills based on the combination of primitive actions (e.g., "move forward", "rotate left", "gripper open"), target objects (e.g., "window", "umbrella", "red block"), and action-specific parameters that quantify the magnitude of the action (e.g., "Degree: 55" for rotation, "Distance: 3" for movement, or "Force: 2" for gripping).

Consider the symbolic action *pickup(bread, StoveBurner)*, which is intended to fulfill the user instruction, "heat the bread and place it on the side table". If the object state of the bread has already changed to "heated" before the agent performs the *pickup* action, the agent should re-plan to go directly to the side table, bypassing the use of the microwave. Continuing with actions intended to heat the bread in this situation could lead to inefficiencies or an incorrect plan due to unintended ramifications.

### D.3.4 EXAMPLES OF PROMPT

---

**Prompt1 of Hypothesis Generator**

**Role:** You are an inductive logic programming agent using Answer Set Programming.
**Task:** Proceed with the inductive logical programming process. Generate the Background knowledge based on learning from interpretation.
**Learning from Interpretations (LFI) Concept:**

1. Background Knowledge (B):
   - father(henry,bill). father(alan,betsy). father(alan,benny).
   - mother(beth,bill). mother(ann,betsy). mother(alice,benny).

2. Positive Examples (E+):
   - e1 = {carrier(alan), carrier(ann), carrier(betsy)}
   - e2 = {carrier(benny), carrier(alan), carrier(alice)}

3. Negative Example (E-):
   - e3 = {carrier(henry), carrier(beth)}

4. Hypothesis Space (H):
   - h1 = carrier(X):- mother(Y,X),carrier(Y),father(Z,X),carrier(Z).
   - h2 = carrier(X):- mother(Y,X),father(Z,X).

LFI Problem Definition:
Find a hypothesis H such that e1 and e2 are models of H ∪ B and e3 is not.
...

**Predicates:** {predicates}
**Instructions:**

1. Read the provided LFI descriptions.

2. Use only the predicates and parameter descriptions provided.

3. Generate the Background Knowledge that can use hypothesis about Positive/Negative Examples.

**Positive/Negative Examples:** {positive_negative_examples}

Make Background knowledge.

---

**Prompt2 of Hypothesis Generator**

**Role:** You are an inductive logic programming agent using Answer Set Programming.
**Task:** Proceed with the inductive logical programming process. Find a hypothesis based on learning from interpretation.
**ASP Rule Formats:**

- Constraint Rules (also known as Integrity Constraints): . . .

**Learning from Interpretations (LFI) Concept:**

1. Background Knowledge (B):
   - father(henry,bill). father(alan,betsy). father(alan,benny).
   - mother(beth,bill). mother(ann,betsy). mother(alice,benny).

2. Positive Examples (E+):
   - e1 = {carrier(alan), carrier(ann), carrier(betsy)}
   - e2 = {carrier(benny), carrier(alan), carrier(alice)}

3. Negative Example (E-):
   - e3 = {carrier(henry), carrier(beth)}

4. Hypothesis Space (H):
   - h1 = carrier(X):- mother(Y,X),carrier(Y),father(Z,X),carrier(Z).
   - h2 = carrier(X):- mother(Y,X),father(Z,X).

LFI Problem Definition:
Find a hypothesis H such that e1 and e2 are models of H ∪ B and e3 is not.
...

**Predicates:** {predicates}
**Feedback:** {interpreter_feedback}
**Instructions:**

1. Carefully read the provided ASP and LFI descriptions. Create rules in the Constraint Rules format.

2. Use only the predicates and parameter descriptions provided.

3. Apply θ-subsumption for Rule ordering.

4. Think generalized rules to include positive examples and Background Knowledge for target predicates while excluding negative ones, ensuring that feedback is incorporated into the generalization process.

5. Generate the most general rules.

**Positive/Negative Examples:** {positive_negative_examples}
**Background Knowledge:** {background_knowledge}
**Target Action Predicate:** {target_action_predicate}

Make generalized rules.

---

**ASP program example for the Alfworld from Rule Generalization**

```
...
% Constraint: An object cannot be picked up if it is not at the
    robot's location.
:- action(pick_up(O, L), T), not at(O, L, T).

% Constraint: An object cannot be picked up if the robot is not at
    the object's location.
:- action(pick_up(O, L), T), not robot_at(L, T).

% Constraint: An object cannot be picked up if it is not openable
    and opened.
:- action(pick_up(O, L), T), not openable(L), is_open(L, T).

% Constraint: An object cannot be picked up if it is not being held
    .
:- action(pick_up(O, L), T), not holding(O, T).
...
```

## Positive/Negative Examples

**Positive Examples:**

- {is_opened(bookcase, 1), robot_at(bookcase_location, 1), at(book, book-case_location, 1), action(pick_up(book, bookcase_location), 2)}

⋮

- {is_opened(toolbox, 3), at(tool, toolbox, 3), robot_at(toolbox_location, 3), action(pick_up(tool, toolbox), 4)}

**Negative Examples:**

- {not is_opened(wardrobe, 5), holding(shirt, 5), robot_at(wardrobe, 5), action(open(wardrobe), 5), action(pick_up(shirt, wardrobe), 6)}

⋮

- {is_opened(refrigerator, 7), holding(food, 7), robot_at(kitchen, 7), action(open(refrigerator), 7), action(pick_up(food, refrigerator), 8)}

## Predicates Examples

⋮

**Basic Predicates:**

- location(L).
- object(O).
- goal(T).
- step(T).

⋮

**State Predicates:**

- at(O, L, T).
- robot_at(L, T).
- holding(O, T).

⋮

- is_opened(O, T).

**Location Properties:**

- is_heater(L).
- is_cooler(L).
- is_cleaner(L).

**Actions:**

- action(go_to(L)., T).
- action(pick_up(O, L)., T).

⋮

- action(close(O)., T)..

**Object Properties:**

- openable(L)..
- cleanable(O).
- coolable(O).
- heatable(O).
- sliceable(O).

**Specific Objects:**

- is_peppershaker(O).
- is_toiletpaper(O).

⋮

- is_pillow(O).

**Furniture and Appliances:**

- is_bed(O).
- is_countertop(O).

⋮

- is_dresser(O).

**Kitchen Items:**

- is_pot(O).
- is_winebottle(O).

⋮

- is_spatula(O).

⋮

---

**Background Knowledge Examples**

**Locations:**

- location(bookcase_location).
- location(dishwasher).

⋮

- location(kitchen).

**Objects:**

- object(book).

⋮

- object(food).

**Openable:**

- openable(bookcase).

⋮

- openable(refrigerator).

**Cleanable:**

- cleanable(plate).

---

# E   ADDITIONAL EXPERIMENTS

## E.1   SIMULATED ENVIRONMENT ROLLOUT TRAJECTORIES

Figure 11 illustrates the demonstration trajectories from the baselines and NESYC for the window manipulation task. LLM-planner in Figure 11(a) and ReAct in Figure 11(b) failed because they did not consider the preceding action of rotating the gripper forward to grasp the window handle and attempted to grab the handle directly. Reflexion in Figure 11(c) successfully grabbed the window handle but failed to account for the preceding action of rotating the handle before pushing the window. AutoGen in Figure 11(d), influenced by the environmental description "consider preceding actions for the main action" in the grounding prompt, successfully completed the task. However, there is a tendency to perform inefficient planning due to the inability to consider all preceding actions. CLMASP in Figure 11(e) and NESYC in Figure 11(f) successfully achieved efficient planning by considering all preceding actions through symbolic execution.

## E.2   ADDITIONAL EVALUATION ON PREDICATE CATEGORIES.

Figure 12 presents graphs that evaluate not only the F1 score, as shown in Figure 4, but also the Recall and Precision scores. The left side of the graph displays Recall scores, while the right side shows Precision scores. The color coding for different Large Language Models (LLMs) corresponds to that used in Figure 3. For the Status category, the Recall scores of each LLM demonstrated similar performance improvement trends to their F1 scores. In contrast, for the Spatiality category, the Precision scores showed trends similar to the F1 scores in terms of performance improvement. This additional analysis provides a more comprehensive view of the models' performance across different evaluation metrics.

## E.3   DETAILED SCORE OF GENERALIZATION LOOP EVALUATION ON EXPERIENCE SET.

Table 15 shows the raw performance data (%) of various models evaluated on their ability to generalize from experiences. This table provides the numerical values represented in Figure 3. Precision, Recall, and F1 score measure the models' prediction accuracy, while Specificity and Accuracy assess their overall correctness. GPT-4 performs best overall, with the highest Precision, Recall, and F1 score, though its Specificity is moderate. Llama-3.0-8B shows the weakest performance, with low scores across the board.

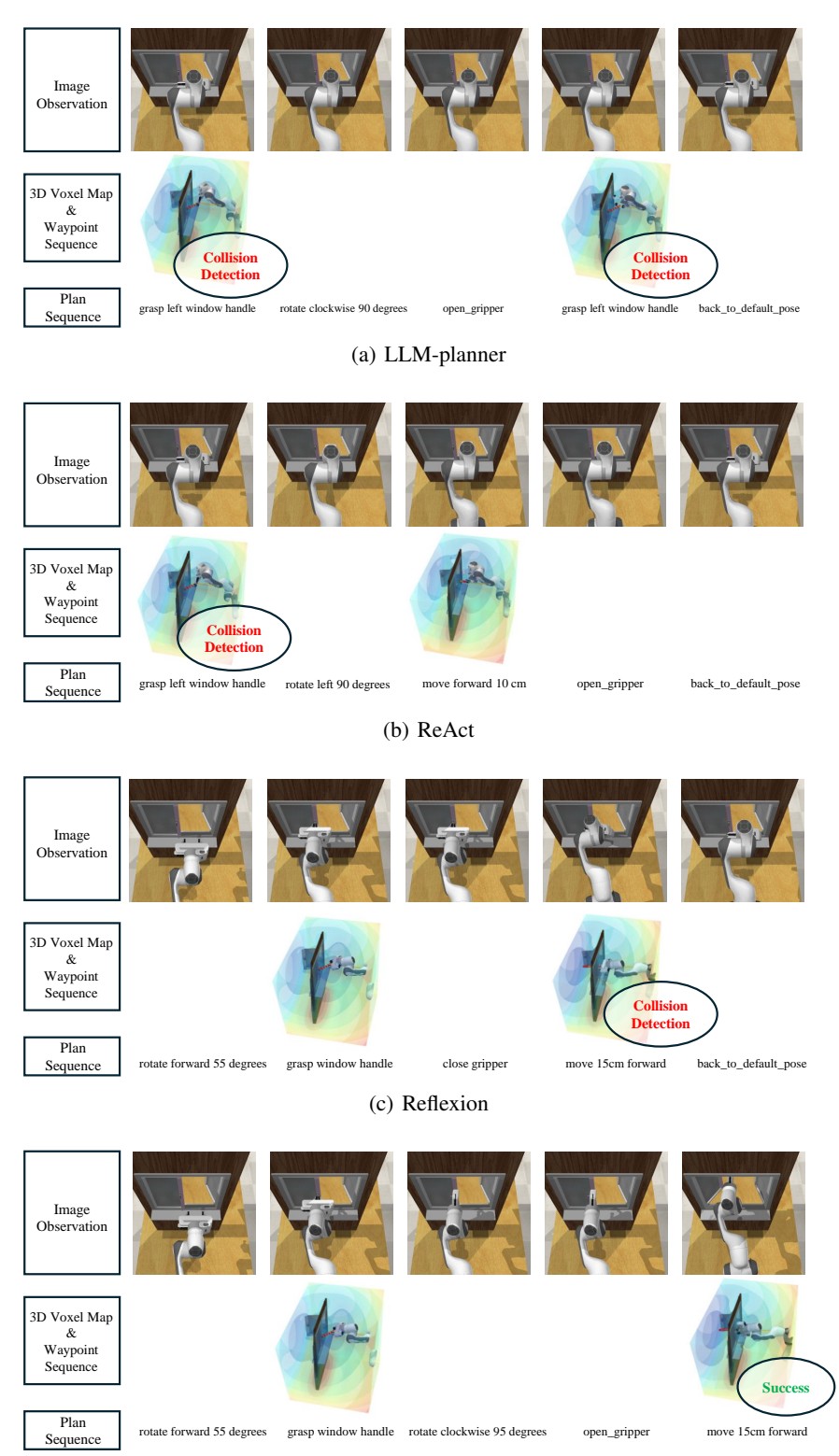

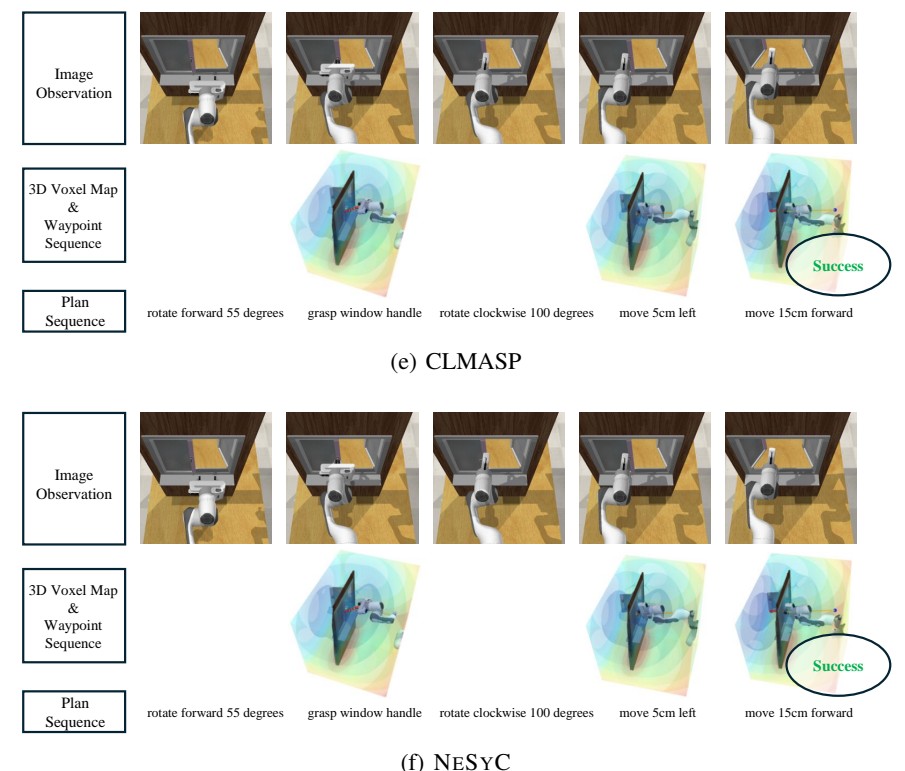

Figure 11: Visualization of trajectories in RLBench.

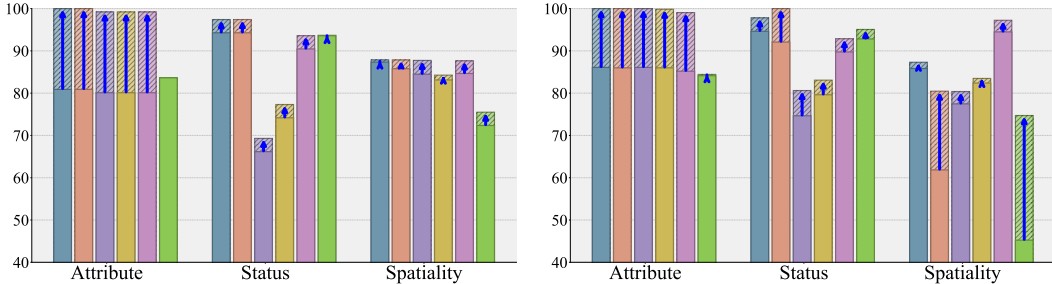

Figure 12: Recall score and Precision score evaluation on predicate categories.

Table 15: Detailed score of Generalization loop evaluation on experience set (B: Before / A: After).

| Model | Precision | | Recall | | F1 | | Specificity | | Accuracy | | HI Score | |
|---|---|---|---|---|---|---|---|---|---|---|---|---|
| | B | A | B | A | B | A | B | A | B | A | B | A |
| GPT-4 | 66.5 | 65.7 | 91.7 | 95.5 | 77.1 | 77.8 | 53.7 | 50.0 | 72.7 | 72.7 | 0.78 | 0.80 |
| GPT-4o | 62.7 | 63.6 | 91.7 | 95.5 | 74.5 | 76.3 | 45.5 | 45.5 | 68.6 | 70.5 | 0.75 | 0.79 |
| GPT-4o-mini | 61.9 | 62.1 | 75.2 | 81.8 | 67.9 | 70.6 | 53.7 | 54.5 | 64.5 | 65.9 | 0.63 | 0.70 |
| Claud-3-Opus | 60.7 | 63.6 | 58.7 | 63.6 | 59.7 | 63.6 | 62.0 | 63.6 | 60.3 | 63.6 | 0.52 | 0.57 |
| Claud-3.5-Sonnet | 59.1 | 61.5 | 66.9 | 72.7 | 62.8 | 66.7 | 53.7 | 54.5 | 60.3 | 63.6 | 0.56 | 0.62 |
| Llama-3.0-70B | 64.3 | 64.6 | 83.5 | 90.9 | 72.7 | 75.5 | 53.7 | 50.0 | 68.6 | 70.5 | 0.71 | 0.76 |
| Llama-3.0-8B | 44.5 | 46.2 | 50.4 | 54.5 | 47.3 | 50.0 | 37.2 | 36.4 | 43.8 | 45.5 | 0.34 | 0.38 |

### E.4 DETAILED EVALUATION ON PREDICATE CATEGORIES.

Table 16 presents comprehensive scores (%) for different models across various predicate categories. This data is visually represented in Figure 4 and Figure 12. Analysis of predicate categories reveals diverse model performance across semantic domains. GPT-4 and GPT-4o consistently excel in after score, achieving near-perfect scores in Attributes and 97.4-100% accuracy in Status categories. Spatiality shows the highest inter-model variability, with Claude-3-Opus leading in precision. Most models demonstrate significant improvement in the 'After' condition, particularly in Attributes. Llama-3.0-70B and Claude-3.5-Sonnet show moderate performance, while GPT-4o-mini, despite lower overall scores, exhibits substantial improvement.

Table 16: Detailed evaluation on predicate categories.

(a) Attribute Scores

| Model | Precision | | Recall | | F1 | |
|---|---|---|---|---|---|---|
| | Before | After | Before | After | Before | After |
| GPT-4 | 86.1 | 100.0 | 80.9 | 100.0 | 83.0 | 100.0 |
| GPT-4o | 86.0 | 100.0 | 80.9 | 100.0 | 83.0 | 100.0 |
| Llama-3.0-70B | 86.1 | 100.0 | 80.1 | 99.2 | 82.6 | 99.5 |
| Claud-3.5-Sonnet | 86.0 | 99.9 | 80.1 | 99.2 | 82.5 | 99.4 |
| Claud-3-Opus | 85.2 | 99.1 | 80.1 | 99.2 | 82.2 | 99.1 |
| GPT-4o-mini | 84.1 | 84.4 | 83.6 | 83.6 | 83.8 | 84.0 |

(b) Status Scores

| Model | Precision | | Recall | | F1 | |
|---|---|---|---|---|---|---|
| | Before | After | Before | After | Before | After |
| GPT-4 | 94.6 | 97.8 | 94.3 | 97.4 | 94.3 | 97.5 |
| GPT-4o | 92.1 | 100.0 | 94.3 | 97.4 | 93.0 | 98.6 |
| Llama-3.0-70B | 74.7 | 80.6 | 66.2 | 69.3 | 69.7 | 74.1 |
| Claud-3.5-Sonnet | 79.6 | 83.1 | 74.2 | 77.3 | 76.7 | 80.0 |
| Claud-3-Opus | 89.8 | 92.9 | 90.5 | 93.6 | 90.0 | 93.1 |
| GPT-4o-mini | 92.8 | 95.1 | 93.6 | 93.7 | 93.1 | 94.3 |

(c) Spatiality Scores

| Model | Precision | | Recall | | F1 | |
|---|---|---|---|---|---|---|
| | Before | After | Before | After | Before | After |
| GPT-4 | 85.8 | 87.3 | 87.3 | 87.8 | 86.4 | 87.6 |
| GPT-4o | 61.8 | 80.5 | 85.8 | 87.8 | 71.7 | 83.8 |
| Llama-3.0-70B | 77.5 | 80.4 | 84.5 | 87.8 | 80.6 | 83.7 |
| Claud-3.5-Sonnet | 82.4 | 83.5 | 83.1 | 84.3 | 82.3 | 83.4 |
| Claud-3-Opus | 94.5 | 97.2 | 84.7 | 87.7 | 88.5 | 91.4 |
| GPT-4o-mini | 45.2 | 74.7 | 72.4 | 75.5 | 55.5 | 75.0 |

### E.5 DETAILED STEP OF REAL-WORLD EXPERIENCE

In our real-world experiment, we implemented the instruction "Clean up the desk and complete the Tower of Hanoi." The experimental steps are illustrated in Figure 13. The objects involved in this task consisted of three disposable paper cups, three Tower of Hanoi disks of varying sizes, three Tower of Hanoi poles, and a trash bin. subtask 1 comprised 13 steps, while subtask 2 consisted of 29 steps. The entire task was completed in a total of 32 steps.

During the experiment, we encountered a challenge at step 8 when attempting to grasp the blue block of the Tower of Hanoi. We found that the existing rules were insufficient to resolve this issue. To address this problem, we utilized the NESYC framework to add additional rules specifically for grasping the Tower of Hanoi blocks. The details of these additional rules and their implementation can be found in Figure 5.

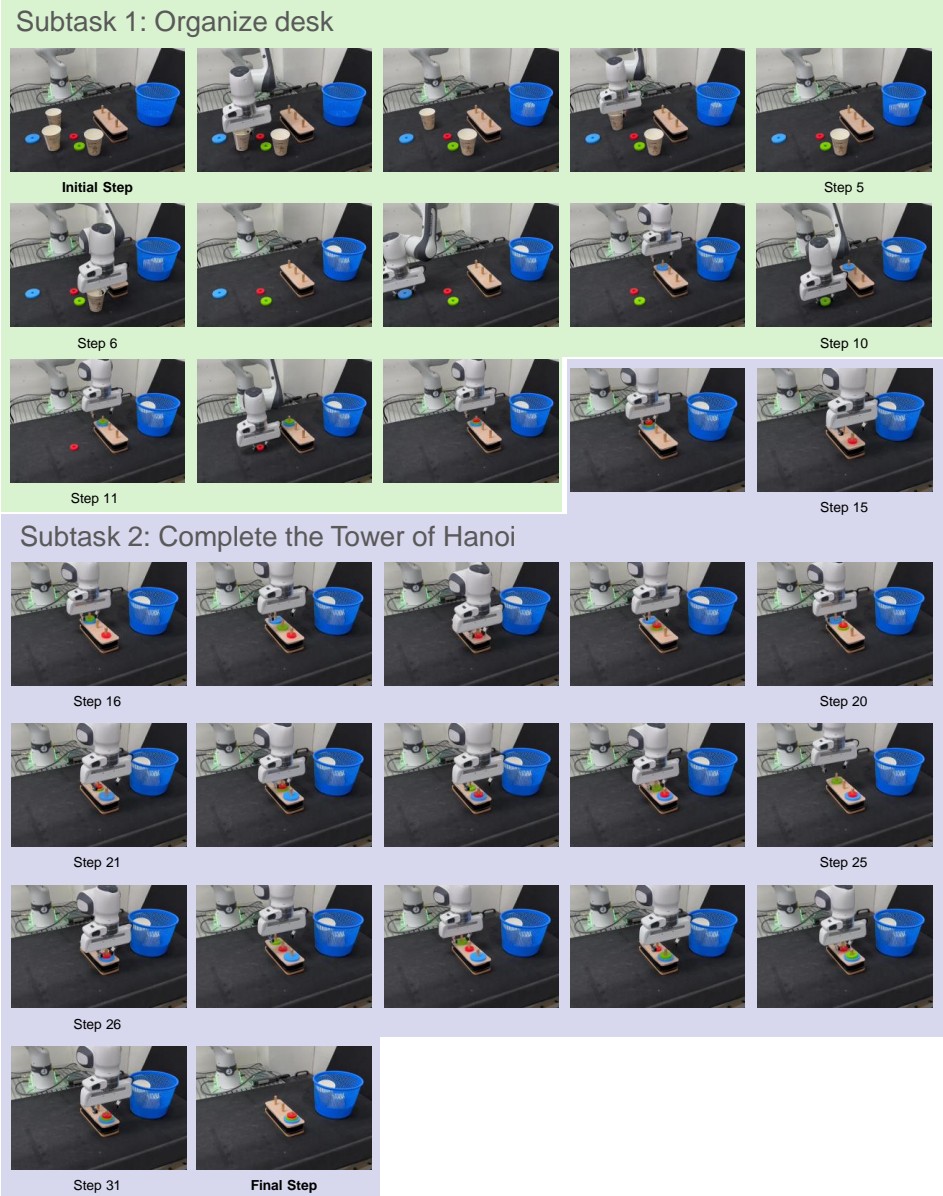

Figure 13: Example of Real-world task

### E.6 ANALYSIS OF CONTINUAL LEARNING AND KNOWLEDGE GENERALIZATION

For further clarification, we conducted additional experiments to demonstrate the generalizability of our framework's actionable knowledge in a continual learning scenario. These experiments evaluate how effectively the actionable knowledge acquired through continual learning has been refined to generalize to future unseen tasks, as demonstrated in Kim et al. (2024); Lee et al. (2025). The results highlight our framework's ability to adapt and refine its knowledge throughout the continual learning

phases, showing performance that supports zero-shot generalization in dynamic and open-domain environments.

Let us first provide a summary of the continual learning scenario and results, along with the Table 17 and Table 18 for reference. The scenario consisted of 8 phases, requiring approximately 50 and 40 steps (based on oracle performance) for task completion, respectively. NESYC continually generalized actionable knowledge by leveraging accumulated experiences.

At each phase, we evaluated the framework's performance on both current and previous tasks to assess how well it retained existing knowledge while adapting to new environments. Throughout these scenarios, we measured the rate of change between successive phases ($\Delta$), based on the number of rules, and assessed the similarity of the refined rules to predefined expert-level rules using the F1 score.

Table 17: Performance of continual learning scenario in ALFWorld

|  | Metric | Phase 1 | Phase 2 | Phase 3 | Phase 4 | Phase 5 | Phase 6 | Phase 7 | Phase 8 |
|---|---|---|---|---|---|---|---|---|---|
| NESYC | SR | 1.0 | 1.0 | 1.0 | 1.0 | 1.0 | 1.0 | 1.0 | 1.0 |
|  | GC | 1.0 | 1.0 | 1.0 | 1.0 | 1.0 | 1.0 | 1.0 | 1.0 |
|  | $\Delta$ (%) | 150 | 0 | 200 | 67 | 40 | 0 | 0 | 0 |
|  | F1 | 28.6 | 28.6 | 50.0 | 73.7 | 90.9 | 90.9 | 90.9 | 90.9 |

Table 18: Performance of continual learning scenario in RLBench

|  | Metric | Phase 1 | Phase 2 | Phase 3 | Phase 4 | Phase 5 | Phase 6 | Phase 7 | Phase 8 |
|---|---|---|---|---|---|---|---|---|---|
| NESYC | SR | 1.0 | 1.0 | 1.0 | 1.0 | 1.0 | 1.0 | 1.0 | 1.0 |
|  | GC | 1.0 | 1.0 | 1.0 | 1.0 | 1.0 | 1.0 | 1.0 | 1.0 |
|  | $\Delta$ (%) | 100 | 250 | 0 | 0 | 100 | 0 | 28 | 0 |
|  | F1 | 30.3 | 63.4 | 63.4 | 63.4 | 80.6 | 80.6 | 96.3 | 96.3 |

Additionally, as shown in the results in the Table 19 and Table 20, we evaluated how the improved actionable knowledge during the continual learning scenarios in each benchmark was refined to generalize effectively to future unseen tasks. For this evaluation, we randomly selected 50 unseen tasks in a static setting and tested the intermediate rules ($R$) generated at specific phases. The results indicate that rules from earlier phases consistently expanded task coverage, demonstrating the framework's ability to adapt and improve over time. This refinement process effectively leverages accumulated experiences to generalize existing rules.

Table 19: Performance of actionable knowledge from each phase on unseen tasks in ALFWorld

| Method | $R$ from Phase 1 | | Phase 3 | | Phase 4 | | Phase 5 | | Phase 8 | |
|---|---|---|---|---|---|---|---|---|---|---|
|  | SR | GC | SR | GC | SR | GC | SR | GC | SR | GC |
| NESYC | 38.3 | 50.4 | 53.2 | 67.9 | 74.5 | 77.3 | 87.2 | 90.8 | 91.5 | 94.3 |

Table 20: Performance of actionable knowledge from each phase on unseen tasks in RLBench

| Method | $R$ from Phase 1 | | Phase 2 | | Phase 5 | | Phase 7 | | Phase 8 | |
|---|---|---|---|---|---|---|---|---|---|---|
|  | SR | GC | SR | GC | SR | GC | SR | GC | SR | GC |
| NESYC | 23.8 | 24.3 | 43.9 | 45.6 | 67.9 | 71.4 | 93.3 | 95.2 | 93.9 | 96.8 |

Table 21: Comparative performance analysis of different LLM models with initial and improved results

| LLM | SR | ➜ SR | GC | ➜ GC |
|-----|------|---------|------|---------|
| Llama-3.1-8B | 44.3 | → 56.8 | 49.6 | → 60.0 |
| Llama-3-8B | 43.9 | → 40.2 | 43.9 | → 40.2 |
| GPT-4o-mini | 41.9 | → 78.7 | 44.7 | → 79.3 |
| Claude-3.0-Opus | 50.7 | → 76.7 | 53.6 | → 78.6 |
| Claude-3.5-Sonet | 51.4 | → 78.4 | 54.2 | → 80.2 |
| Llama-3-70B | 58.8 | → 85.1 | 60.4 | → 86.6 |
| GPT-4o | 64.2 | → 90.2 | 67.0 | → 90.5 |
| GPT-4 | 69.6 | → 89.2 | 73.3 | → 89.8 |

### E.7 EXTENDED LLM PERFORMANCE ANALYSIS

We conducted experiments similar to those presented in Table 3 using Llama-3.1-8B and Llama-3.2-3B, with results shown in Table 21. For Llama-3.1-8B, while it outperformed Llama-3-8B with SR improvement from 44.3% to 56.8% and GC improvement from 49.6% to 60.0%, its performance remained below practically meaningful levels compared to other models. For Llama-3.2-3B, issues with the semantic parsing module prevented it from functioning correctly, making it challenging to obtain meaningful experimental results and thus excluding it from Table 21.

Specifically, Llama-3.2-3B frequently introduced syntax errors during the semantic parsing process, such as incorrectly mapping predicates (e.g., using 'object1' instead of the appropriate 'object' type parameter), creating non-standard predicates like 'inside_room' instead of the correct 'inside' predicate, and generating non-existent predicates such as 'variable_type'. These errors disrupted subsequent processing and affected overall performance.

