# OpenReview forum: "NeSyC: A Neuro-symbolic Continual Learner For Complex Embodied Tasks In Open Domains"
_ICLR.cc/2025/Conference — ICLR 2025 Poster_

### Official Review · Reviewer_CpeJ · 2024-10-30

**Soundness:** 3
**Presentation:** 3
**Contribution:** 3
**Rating:** 8
**Confidence:** 3

**Summary:**

This work introduces a continual learning framework for dynamically restructuring neural-symbolic knowledge into rules via a Rule Reformulation process. It employs a contrastive generality improvement scheme based on Learning from Interpretations (LFI), a form of Inductive Logic Programming (ILP). The learned rules are utilized for interaction during Rule Application, supported by a memory-based monitoring mechanism within the Answer Set Programming (ASP) framework. Feedback from the Rule Application phase collects new data and periodically triggers Rule Reformulation to further refine the logical rules.

The experiments feature a diverse set of baselines, robustness evaluations, and ablation studies on key components, such as backbone VLMs and prompts. The results highlight the framework’s potential to seamlessly integrate low-level execution with logical reasoning and natural language instructions.

**Strengths:**

The work provides a solid introduction to the foundational neural-symbolic knowledge required for its framework, including Learning from Interpretations (LFI) within Inductive Logic Programming (ILP) and Answer Set Programming (ASP), as part of the background preliminaries.

The explanations effectively convey the system's core ideas, offering essential details about key functions, such as the use of LFI in Rule Reformulation and the mechanisms for memory modification.

Comprehensive information about NeSYC is provided in the Appendix, along with examples of neural-symbolic designs, making it easier for readers to understand the underlying processes.

The experiments are extensive, covering diverse environments and settings, demonstrating NeSYC’s capability and robustness.

**Weaknesses:**

Some aspects of the method remain unclear. For example, what exactly is the role of $\mathcal{I}_d$, and how does it relate to $\mathcal{G}_d$ as introduced? A reorganization of Section 2.3 may be necessary, as it forms the foundation of the problem setup.

Additionally, while the authors seem intent on making NeSYC capable of handling a broad range of situations, many of the symbolic formulations explaining the method feel too abstract. This is not a major issue, but it would be helpful if the authors provided more concrete examples of how these symbols are implemented within the main body of the method section, rather than placing all such details in the appendix.

The left portion of Figure 1 is also confusing. It is not adequately addressed in the annotations, leaving the reader unsure of its significance.

Finally, certain experimental settings could benefit from clearer descriptions. For example, in the Dataset section (Section 4.1), the format of the expert-level episodic experience data is ambiguous. Is the data presented in natural language, code, or another format? Providing compact and clear explanations of these settings would make it easier for others to understand the data processing steps necessary for reproducing a similar system.

**Questions:**

### Questions Regarding $itermax$ and Rule Learning ($R$)

- I am curious about the role of the fixed **$itermax$** in learning the rules $R$. Does setting a fixed $itermax$ always guarantee a reasonable system with appropriate rules? Could you provide results for different $itermax$ values to show their impact on performance and the resulting rules?
- Additionally, I am particularly interested in the notion of **convergence** within the NeSYC system. It might be valuable to explore how convergence is defined in your case. How sensitive is the system to environmental changes—would significant changes disrupt convergence? Also, is it possible for the rules $R$ to exhibit a **cyclical pattern** rather than converging to a stable state?
- On the topic of **continual learning**, I noticed that your system appears to stop growing its rules $R$ at some point (**$itermax$**), with subsequent testing based on a fixed set of rules. This seems different from my understanding of continual learning as a life-long process where rules evolve continuously based on new data. Could you conduct experiments where NeSYC and baseline agents keep learning and testing in various settings without a fixed $itermax$? It would be interesting to compare their performance, especially in terms of how quickly each system improves its success rate across tasks.

### Questions Regarding Experiments

- I have some concerns about the "Step" evaluation in your experiments. Although this evaluation follows prior work, I worry about potential overfitting. Achieving tasks does not always require strict adherence to ground truth; instead, achieving key milestones or sub-goals should suffice. This is particularly true for manipulation tasks—for example, a robot’s trajectory does not need to exactly match the demonstration. In Table 1, I noticed that both NeSYC and other systems show high "Step" evaluation scores, indicating close similarity to demonstrations. This raises questions about these systems' generalization capabilities in new environments. Could you clarify how NeSYC ensures generalization beyond following exact demonstrations? Are there additional experiments or insights that address this concern?
- Similarly, in the **Ablation Study** (Section 4.3), could you provide more insight into why simplified planning improves Goal Completion (GC) and "Step" evaluation?

- For the **Robustness to Experience Incompleteness** experiment in Section 4.3, could you elaborate on what you mean by *Noisy* and *Imperfect* experiences? What preprocessing steps did you apply to the datasets or system to create these noisy or imperfect conditions?

- In the **comparison of different dynamics predicates** (Section 4.3), could you clarify what your **F1 score** represents? How was the evaluation data collected to calculate the F1 score? What insights are you aiming to provide with the changes in F1 score, especially in relation to the characteristics of the predicates?

- Lastly, I would appreciate more detailed observations from your **real-world experiments**. Much of the current description is qualitative. Were these real-world scenarios tested multiple times, and did you observe recurring patterns? It would also be helpful to quantify these observations with metrics, rather than relying solely on descriptive results.

---

> ### Author Response · Authors · 2024-11-19
>
> Dear Reviewer CpeJ,
>
> We are grateful for the reviewer’s passionate and valuable feedback, as well as the opportunity to enhance the clarity of our paper. We especially appreciate the reviewer’s effort in thoroughly reviewing the paper, including the Appendix, which goes beyond the mandatory scope. Additionally, we are thankful for the reviewer’s additional insights, intuition, and thoughtful advice. We will provide detailed and comprehensive responses to the weaknesses and questions raised.
>
> ---
> > **Weakness 1**: Some aspects of the method remain unclear. For example, what exactly is the role of $\mathcal{I}_d$, and how does it relate to $\mathcal{G}_d$ as introduced? A reorganization of Section 2.3 may be necessary, as it forms the foundation of the problem setup.
>
> In Section 2.3 of our paper, we define $\mathcal{I}_d$ and $\mathcal{G}_d$ for a given domain $d$. $\mathcal{I}_d$ represents the set of possible high-level instructions, while $\mathcal{G}_d$ denotes the set of goal states within the environment. The relationship between these sets is characterized by a mapping that associates instructions with their corresponding goal states.
> While $\mathcal{I}_d$ can be mapped directly to representations within the state space, our framework does not assume this as a default. This design allows for more diverse instructions in open-domain settings.
> For example, consider the instruction, "Locate two glass bottles and place them on the shelf". This element of $\mathcal{I}_d$ maps to a goal state in $\mathcal{G}_d$, which can be formally represented using predicates:
> `at(X1, Y, T), at(X2, Y, T), is_glassbottle(X1), is_glassbottle(X2), is_shelf(Y)$.`.
> Here, the predicates define the concrete goal state by specifying the required spatial relationships (e.g., `roboa_at`, `at`) and object properties (e.g., `is_glassbottle`, `is_shelf`).
>
> We acknowledge a potential source of confusion in our notation. While $\mathcal{I}_d$ and $\mathcal{G}_d$ represent sets, the inputs to the policy $\pi$ in Eq. (3) of our paper should be individual element $i_d$, where $i_d \in \mathcal{I}_d$ correspond to $g_d \in \mathcal{G}_d$ [1,2,3]. We revised the notation in Eq. (3) and Section 2.3 to clarify this distinction.
>
> $[1]$ Ahn et al. Do As I Can, Not As I Say: Grounding Language in Robotic Affordances. CoRL 2022.
>
> $[2]$ Huang et al. Grounded Decoding: Guiding Text Generation with Grounded Models for Embodied Agents. NeurIPS 2023.
>
> $[3]$ Yoo et al. Exploratory Retrieval-Augmented Planning For Continual Embodied Instruction Following.  NeurIPS 2024.
>
> ---
> > **Weakness 2**: Additionally, while the authors seem intent on making NeSyC capable of handling a broad range of situations, many of the symbolic formulations explaining the method feel too abstract. This is not a major issue, but it would be helpful if the authors provided more concrete examples of how these symbols are implemented within the main body of the method section, rather than placing all such details in the appendix.
>
> To provide more intuitive guidance on how symbols are implemented, we incorporated specific examples directly into Figure 2 of our paper, focusing on the output of general knowledge $R$.
> These examples will demonstrate key symbolic formulations in NeSyC, such as the construction of rules related to the 'pickup' action.
> For clarity, here is a concise explanation of the ASP implementation guidelines.
> ASP rules for actions are primarily divided into two categories: precondition rules, which specify when actions are allowed, and effect rules, which define how actions change the environment state.
>
> Precondition rules are represented using integrity constraints in the following format:
>
> `:- action(actionname(Args), Time), condition(Args, Time), not condition(Args, Time).`
> These rules use the integrity constraint operator `:-` to define invalid action conditions. When the body of the constraint is satisfied, the specified action is considered forbidden. The `not` operator is used to indicate that the precondition must be satisfied for the action to be valid. Furthermore, multiple constraints can be defined for a single action to capture various invalid scenarios.
>
> Effect rules, on the other hand, are represented as:
>
> `state(Args, Time) :- action(ActionName(Args), Time).`
> These rules define the state changes resulting from actions. The `State` can represent any predicate, such as `holding/2`, `at/3`, capturing the direct causal relationship between the action and its effects. Additional conditions can be included in the rule by adding multiple body literals to specify more complex dependencies.
>
> For clarity, we are considering adding these implementation guidelines for ASP action rules to Section 3 of our paper. Alternatively, we may include the detailed guidelines in the Appendix D.3.2 and reference them from our paper for easier navigation.

---

> ### Author Response · Authors · 2024-11-19
>
> ---
> > **Weakness 3**: The left portion of Figure 1 is also confusing. It is not adequately addressed in the annotations, leaving the reader unsure of its significance.
>
> Figure 1 in our paper illustrates the process by which the embodied agent computes actions to achieve a given task while interacting with an open-domain environment. During this process, the agent observes its surroundings and perceives the success or failure of its actions.
> In the example depicted in the left portion of Figure 1, the action `move(R5, Drawer, Corner, Pick_Place_top)` fails because the robot attempts to grasp the broad surface of an oversized drawer, which is not feasible.
> This failure is recorded in the Experience Set, where it serves as valuable data for refining future decision-making.
>
> While conventional neuro-symbolic frameworks lack effective mechanisms to infer appropriate actions from such experiences, our framework learns from these experiences through the generality improvement process and successfully infers appropriate actions for subsequent trials.
>
> We revised the caption in Figure 1 of our paper to clearly reflect these details, enhancing the reader's understanding of the depicted process.
>
> ---
> > **Weakness 4**: Finally, certain experimental settings could benefit from clearer descriptions. For example, in the Dataset section (Section 4.1), the format of the expert-level episodic experience data is ambiguous. Is the data presented in natural language, code, or another format? Providing compact and clear explanations of these settings would make it easier for others to understand the data processing steps necessary for reproducing a similar system.
>
> In Section 4.1 of our paper, the expert-level episodic experience data is presented in a dictionary format, combining state-action-state pairs with observed outcomes in text form.
> Each episode is structured to include: (1) the observations and instructions, (2) the actions taken by the expert along with their affordances, and (3) the resulting action effects.
> The size of the data used for each benchmark is as follows: ALFWorld used 10 episodes, VirtualHome used 15 episodes, Minecraft used 6 episodes, and RLBench used 5 episodes. Although the dataset description was included in the Appendix, the lack of clearly defined sections may have made it difficult to locate.
>
> We update Appendix B.6 Datasets section to provide a clearer and more detailed explanation to enhance reproducibility. Below is an example of one episode data for ALFWorld:
>
> ```
> {
>    "instruction": "The robot needs to extract a fragile antique teacup from a high kitchen cabinet and place it safely on the dining table for inspection.",
>    "initial_observation": "You're in a traditional kitchen with oak cabinets. It's mid-morning, and sunlight streams through lace curtains, creating intricate shadow patterns. The high cabinet's glass door reveals several pieces of antique china. The valuable teacup in question sits on the top shelf, partially hidden behind a larger serving plate. A stepstool is visible near the refrigerator, and there's a slight layer of dust on some of the cabinet surfaces, suggesting these items aren't frequently accessed.",
>    "trajectory": {
>        "0": {
>            "observation": "The antique teacup is clearly visible through the glass cabinet door, sitting on the highest shelf approximately 7 feet from the floor. It's a delicate Bone China piece with hand-painted roses and gold trim, estimated to be from the 1920s. The cup is positioned behind a heavy ceramic serving plate, which partially blocks access. The cabinet's brass handle shows fingerprints from previous use, and the glass panes have some smudges that distort the view slightly. The overhead lighting reflects off the cabinet's glass, creating glare spots that make it difficult to see all angles of the teacup. A thin film of dust is visible on the shelf surface.",
>            "action": "pick_up stepstool from floor",
>            "affordance": "true",
>            "next_observation": "Standing with the stepstool, you're facing the cabinet. The stool's rubber feet rest on the recently waxed hardwood floor, which shows a noticeable sheen. The teacup remains visible through the glass, but from this angle, you can now see that the serving plate in front appears to be leaning slightly against the cup. The cabinet's hinges look slightly loose, with one screw not fully tightened. The morning sunlight has shifted, causing stronger glare on the cabinet glass."
>        },
>       "1":{
>         ....
>
>    }
> }
> ```

---

> ### Author Response · Authors · 2024-11-19
>
> > **Question 1**: Questions Regarding $itermax$ and Rule Learning ($R$)
> >
> > * I am curious about the role of the fixed $itermax$ in learning the rules $R$. Does setting a fixed $itermax$ always guarantee a reasonable system with appropriate rules? Could you provide results for different $itermax$ values to show their impact on performance and the resulting rules?
> > * Additionally, I am particularly interested in the notion of **convergence** within the NeSyC system. It might be valuable to explore how convergence is defined in your case. How sensitive is the system to environmental changes—would significant changes disrupt convergence? Also, is it possible for the rules $R$ to exhibit a **cyclical pattern** rather than converging to a stable state?
> > * On the topic of continual learning, I noticed that your system appears to stop growing its rules $R$ at some point ($itermax$), with subsequent testing based on a fixed set of rules. This seems different from my understanding of continual learning as a life-long process where rules evolve continuously based on new data. Could you conduct experiments where NeSyC and baseline agents keep learning and testing in various settings without a fixed $itermax$? It would be interesting to compare their performance, especially in terms of how quickly each system improves its success rate across tasks.
>
> For $itermax$ in Eq. (6) of our paper, the $itermax$ value serves as a practical parameter for limiting iterations rather than being a critical factor in rule quality.
> The framework's robustness stems from its ability to generalize knowledge through accumulated experiences and effective hypothesis generation process, rather than the specific value of $itermax$.
>
> The ability to generate appropriate hypotheses within one or several iterations depends on the specific action, task, and LLM used. For this reason, we typically set $itermax$ to 3 or more, as this value has proven sufficient for generating effective hypotheses through our LLM-based ILP process. During each iteration, the HI scoring and feedback mechanism ensures that proper feedback is provided, enabling efficient rule refinement without requiring an excessive number of iterations.
>
> Here are some example cases from our experiment in the RLBench environment using a window manipulation task with the instruction, "Open the left window through the handle".
> Although one iteration appeared successful with 12 rules, we continued to refine and expand the rule set until it converged to 13 rules, which were necessary to handle both left and right window operations. We observed varying convergence rates across different LLMs. GPT-4o achieved convergence within two iterations, whereas GPT-4o-mini required three iterations in this case.
>
> **Impact of iteration count on performance**
> |**Model**|**itermax**|0|1|2|3|4|5|
> |-|-|-|-|-|-|-|-|
> |**GPT-4o**|**# of rule**|9|12|13|13|13|13|
> ||**Task Success/Failure**|Failure|Success|Success|Success|Success|Success|
> |**GPT-4o-mini**|**# of rule**|1|9|10|13|13|13|
> ||**Task Success/Failure**|Failure|Failure|Failure|Success|Success|Success|

---

> ### Author Response · Authors · 2024-11-19
>
> In response to the reviewer's concern regarding **continual learning**, we designed a scenario in which 8 tasks are provided continually within a single environment following [1], **without setting a fixed $itermax$** value in Algorithm 2.
> This scenario allowed us to analyze both the continual learning performance and the convergence of the overall rules $R$.
>
> The experiments presented in the tables below were conducted on ALFWorld and RLBench. Each continual learning scenario consisted of 8 phases, requiring approximately 50 and 40 steps (based on oracle performance) for task completion, respectively.
> In each phase, different tasks were given, and the framework continuously generalized actionable knowledge by leveraging accumulated experiences. During evaluation at each phase, we also assessed the performances of previous tasks to determine how well NeSyC retained existing knowledge while adapting to new environments.
> Results demonstrate that our framework maintains a robust task success rate across successive tasks, outperforming Reflexion which leverages long-term memory to avoid repeating mistakes from previous experiences.
>
> **Performance of continual learning in ALFWorld**
> |**Method**|**Phase 1**||**Phase 2**||**Phase 3**||**Phase 4**||**Phase 5**||**Phase 6**||**Phase 7**||**Phase 8**||
> |-|-|-|-|-|-|-|-|-|-|-|-|-|-|-|-|-|
> ||**SR**|**GC**|**SR**|**GC**|**SR**|**GC**|**SR**|**GC**|**SR**|**GC**|**SR**|**GC**|**SR**|**GC**|**SR**|**GC**|
> |**Reflexion**|100.0|100.0|100.0|100.0|33.3|66.7|75.0|75.0|80.0|93.3|83.3|83.3|71.4|82.1|50.0|70.8|
> |**NeSyC**|100.0|100.0|100.0|100.0|100.0|100.0|100.0|100.0|100.0|100.0|100.0|100.0|100.0|100.0|100.0|100.0|
>
>
> **Performance of continual learning in RLBench**
> |**Method**|**Phase 1**||**Phase 2**||**Phase 3**||**Phase 4**||**Phase 5**||**Phase 6**||**Phase 7**||**Phase 8**||
> |-|-|-|-|-|-|-|-|-|-|-|-|-|-|-|-|-|
> ||**SR**|**GC**|**SR**|**GC**|**SR**|**GC**|**SR**|**GC**|**SR**|**GC**|**SR**|**GC**|**SR**|**GC**|**SR**|**GC**|
> |**Reflexion**|100.0|100.0|50.0|50.0|66.7|66.7|50.0|50.0|40.0|50.0|33.3|50.0|29.1|43.3|25.0|38.2|
> |**NeSyC**|100.0|100.0|100.0|100.0|100.0|100.0|100.0|100.0|100.0|100.0|100.0|100.0|100.0|100.0|100.0|100.0|

---

> ### Author Response · Authors · 2024-11-19
>
> Additionally, to analyze the convergence of rules $R$ in our framework, we observe how the rules are continually refined and updated throughout the same continual learning process.
> During this process, we measured the rate of change between successive phases, denoted as $\Delta$, based on the number of rules, and the similarity of the refined rules to predefined expert-level rules, represented by the F1 score.
> The F1 score measures the similarity and alignment with the expert rule set, calculated by balancing precision (the proportion of correctly predicted rules among all predicted rules) and recall (the proportion of correctly predicted rules among all expert rules). The table below demonstrates that initially, there are significant changes to the overall rules $R$, but these changes gradually decrease over time, eventually indicating convergence in terms of the number of rules and their similarity to expert rules.
>
> **Analysis of continual learning in ALFWorld**
>
> |**Method**|**Phase 1**||**Phase 2**||**Phase 3**||**Phase 4**||**Phase 5**||**Phase 6**||**Phase 7**||**Phase 8**||
> |-|-|-|-|-|-|-|-|-|-|-|-|-|-|-|-|-|
> ||**Δ**|**F1**|**Δ**|**F1**|**Δ**|**F1**|**Δ**|**F1**|**Δ**|**F1**|**Δ**|**F1**|**Δ**|**F1**|**Δ**|**F1**|
> |**NeSyC**|150.0%|28.6|0.0%|28.6|200.0%|50.0|66.7%|73.7|40.0%|90.9|0.0%|90.9|0.0%|90.9|0.0%|90.9|
>
> To the best of our knowledge, there are no established metrics or prior studies that provide a clear reference for evaluating the **convergence of the rule set**. Therefore, we utilize rule counting and similarity as practical measures to assess convergence.
> We believe that studying the convergence of rule sets within the context of neuro-symbolic approaches holds great potential for advancing the field of rule learning. Gaining a deeper understanding of how rule sets evolve and stabilize could offer foundational insights into achieving consistency, efficiency, and robustness in neuro-symbolic systems. Such research would not only address an important theoretical gap but also contribute to the development of more reliable and interpretable frameworks for practical applications.
>
> In our framework, the generalization of actionable knowledge primarily depends on accumulating new experiences in diverse environments [2]. This is facilitated by the memory-based monitoring scheme, which effectively handles errors during task execution and updates the experience set. These updates are then used in the contrastive generality improvement process to refine rules iteratively.
>
> Regarding the reviewer's concerns about **cyclical patterns** and **sensitivity to environmental changes**, we acknowledge that, as the reviewer insightfully pointed out, the rule learning process can indeed exhibit cyclical patterns or be heavily influenced by environmental changes.
> In fact, during the initial stages of designing our framework, we encountered challenges in effectively controlling the LLM-based ILP process, which sometimes resulted in such issues.
> However, we addressed these challenges by continually leveraging existing experience data and ensuring that hypotheses are updated incrementally, rather than regenerating them from scratch for each new experience.
> This approach ensures that existing experiences and rules are always utilized in the refinement process.
> These efforts are reflected in the design of our HI scoring function, where the $\lambda$ parameter in Eq. (9) of our paper balances the contributions of existing and newly added experiences.
> Additionally, the memory-based monitoring scheme effectively incorporates experience data incrementally, ensuring a seamless and efficient refinement process.
>
> [1] Kim et al. Online Continual Learning for Interactive Instruction Following Agents, ICLR 2024.
>
> [2] Abel et al. A definition of continual reinforcement learning. NeurIPS 2023.

---

> ### Author Response · Authors · 2024-11-19
>
> > **Question 2**: Questions Regarding Experiments
> >
> > * I have some concerns about the "Step" evaluation in your experiments. Although this evaluation follows prior work, I worry about potential overfitting. Achieving tasks does not always require strict adherence to ground truth; instead, achieving key milestones or sub-goals should suffice. This is particularly true for manipulation tasks—for example, a robot’s trajectory does not need to exactly match the demonstration. In Table 1, I noticed that both NeSYC and other systems show high "Step" evaluation scores, indicating close similarity to demonstrations. This raises questions about these systems' generalization capabilities in new environments. Could you clarify how NeSYC ensures generalization beyond following exact demonstrations? Are there additional experiments or insights that address this concern?
>
> The "Step" metric in our evaluation is not intended to directly measure the framework's generalization capabilities but serves as an auxiliary metric to assess how well the agent's action sequences align with one of the valid trajectories derived from multiple expert plans.
> This approach allows flexibility by not constraining the evaluation to a single trajectory, enabling multiple valid paths to task completion.
> The metric is particularly useful for evaluating whether the agent generates logically coherent, task-aligned actions.
> To establish reliable reference trajectories, we developed expert planning policies based on the benchmark's rule-based expert system and prior work [1,2,3,4,5,6], iteratively validating these trajectories to ensure they represent optimal and reliable solutions across all task scenarios.
>
> As the reviewer noted, while the "Step" metric reflects how closely the agent's actions align with valid expert plans, it is not intended to directly assess the agent's ability to generalize across diverse tasks or novel scenarios.
> Instead, metrics such as GC (goal conditions) and SR (success rate) are more suitable for evaluating generalization capabilities, as they measure the agent's success in achieving goals regardless of the specific action sequence followed.
>
> The inclusion of the "Step" metric complements the evaluation process by offering insights into the agent's ability to compute valid and efficient action sequences.
> For example, a high "Step" score indicates that the agent's trajectory aligns well with valid plans but does not imply overfitting to a specific demonstration or expert policy.
> By combining the "Step" metric with GC and SR, we provide a more comprehensive understanding of the agent's performance across different dimensions.
>
> In summary, the "Step" metric is a measure of trajectory validity rather than a direct indicator of generalization. Its role is to provide additional insights into the quality of the agent's action sequences from the perspective of expert plans, complementing the broader evaluation of generalization through GC and SR.
>
> [1] Aeronautiques et al. Pddl the planning domain definition language. 1998.
>
> [2] Shridhar et al. ALFRED: A Benchmark for Interpreting Grounded Instructions for Everyday Tasks. CVPR 2020.
>
> [3] Shridhar. ALFWorld: Aligning Text and Embodied Environments for Interactive Learning. ICLR 2021.
>
> [4] Li et al. BEHAVIOR-1K: A Human-Centered, Embodied AI Benchmark with 1,000 Everyday Activities and Realistic Simulation. CoRL 2022.
>
> [5] Puig et al. VirtualHome: Simulating Household Activities via Programs. CVPR 2018.
>
> [6] James et al. RLBench: The Robot Learning Benchmark \& Learning Environment. IEEE Robotics and Automation Letters 2020.

---

> ### Author Response · Authors · 2024-11-19
>
> > **Question 2**: Questions Regarding Experiments
> >
> > * Similarly, in the **Ablation Study** (Section 4.3), could you provide more insight into why simplified planning improves Goal Completion (GC) and "Step" evaluation?
>
> Regarding **Ablation Study** (Section 4.3), NeSyC leverages a symbolic tool for planning, ensuring logical rigor in generating action sequences. However, this approach can sometimes deviate from the optimal path, especially in the initial stages of planning, as it attempts to compute actions based on prior rules to be refined.
> In contrast, the simplified planning approach (replacing the ASP solver with SymbCoT [1] or CoT [2]) introduces more flexibility, allowing the agent to compute actions that may not strictly adhere to the current rules. This flexibility increases the likelihood of reaching intermediate goals, resulting in higher GC and "Step" scores. However, since this approach does not guarantee SR, it tends to show relatively lower performance in terms of SR.
>
> This result is closely related to the limitations of symbolic reasoning using LLMs. Studies such as [3] and [4] have shown that even when LLMs generate incorrect reasoning processes during CoT, model performance can still improve due to the flexibility provided by the reasoning framework. In SymbCoT, symbolic representations are combined with unstructured text inputs to enable more flexible reasoning compared to strictly using symbolic tools.
> From this perspective, the simplified planning approach leverages this flexibility, which can provide benefits by generating somewhat reasonably proper plans that improve "Step" and GC, albeit at the cost of reduced reliability in ensuring overall task success.
>
> [1] Xu et al. Faithful Logical Reasoning via Symbolic Chain-of-Thought. ACL 2024.
>
> [2] Wei et al. Chain-of-Thought Prompting Elicits Reasoning in Large Language Models. NeurIPS 2022.
>
> [3] Wang et al. SCOTT: Self-Consistent Chain-of-Thought Distillation. ACL 2023.
>
> [4] Chia et al. Contrastive Chain-of-Thought Prompting. arXiv 2023.

---

> ### Author Response · Authors · 2024-11-19
>
> > **Question 2**: Questions Regarding Experiments
> >
> > * For the **Robustness to Experience Incompleteness** experiment in Section 4.3, could you elaborate on what you mean by Noisy and Imperfect experiences? What preprocessing steps did you apply to the datasets or system to create these noisy or imperfect conditions?
>
> For the **Robustness to Experience Incompleteness** experiment in Section 4.3 of our paper, we evaluated how the quality of experience data affects our framework from a practical perspective.
> To establish a baseline for incomplete conditions, we defined the Complete Experience Set as a scenario where the experience data is sufficient to generate generalized rules.
>
> In the following examples, we use variables in our predicates where `O` represents an object being manipulated (e.g., items to be picked up), `L` represents a location (e.g., tables, receptacles), and `T` represents a timestep of the action.
> From this baseline, we constructed two modified experience sets:
>
> * Noisy Experience Set: Created by randomly altering affordance labels in the Complete Experience Set. For example, in the existing experience set, actions for picking up heavy objects were incorrectly labeled as True, which led to an incorrect inference of the `pick_up action`'s precondition rule as `:- action(pick_up(O, L), T), holding(T).` However, this was inaccurate, as the action failed due to weight constraints. The correct labeling should be False, and the precondition rules for the `pick_up` action should be updated to `:- action(pick_up(O, L), T), not~heavy(O).` and `:- action(pick_up(O, L), T), holding(T).`
> * Imperfect Experience Set: Created by removing all experiences related to a specific rule. For example, the existing experiences only contained cases where the pick action was performed on objects inside closed receptacles, leading to a generalized rule `:- action(pick(O,L),T), not is_open(L,T).` However, in new environments, when instructed to retrieve objects from locations like dining tables that don't have an opening/closing mechanism, items can be picked up without checking their open state. In such environments, the rule should be modified to `:- action(pick_up(O, L), T), openable(L), not is_open(L, T).` to accommodate both types of receptacles.
>
> Under the same criteria, We provided other neuro-symbolic baselines with complete rule sets corresponding to the Complete Experience Set.
> For the Noisy Experience Set, we introduced noise into the predicates forming certain rules, and for the Imperfect Experience Set, we removed rules associated with a specific action.
> For the Natural Language baseline, we used prompts in natural language to represent the logic corresponding to these conditions.
>
> In existing ILP research, incompleteness in example sets is a general problem. Typical scenarios include cases where the example set for hypothesis generation contains mislabeled positive or negative examples or where the data has inherent ambiguity [1].
> In a similar vein, we evaluated the robustness of our framework in handling various conditions: when the experience set contains sufficient data, when it includes mislabeled experiences, and when experiences for specific actions are deliberately omitted.
>
> [1] Cropper et al. Inductive logic programming at 30. Machine Learning 2022.

---

> ### Author Response · Authors · 2024-11-19
>
> > **Question 2**: Questions Regarding Experiments
> >
> > * In the **comparison of different dynamics predicates** (Section 4.3), could you clarify what your **F1 score** represents? How was the evaluation data collected to calculate the F1 score? What insights are you aiming to provide with the changes in F1 score, especially in relation to the characteristics of the predicates?
>
> For the **F1 score** used in the **comparison of different dynamics predicates** (Section 4.3 of our paper), this experiment evaluates how closely the rules generated by the LLM match predefined expert rules for specific target predicates. The F1 score is used to quantify this similarity, where a higher F1 score indicates a greater overlap between the LLM-generated predicates and those in the expert rule as in [1].
>
> For example, True positives (TP) in this context are defined as predicates that match between the LLM-generated rule and the expert rule for the same target predicate. These predicates may appear either in the head or body of the rule.
> Consider a target predicate `is_cooled(O, T+1)`. The expert rule for `is_cooled(O, T+1)` might have the preconditions `action(cool(O, L), T), holding(O, T), and robot_at(L, T)`. If the LLM generates predicates as `action(cool(O, L), T), grasping(O, T), and robot_at(L, T)`, then:
> True Positives (TP) = 2 (matching predicates `action(cool(O, L), T)` and `robot_at(L, T)`).
> False Positives (FP) = 1 (LLM-generated `grasping(O, T)` not present in the expert rule).
> False Negatives (FN) = 1 (expert rule’s `holding(O, T)` not generated by the LLM).
> In this example:
> Precision = TP / (TP + FP) = 2 / (2 + 1) = 2/3.
> Recall = TP / (TP + FN) = 2 / (2 + 1) = 2/3.
> F1 Score = 2 * (Precision * Recall) / (Precision + Recall) = 2/3.
>
> While the improvement varies across different predicates and LLMs, predicates related to common-sense knowledge generally show greater improvement overall. The Attribute category, which involves relatively stable features, consistently achieves high F1 scores. In contrast, the Status and Spatiality categories exhibit more variability due to the dynamic nature of physical states and spatial relations. These findings suggest that NeSyC, by using a neuro-symbolic approach, not only deduces what is considered challenging for LLMs [2,3] but also facilitates improvement in embodied control.
>
> [1] Li et al. Embodied Agent Interface: Benchmarking LLMs for Embodied Decision Making. arXiv 2024.
>
> [2] Wang et al. NEWTON: Are large language models capable of physical reasoning? arXiv 2023.
>
> [3] Yamada et al. Evaluating spatial understanding of large language models. arXiv 2023.
>
>
> > **Question 2**: Questions Regarding Experiments
> >
> > * Lastly, I would appreciate more detailed observations from your **real-world experiments**. Much of the current description is qualitative. Were these real-world scenarios tested multiple times, and did you observe recurring patterns? It would also be helpful to quantify these observations with metrics, rather than relying solely on descriptive results.
>
> **Quantitative results for our real-world experiments** can be found in Table 6 of our paper, which provides results across different settings. These results quantify the SR and GC for various feedback types (LLM and human) and environmental settings (Static, Dynamic).
> Figure 5 of our paper presents a case from the static environment experiments shown in Table 6, where causal feedback was generated through the LLM rewrite process, achieving a success rate (SR) of 66.6\%. Each data point in Table 6 represents the results from nine independent trials, where each trial used unique combinations of objects and instructions. By conducting trials with diverse scenarios, we were able to observe consistent patterns in the LLM's corrective actions, demonstrating NeSyC's reliability across various real-world robot manipulation tasks.
>
> ---
>
> Thank you once again for your valuable feedback. We hope our responses have addressed your remaining concerns. If you have any additional questions or require further information, please don't hesitate to discuss, and we will gladly provide further clarification. We sincerely appreciate your time and thoughtful review.

---

> ### Author Response · Authors · 2024-11-23
>
> Dear Reviewer CpeJ,
>
> As the discussion period nears its end, we kindly request your review of our responses to ensure that we have adequately addressed your concerns and questions.
>
> **Additionally**, we conducted further experiments to demonstrate the generalizability of our framework’s actionable knowledge in a continual learning scenario.
> The reviewer has specifically expressed interest in the **convergence of rules** within our framework during **continual learning**, and we kindly ask you to take this into consideration.
>
> These experiments evaluate how effectively the actionable knowledge acquired through continual learning has been refined to generalize to future unseen tasks, as demonstrated in [1]. The results demonstrate our framework’s ability to adapt and refine its knowledge throughout the continual learning phases, supporting zero-shot generalization in dynamic and open-domain environments.
>
> Let us first provide a summary of the continual learning scenario and results, along with the table below for reference. The scenario consisted of 8 phases, with different tasks presented in each phase and no fixed $itermax$ value in Algorithm 2 of our paper. NeSyC continually generalized actionable knowledge by leveraging accumulated experiences. At each phase, we evaluated its performance on both current and previous tasks to assess how well it retained existing knowledge while adapting to new environments. To demonstrate convergence, we measured the rate of change between successive phases ($\Delta$), based on the number of rules, and assessed the similarity of the refined rules to predefined expert-level rules using the F1 score.
>
> **Performance of continual learning scenario in ALFWorld**
> | Method | Metric | Phase 1 | Phase 2 | Phase 3 | Phase 4 | Phase 5 | Phase 6 | Phase 7 | Phase 8 |
> |--------|---------|----------|----------|----------|----------|----------|----------|----------|----------|
> | NeSyC | SR | 100.0| 100.0| 100.0| 100.0| 100.0| 100.0| 100.0| 100.0|
> |  | GC | 100.0| 100.0| 100.0| 100.0| 100.0| 100.0| 100.0| 100.0|
> |  | Δ | 150.0% | 0.0% | 200.0% | 66.7% | 40.0% | 0.0% | 0.0% | 0.0% |
> |  | F1 | 28.6 | 28.6 | 50.0 | 73.7 | 90.9 | 90.9 | 90.9 | 90.9 |
>
> **Performance of continual learning scenario in RLBench**
> | Method | Metric | Phase 1 | Phase 2 | Phase 3 | Phase 4 | Phase 5 | Phase 6 | Phase 7 | Phase 8 |
> |--------|---------|----------|----------|----------|----------|----------|----------|----------|----------|
> | NeSyC | SR | 100.0| 100.0| 100.0| 100.0| 100.0| 100.0| 100.0| 100.0|
> |  | GC | 100.0 | 100.0 | 100.0 | 100.0| 100.0| 100.0| 100.0| 100.0|
> |  | Δ | 100.0% | 250.0% | 0.0% | 100.0% | 0.0% | 0.0% | 28.6% | 0.0% |
> |  | F1 | 30.3 | 63.4 | 63.4 | 80.6 | 80.6 | 96.3 | 90.9 | 96.3 |
>
> **Additionally**, as shown in the results in the tables below, we evaluated how the improved actionable knowledge during the continual learning scenarios in each benchmark was refined to generalize effectively to future unseen tasks.
> For this evaluation, we randomly selected 47 unseen tasks in a static setting and tested the intermediate rules ($R$) generated at specific phases. The results demonstrate that rules from earlier phases consistently expanded task coverage, highlighting the framework's ability to adapt and improve over time. This refinement process effectively leverages accumulated experiences to generalize existing rules.
>
> **Performance of actionable knowledge from each phase on unseen tasks in ALFWorld**
> | Method | Metric || $R$ from Phase 1 || $R$ from Phase 3 || $R$ from Phase 4 || $R$ from Phase 5 || $R$ from Phase 8 ||
> |-|-|-|-|-|-|-|-|-|-|-|-|-|
> | NeSyC | SR || 38.3 || 53.2 || 74.5 || 87.2 || 91.5 ||
> |  | GC || 50.4 || 67.9 || 77.3 || 90.8 || 94.3 ||
>
> **Performance of actionable knowledge from each phase on unseen tasks in RLBench**
> | Method | Metric || $R$ from Phase 1 || $R$ from Phase 2  || $R$ from Phase 5 || $R$ from Phase 7 || $R$ from Phase 8 ||
> |-|-|-|-|-|-|-|-|-|-|-|-|-|
> | NeSyC | SR || 23.8 || 43.9 || 67.9 || 93.3 || 93.9 ||
> |  | GC || 24.3 || 45.6 || 71.4 || 95.2 || 96.8 ||
>
> [1] Kim et al. Online Continual Learning for Interactive Instruction Following Agents, ICLR 2024.
>
> ---
>
> We hope our additional response has been helpful in addressing the reviewer’s comments. We greatly appreciate your time and thoughtful review and look forward to receiving your valuable insights and any further feedback you may have. Thank you once again!

---

> > ### Comment · Reviewer_CpeJ · 2024-11-24
> >
> > Thank you for your detailed response. Most of my key concerns have been resolved, and I've adjusted my score accordingly.

---

> > > ### Author Response · Authors · 2024-11-24
> > >
> > > Thank you for your prompt response to our reply and for acknowledging the improvement in scores! Most importantly, we are pleased to hear that your concerns have been addressed. We are especially grateful for your deep interest in our topic and the insightful feedback you have provided, which has been incredibly rewarding for us during the discussion period. The additional analyses and experiments we conducted will be carefully organized and incorporated to further enhance the clarity of our paper. Thank you once again for your thoughtful and detailed review!

---

### Official Review · Reviewer_HoAB · 2024-11-03

**Soundness:** 2
**Presentation:** 3
**Contribution:** 2
**Rating:** 5
**Confidence:** 3

**Summary:**

This submission proposes NeSyC, a neuro-symbolic continual learner for open-domain embodied tasks. It combines LLM with symbolic tools to improve the generalization capability with limited experience. Extensive experiments are conducted on four benchmarks and on a Franka robot in the real world. The proposed framework shows competitive performance compared to multiple existing methods.

**Strengths:**

- Good presentation.
- Extensive evaluation on multiple benchmarks and on a real robot.

**Weaknesses:**

- The overall system seems very complicated. It is unclear which part contributes to the overall performance. More detailed analysis and ablations are necessary to shed light into the framework design.
- Tasks are too easy. How dynamics are the environment is also unclear (in paper it only mentioned low dynamics and high dynamics).
- Since it's a planning framework, we'd expect it can handle long-horizon tasks. Considered tasks are instead short-horizon.

**Questions:**

See above.

---

> ### Author Response · Authors · 2024-11-19
>
> Dear Reviewer HoAB,
>
> We sincerely appreciate the reviewer’s valuable feedback, as well as the opportunity to improve the clarity of our paper. We will provide detailed and thorough responses to the weaknesses mentioned by the reviewer.
>
> ---
> > **Weakness 1**: The overall system seems very complicated. It is unclear which part contributes to the overall performance. More detailed analysis and ablations are necessary to shed light into the framework design.
>
> To address the reviewer’s concern about the **complexity of our methodology**, we will first provide a clear and structured explanation of our framework.
>
> The goal of our framework is to enable embodied agents to generalize actionable knowledge from limited experiences in open-domain environments. These environments are inherently dynamic and unpredictable, requiring continual learning to adapt to diverse scenarios effectively. To address these challenges, our framework combines the generalization capabilities of LLMs with the logical rigor of symbolic tools. This integration enhances actionable knowledge by facilitating iterative rule refinement, enabling agents to generate robust, generalizable plans for novel domains by leveraging both successful and failed experiences.
>
> Our framework operates through two key phases: **rule reformulation** and **rule application**, both of which are supported by **contrastive generality improvement** and **memory-based monitoring** schemes. These phases are implemented through an interleaved collaboration of LLMs and symbolic tools.
> In the rule reformulation phase, hypotheses are generated by the LLM based on the agent's environmental experiences. These hypotheses are then validated through symbolic tools to establish the most general hypothesis capable of explaining all given experiences. This ensures the logical consistency and adaptability of the actionable knowledge.
> In the rule application phase, the framework detects action errors during task execution using a memory-based monitoring scheme. Detected errors trigger a knowledge refinement process with collected experiences, allowing the framework to continuously expand and adapt the agent’s coverage of actionable knowledge to handle new domains effectively.
>
> Through the iterative operation of these phases and the integration of LLMs with symbolic tools, leveraging both successful and failed experiences from the agent's interaction with the environment, our framework **ensures robust performance and strong generalization capabilities** in open-domain environments.
> This approach aligns with prior works [1, 2, 3] that enhance LLMs by utilizing external feedback from symbolic tools and environments, achieving superior task performance compared to self-correction methods that rely solely on LLMs.
>
> In Section 4.3 of our paper, we report the results of an **Ablation study** to evaluate the impact of each proposed methodology—contrastive generality improvement scheme and memory-based monitoring scheme—on **overall performance** within the respective phases of the framework. In Table 5 of our paper, we evaluated the performance drop caused by replacing key components, such as substituting the $\theta$-subsumption CoT prompt $l_\text{sub}$ in the contrastive generality improvement scheme with simple LLM prompts and replacing the ASP solver in the memory-based monitoring scheme with alternative LLM-based symbolic reasoning methods.
> The results show that ablation of the contrastive generality improvement scheme results in an average success rate drop of 38.5%, while ablation of the memory-based monitoring scheme causes a 31.6% decline in success rate. When both schemes were ablated, the task success rate dropped significantly, with an average decline of 64.7%."

---

> ### Author Response · Authors · 2024-11-19
>
> Additionally, in Section 4.3, we also conducted another comprehensive evaluation of our framework, analyzing the **robustness to experience incompleteness** (Table 2), the **impact by different LLMs** (Table 3 and Figure 3), **predicate-type analysis on contrastive generality improvement scheme** (Figure 4 and Table 4), and **real-world scenario** (Figure 5 and Table6).
>
> In response to the reviewer’s request, we further **expand our analysis** to include the effects of batch size in contrastive generality improvement scheme and the convergence of actionable knowledge in a continual learning scenario.
>
> When the batch size is too small, the generated hypotheses often include an excessive number of overly specific predicates tailored to individual cases, which undermines sample efficiency and limits adaptability.
> On the other hand, including all examples in a single batch often results in overly abstract hypotheses that fail to capture necessary task-specific details.
> A balanced batch size of approximately 6 is achieved for our experiment's sufficient results. This size enables the generation of appropriately generalized hypotheses that avoid overfitting specific cases while maintaining sufficient specificity to handle diverse scenarios effectively.
> Here are examples of the process for generating preconditions and related rules for 'pick up' and 'put down' actions from experiences, comparing cases with a small batch size, all experiences, and a balanced batch size.
>
>
> **Small Batch (batch size=3)**
> ```
> :- action(pick_up(O, L), T), not object(O), not location(L).
> :- action(put_down(O, L), T), not object(O), not location(L).
> :- action(heat(O, L), T), not object(O), not location(L).
> :- action(cool(O, L), T), not object(O), not location(L).
> :- action(clean(O, L), T), not object(O), not location(L).
> :- action(use(O), T), not object(O).
> :- action(open(O), T), not openable(O).
> :- action(close(O), T), not openable(O).
> :- action(go_to(L), T), not location(L).
> :- action(pick_up(O, L), T), not openable(L), not is_open(L, T).
> :- action(put_down(O, L), T), not openable(L), not is_open(L, T).
> :- action(pick_up(O, L), T), not is_open(O, T), not openable(O).
> :- action(put_down(O, L), T), not is_open(O, T), not openable(O).
> :- action(heat(O, L), T), not is_heater(L), not heatable(O).
> :- action(cool(O, L), T), not is_cooler(L), not coolable(O).
> :- action(clean(O, L), T), not is_cleaner(L), not cleanable(O).
> :- action(pick_up(O, L), T), not is_open(L, T).
> :- action(put_down(O, L), T), not is_open(L, T).
> :- action(pick_up(O, L), T), not is_open(O, T).
> :- action(put_down(O, L), T), not is_open(O, T).
> :- action(put_down(O, L), T), not is_cleaned(O, T), not cleanable(O).
> ```
>
> **Large Batch (batch size=all examples)**
> ```
> :- action(pick_up(O, L), T), not is_open(L).
> :- action(put_down(O, L), T), not is_open(L).
> ```
>
> **Balanced Batch(ours) (batch size=6)**
> ```
> :- action(pick_up(O, L), T), not at(O, L, T).
> :- action(pick_up(O, L), T), not is_open(L, T).
> :- action(pick_up(O, L), T), not robot_at(L, T).
> :- action(pick_up(O, L), T), holding(O, T).
> :- action(put_down(O, L), T), not is_open(L, T).
> :- action(put_down(O, L), T), not holding(O, T).
> :- action(put_down(O, L), T), not robot_at(L, T).
> :- action(put_down(O, L), T), not openable(L).
> ```
>
> For the analysis of the convergence of actionable knowledge in a continual learning scenario, we will explain the results as part of our responses to **Weakness 3**.
>
> [1] Wu et al. Large Language Models Can Self-Correct with Key Condition Verification. EMNLP 2024.
>
> [2] Shinn et al. Reflexion: Language Agents with Verbal Reinforcement Learning. NeurIPS 2023.
>
> [3] Pan et al. Logic-LM: Empowering Large Language Models with Symbolic Solvers for Faithful Logical Reasoning. EMNLP 2023.

---

> ### Author Response · Authors · 2024-11-19
>
> > **Weakness 2**: Tasks are too easy. How dynamics are the environment is also unclear (in paper it only mentioned low dynamics and high dynamics).
>
> The tasks in our framework are **not trivial**.
> They are designed to reflect open-domain settings, building upon existing benchmark configurations, which are inherently complex due to their dynamic and unpredictable nature.
> These settings involve: (1) expanded observations, including spatial relations and object physical states, along with diverse and rephrased instructions that go beyond simple templates, requiring agents to handle varied linguistic inputs [1,2],
> and (2) frequent environmental changes, such as unpredictable state transitions and variable object affordances, which demand robust reasoning and adaptability [3,4].
>
> In Section 4.1, Experiment Setting, of our paper, we describe how the existing environments were extended to suit our open-domain setting. Detailed descriptions of the original environment configurations and the modifications for the open-domain setting are provided in Appendix B.
>
> To address the reviewer’s comment on the lack of clarity regarding environmental dynamics, we will provide detailed explanations of the Static, Low Dynamics, and High Dynamics settings used in our experiments.
>
> Basically, for observations, we configured the environment to include a wide range of object relations (9 in VirtualHome) and states (30 in VirtualHome), extending beyond a simple list of object types (188 in VirtualHome). This setup incorporates richer details using various predicates related to object states and relations (e.g., grabbable, cuttable, can open, sitting, plugged in, dirty, obj next to, next to, between).
> For instructions, instead of relying solely on about ten types of template-based instructions (e.g., "Heat some tomato and put it in the fridge.") from the original benchmarks (e.g., ALFWorld), we utilized a diverse set of paraphrased instructions (e.g., "Refrigerate the heated tomato."). These extensions were designed to effectively capture and utilize the changes in environmental dynamics as inputs for the agent.
>
> In terms of environment dynamics, we categorized the settings into three levels based on their dynamics, each affecting state changes.
> For **Static** settings, the environment remains consistent across episodes, with stable object states, action preconditions and effects, and goal conditions.
> For **Low Dynamic** settings, objects may change locations or conditions within episodes, but these changes require only minor adjustments to the agent's existing transition rules.
> For **High Dynamic** settings, object states and attributes can change unpredictably within episodes, affecting goal conditions and action preconditions, significantly increasing complexity. The agent must continuously re-evaluate its plans and refine its knowledge to effectively handle frequent state shifts and varying affordances.
>
> [1] Zheng et al. JARVIS: A Neuro-Symbolic Commonsense Reasoning Framework for Conversational Embodied Agents. arXiv 2022.
>
> [2] Chen et al. Language-Augmented Symbolic Planner for Open-World Task Planning. RSS 2024.
>
> [3] Yoo et al. Exploratory Retrieval-Augmented Planning For Continual Embodied Instruction Following. NeurIPS 2024.
>
> [4] Cai et al. Open-world multi-task control through goal-aware representation learning and adaptive horizon prediction. CVPR 2023.

---

> ### Author Response · Authors · 2024-11-19
>
> > **Weakness 3**: Since it's a planning framework, we'd expect it can handle long-horizon tasks. Considered tasks are instead short-horizon.
>
> The **planning horizon** in conventional embodied planning tasks, with values such as 10 in ALFWorld, 18 in VirtualHome, 8 in Minecraft, and 6 in RLBench (based on oracle performance), is generally considered long-horizon [1,2]. These values represent complex multi-step planning tasks, requiring agents to perform extended sequences of actions while reasoning over intermediate goals and adapting to dynamic environments.
>
> However, to further evaluate our framework's capability in handling longer-horizon tasks and to address the reviewer's expectations, we designed a continual learning scenario involving a series of sequential tasks within the same environment, following [3]. This setup naturally extends the planning horizon beyond conventional limits by requiring the agent to complete multiple tasks sequentially.
>
> Following this design, the experiments presented in the tables below were conducted on ALFWorld and RLBench. Each continual learning scenario consisted of 8 phases, requiring approximately 50 and 40 steps (based on oracle performance) for task completion, respectively. During evaluation at each phase, we also assessed the performances of previous tasks to determine how well NeSyC retained existing knowledge while adapting to new environments.
> In each phase, different tasks were given, and the framework continually generalized actionable knowledge by leveraging accumulated experiences.
> Results demonstrate that our framework maintains a robust task success rate across successive tasks, outperforming Reflexion which leverages long-term memory to avoid repeating mistakes from previous experiences.
>
> **Performance of continual learning in ALFWorld**
> |**Method**|**Phase 1**||**Phase 2**||**Phase 3**||**Phase 4**||**Phase 5**||**Phase 6**||**Phase 7**||**Phase 8**||
> |-|-|-|-|-|-|-|-|-|-|-|-|-|-|-|-|-|
> ||**SR**|**GC**|**SR**|**GC**|**SR**|**GC**|**SR**|**GC**|**SR**|**GC**|**SR**|**GC**|**SR**|**GC**|**SR**|**GC**|
> |**Reflexion**|100.0|100.0|100.0|100.0|33.3|66.7|75.0|75.0|80.0|93.3|83.3|83.3|71.4|82.1|50.0|70.8|
> |**NeSyC**|100.0|100.0|100.0|100.0|100.0|100.0|100.0|100.0|100.0|100.0|100.0|100.0|100.0|100.0|100.0|100.0|
>
>
> **Performance of continual learning in RLBench**
> |**Method**|**Phase 1**||**Phase 2**||**Phase 3**||**Phase 4**||**Phase 5**||**Phase 6**||**Phase 7**||**Phase 8**||
> |-|-|-|-|-|-|-|-|-|-|-|-|-|-|-|-|-|
> ||**SR**|**GC**|**SR**|**GC**|**SR**|**GC**|**SR**|**GC**|**SR**|**GC**|**SR**|**GC**|**SR**|**GC**|**SR**|**GC**|
> |**Reflexion**|100.0|100.0|50.0|50.0|66.7|66.7|50.0|50.0|40.0|50.0|33.3|50.0|29.1|43.3|25.0|38.2|
> |**NeSyC**|100.0|100.0|100.0|100.0|100.0|100.0|100.0|100.0|100.0|100.0|100.0|100.0|100.0|100.0|100.0|100.0|
>
> Additionally, to analyze the convergence of rules $R$ in our framework, we observe how the rules are continually refined and updated throughout the same continual learning process.
> During this process, we measured the rate of change between successive phases, denoted as $\Delta$, based on the number of rules, and the similarity of the refined rules to predefined expert-level rules, represented by the F1 score.
> The F1 score measures the similarity and alignment with the expert rule set, calculated by balancing precision (the proportion of correctly predicted rules among all predicted rules) and recall (the proportion of correctly predicted rules among all expert rules). The table below demonstrates that initially, there are significant changes to the overall rules $R$, but these changes gradually decrease over time, eventually indicating convergence in terms of the number of rules and their similarity to expert rules.
>
>
> **Analysis of continual learning in ALFWorld**
> |**Method**|**Phase 1**||**Phase 2**||**Phase 3**||**Phase 4**||**Phase 5**||**Phase 6**||**Phase 7**||**Phase 8**||
> |-|-|-|-|-|-|-|-|-|-|-|-|-|-|-|-|-|
> ||**Δ**|**F1**|**Δ**|**F1**|**Δ**|**F1**|**Δ**|**F1**|**Δ**|**F1**|**Δ**|**F1**|**Δ**|**F1**|**Δ**|**F1**|
> |**NeSyC**|150.0%|28.6|0.0%|28.6|200.0%|50.0|66.7%|73.7|40.0%|90.9|0.0%|90.9|0.0%|90.9|0.0%|90.9|
>
> ---
>
> [1] Rozanov et al. StateAct: State Tracking and Reasoning for Acting and Planning with Large Language Models. arXiv 2024.
>
> [2] Zhao et al. Large Language Models as Commonsense Knowledge for Large-Scale Task Planning. NeurIPS 2023.
>
> [3] Kim et al. Online Continual Learning for Interactive Instruction Following Agents. ICLR 2024.
>
> ---
>
> We hope our responses have effectively addressed your concerns, and we would be happy to provide further clarification or engage in additional discussion if needed. Thank you again for your thoughtful and valuable review! We look forward to any further feedback you may have.

---

> ### Author Response · Authors · 2024-11-23
>
> Dear Reviewer HoAB,
>
> As the discussion period approaches its end, we kindly request your review of our responses to ensure that we have adequately addressed your concerns.
>
> For further clarification, we conducted **additional experiments** to demonstrate the generalizability of our framework’s actionable knowledge in a continual learning scenario. The reviewer specifically expressed interest in the **analysis** and performance of our framework on **longer-horizon tasks**, and we kindly ask you to take this into consideration.
>
> These experiments evaluate how effectively the actionable knowledge acquired through continual learning has been refined to generalize to future unseen tasks, as demonstrated in [1]. The results highlight our framework’s ability to adapt and refine its knowledge throughout the continual learning phases, showing performance that supports zero-shot generalization in dynamic and open-domain environments.
>
> Let us first provide a summary of the continual learning scenario and results, along with the table below for reference. The scenario consisted of 8 phases, requiring approximately 50 and 40 steps (based on oracle performance) for task completion, respectively. NeSyC continually generalized actionable knowledge by leveraging accumulated experiences.
>
> At each phase, we evaluated the framework's performance on both current and previous tasks to assess how well it retained existing knowledge while adapting to new environments. Throughout these scenarios, we measured the rate of change between successive phases ($\Delta$), based on the number of rules, and assessed the similarity of the refined rules to predefined expert-level rules using the F1 score.
>
> **Performance of continual learning scenario in ALFWorld**
> | Method | Metric | Phase 1 | Phase 2 | Phase 3 | Phase 4 | Phase 5 | Phase 6 | Phase 7 | Phase 8 |
> |--------|---------|----------|----------|----------|----------|----------|----------|----------|----------|
> | NeSyC | SR | 100.0| 100.0| 100.0| 100.0| 100.0| 100.0| 100.0| 100.0|
> |  | GC | 100.0| 100.0| 100.0| 100.0| 100.0| 100.0| 100.0| 100.0|
> |  | Δ | 150.0% | 0.0% | 200.0% | 66.7% | 40.0% | 0.0% | 0.0% | 0.0% |
> |  | F1 | 28.6 | 28.6 | 50.0 | 73.7 | 90.9 | 90.9 | 90.9 | 90.9 |
>
> **Performance of continual learning scenario in RLBench**
> | Method | Metric | Phase 1 | Phase 2 | Phase 3 | Phase 4 | Phase 5 | Phase 6 | Phase 7 | Phase 8 |
> |--------|---------|----------|----------|----------|----------|----------|----------|----------|----------|
> | NeSyC | SR | 100.0| 100.0| 100.0| 100.0| 100.0| 100.0| 100.0| 100.0|
> |  | GC | 100.0 | 100.0 | 100.0 | 100.0| 100.0| 100.0| 100.0| 100.0|
> |  | Δ | 100.0% | 250.0% | 0.0% | 100.0% | 0.0% | 0.0% | 28.6% | 0.0% |
> |  | F1 | 30.3 | 63.4 | 63.4 | 80.6 | 80.6 | 96.3 | 90.9 | 96.3 |
>
> **Additionally**, as shown in the results in the tables below, we evaluated how the improved actionable knowledge during the continual learning scenarios in each benchmark was refined to generalize effectively to future unseen tasks. For this evaluation, we randomly selected 47 unseen tasks in a static setting and tested the intermediate rules ($R$) generated at specific phases.
> The results indicate that rules from earlier phases consistently expanded task coverage, demonstrating the framework's ability to adapt and improve over time. This refinement process effectively leverages accumulated experiences to generalize existing rules.
>
> **Performance of actionable knowledge from each phase on unseen tasks in ALFWorld**
> | Method | Metric || $R$ from Phase 1 || $R$ from Phase 3 || $R$ from Phase 4 || $R$ from Phase 5 || $R$ from Phase 8 ||
> |-|-|-|-|-|-|-|-|-|-|-|-|-|
> | NeSyC | SR || 38.3 || 53.2 || 74.5 || 87.2 || 91.5 ||
> |  | GC || 50.4 || 67.9 || 77.3 || 90.8 || 94.3 ||
>
> **Performance of actionable knowledge from each phase on unseen tasks in RLBench**
> | Method | Metric || $R$ from Phase 1 || $R$ from Phase 2  || $R$ from Phase 5 || $R$ from Phase 7 || $R$ from Phase 8 ||
> |-|-|-|-|-|-|-|-|-|-|-|-|-|
> | NeSyC | SR || 23.8 || 43.9 || 67.9 || 93.3 || 93.9 ||
> |  | GC || 24.3 || 45.6 || 71.4 || 95.2 || 96.8 ||
>
> [1] Kim et al. Online Continual Learning for Interactive Instruction Following Agents, ICLR 2024.
>
> ---
>
> We hope our additional response has effectively addressed the reviewer’s comments. We deeply appreciate your time and thoughtful review and look forward to your valuable insights and any additional feedback you may have. Thank you again!

---

> > ### Comment · Reviewer_HoAB · 2024-11-24
> >
> > Thank you authors for the response and clarifications. My concerns about the system complexity and task triviality still remain. I'd like to maintain my original evaluation.

---

> > > ### Author Response · Authors · 2024-11-25
> > >
> > > Thank you for your response to our rebuttal and for pointing out the remaining concerns regarding system complexity and task triviality that require further attention.
> > > As the discussion period is still ongoing, we would like to provide an additional response to address this feedback.
> > >
> > > ---
> > >
> > > ### Regarding system complexity.
> > >
> > > Our framework is thoughtfully designed with a clear structure and independent contributions, emphasizing coherence rather than **complexity**. Below, we outline its key components and their contributions:
> > >
> > > * NeSyC comprises two key components: the *contrastive generality improvement* scheme and the *memory-based monitoring* scheme.
> > > * The *contrastive generality improvement* significantly contributes to performance improvement by generalizing actionable knowledge through an iterative process that incorporates environmental and symbolic tool feedback. This objective aligns with the framework’s focus on refining and expanding knowledge for open-domain adaptability.
> > > * The *memory-based monitoring* scheme is crucial for maintaining and enhancing performance. It leverages symbolic tools to compute precise actions based on current actionable knowledge, efficiently manages interaction data, and triggers knowledge refinement when necessary. This seamlessly aligns with the framework’s objective of maintaining accurate and adaptable knowledge in dynamic environments.
> > >
> > > Our work introduces and interprets a neuro-symbolic approach tailored for embodied agents, involving components such as the 'Task Planner' and 'Action Executor' within our framework. We understand that these components might have been challenging for some readers to follow. To address this, we revised the introduction section and Figure 1 to improve clarity and accessibility.
> > >
> > > The revised Figure 1 provides a high-level overview of NeSyC, effectively illustrating the overall functionality and highlighting how our approach addresses specific challenges compared to existing methods. These revisions are expected to reduce the **complexity** of understanding our framework. The improvements include:
> > >
> > > * The leftmost panel illustrates the task setup and domain shifts, emphasizing the open-domain nature of the environments with which the agent interacts.
> > > * The middle panels introduce key components such as the accumulated *Experience Set* and prior *Actionable Knowledge*, and the *Reasoning Models* that form the basis of the neuro-symbolic approach. As shown in Case 1, the conventional neuro-symbolic framework treats the LLM and symbolic tool as separate functions for semantic parsing and logical reasoning, respectively. In Case 2, NeSyC integrates these models into a collaborative process, where LLMs and symbolic tools refine actionable knowledge together, enabling continual learning and adaptation based on new experiences.
> > > * The rightmost panel demonstrates that, compared to conventional neuro-symbolic approaches, our framework computes logically valid actions more effectively, allowing the agent to transition seamlessly to the next step.

---

> ### Author Response · Authors · 2024-11-25
>
> ### Regarding task triviality.
>
> As noted in Appendix A Related Work section, our investigation reveals that numerous recent studies regard embodied tasks as **challenging problems** requiring logical reasoning [1,2,3,4]. These studies are increasingly adopting neuro-symbolic approaches to address challenges in embodied domains. However, while existing approaches rely on the complete provision of expert-level symbolic knowledge for embodied control, our work addresses scenarios where such knowledge is either insufficient or inapplicable due to the unpredictable nature of open-domain environments. These scenarios further elevate the **complexity and difficulty** of embodied tasks.
>
> Baselines such as ReAct [5] and Reflexion [6] exhibit performance differences even in the static setting due to the increased **difficulty** of our experimental environment compared to the original benchmark settings used in their respective papers. To ensure fair and meaningful comparisons under these open-domain conditions, we made the following adjustments for the baselines, as demonstrated in [7].
>
> In the original papers, ReAct and Reflexion utilize task-specific demonstrations as few-shot input prompts, where detailed task information is explicitly provided. In contrast, our experimental setup employs task-agnostic demonstrations as input prompts, omitting specific task information. This significantly increases the **difficulty** of grounding within the environment, as the models must independently infer task-relevant information.
>
> To quantitatively validate these performance differences, we conducted ablation studies with ReAct focusing on two key factors: expanded observations and task-specific few-shot prompts, both of which can influence model performance. For these experiments, we randomly selected 49 tasks and used GPT-4o-mini as the LLM.
>
> The results show that when we adjusted our Static settings to replicate the original experimental settings used for ReAct—by reducing the complexity of observations and providing task-specific prompts—the performance aligned more closely with the results reported in the original paper. This demonstrates that the observed performance differences stem from the **additional challenges** introduced by our open-domain settings. Furthermore, when considering the overall average performance across all task categories, the reported SR in the original ReAct paper is 57%. In our experimental results below, when comparing the average performance across all tasks, ReAct achieved approximately 59.2% in the reproduced original setting, aligning closely with the score originally reported in [5].
>
> We add these clarifications in Appendix B.7 to avoid any confusion regarding our experimental settings.
>
> **Performance comparison of ReAct on variants of ALFWorld environmental settings**
> | Configuration                                             | **SR** | **GC** |
> |-----------------------------------------------------------|--------|--------|
> | Static setting (without task-specific prompt and with expanded observations) | 14.3   | 33.7   |
> | Original setting (with task-specific prompt and without expanded observations) | 59.2   | 64.8   |

---

> ### Author Response · Authors · 2024-11-25
>
> ### Regarding open-domain environmental settings.
>
> To implement **open-domain environment setups** that increase the **complexity and difficulty** of embodied tasks, we referred to various established methods, as mentioned in our response to Weakness 2. Specifically, we referenced [1,2] for implementing expanded observations and [3,4] for introducing unpredictable environmental changes within episodes.
>
> To introduce unpredictability into the environment under dynamic settings, we defined parameters that primarily affect three key aspects: (1) object states, (2) action preconditions and effects, and (3) goal conditions.
> To quantitatively evaluate how changes in each aspect influence model performance, we adopted a three-type evaluation process:
>
> * Rate of Environmental Changes per Step: We first set the rate of environmental changes per step to measure the degree of unpredictability introduced.
> * Action Execution Success Rate: To assess how changes in individual aspects affect the model's detailed action execution success rates, we evaluated whether the expected effects of a computed action matched the subsequent observations, providing a more task-agnostic metric than step accuracy in Table 1 of our paper.
> * Task Success Rate (SR): We measured the overall task success rate (SR).
>
> For the experiments, we sampled 54 tasks from the ALFWorld benchmark.
>
> The experimental results in the table below highlight the robustness of our framework in adapting to various aspects of environmental dynamics. While changes in individual aspects, such as object states, action preconditions and effects, and goal conditions, influence the model's detailed action execution success rates, our framework effectively adapts to these dynamics, maintaining consistent and robust performance. Notably, the most complex and dynamic setting—where (1) object states, (2) action preconditions and effects, and (3) goal conditions simultaneously influence the environment—corresponds to the high dynamic setting described in Table 1 of our paper.
>
>
> **Performance Analysis on Environmental Dynamics Settings in ALFWorld**
>
> | **Metric**                       | **Affected Aspect: None** | **Affected Aspect: (1)** | **Affected Aspect: (2)** | **Affected Aspect: (3)** | **Affected Aspect: (1), (2), (3)** |
> |----------------------------------|----------|----------|----------|----------|--------------------|
> | **Rate of Environmental Changes per Step** | 0%       | 33.3%    | 50%      | 37.5%    | 70.8%             |
> | **Action Execution Success Rate**          | 79.5%    | 53.7%    | 49.3%    | 76.0%    | 34.2%             |
> | **Task Success Rate (SR)**                 | 88.9%    | 79.6%    | 74.1%    | 87.0%    | 68.5%             |
>
>
>
> [1] Lin et al. CLMASP: Coupling Large Language Models with Answer Set Programming for Robotic Task Planning. arXiv 2024.
>
> [2] Liu et al. Llm+ p: Empowering large language models with optimal planning proficiency. arXiv 2023.
>
> [3] Agarwal et al. Llm+ reasoning+ planning for supporting incomplete user queries in presence of apis. arXiv 2023.
>
> [4] Cornelio \& Diab. Recover: A neuro-symbolic framework for failure detection and recovery. arXiv 2023.
>
> [5] Yao et al. ReAct: Synergizing Reasoning and Acting in Language Models. ICLR 2023.
>
> [6] Shinn et al. Reflexion: Language Agents with Verbal Reinforcement Learning. NeurIPS 2023
>
> [7] Xie et al. TravelPlanner: A Benchmark for Real-World Planning with Language Agents. ICML 2024.
>
> ---
>
> Throughout this discussion period, addressing all of the reviewer’s concerns has been our top priority. We sincerely hope that our additional responses effectively resolve any remaining issues. We greatly appreciate further comments and will do our utmost to address them promptly until the end of the discussion. Thank you once again for your valuable time and thorough review of our paper!

---

> ### Author Response · Authors · 2024-11-26
>
> We would like to provide additional responses addressing Weakness 1 concerning system complexity.
>
> ---
>
> Our framework is systematically well-defined and designed to reduce system complexity across multiple dimensions: 1. minimizing **framework adaptation complexity** for diverse scenarios, 2. lowering **knowledge maintenance complexity** by reducing expert involvement, and 3. decreasing **data curation complexity** for effective knowledge generalization.
>
> #### 1. Regarding framework adaptation complexity.
>
> NeSyC introduces a well-defined modular design with clearly distinct components, each serving a specific and well-defined purpose. This modular approach enables seamless adaptability, allowing individual modules to be replaced or customized for a wide range of scenarios without increasing the complexity of restructuring the overall framework.
>
> We demonstrated the generality of our framework through various experiments.
> As shown in Table 1 and Figure 5 of Section 4, NeSyC enabled us to conduct experiments across a wide range of benchmarks—including ALFWorld, VirtualHome, Minecraft, RLBench, and real-world scenarios—by **simply replacing the executor.**
> As demonstrated in Table 3 and Figure 3, our framework achieves consistent performance improvements and broad applicability, proving its effectiveness even **when different LLMs are utilized in the hypothesis generator and interpreter.**
> Additionally, Table 6 demonstrates that our framework leverages the **error handler to seamlessly incorporate human feedback**, resulting in further performance enhancements.
>
> #### 2. Regarding knowledge maintenance complexity.
>
> Conventional neuro-symbolic systems often rely on static, predefined rules crafted by domain experts [1,2,3,4].
> While suitable for well-defined, closed domains [5,6], this reliance on rigid knowledge makes them impractical for dynamic open-domain environments, where tasks and conditions vary significantly. Open-domain scenarios demand frequent updates to domain-specific knowledge, requiring continuous expert involvement to refine or expand the rule base. This process is time-consuming, resource-intensive, and limits scalability and efficiency.
>
> NeSyC overcomes these challenges by dynamically generalizing rules through a seamless integration of inductive and deductive reasoning, significantly reducing the need for extensive manual intervention.
> Using inductive reasoning, hypotheses are automatically generated through the pre-trained knowledge of LLMs, minimizing the reliance on experts for crafting rules and enabling rapid adaptation to diverse scenarios.
> These hypotheses are then validated using deductive reasoning via symbolic tools, ensuring logical consistency and domain relevance. This combination makes the rules both reliable and actionable. By reducing dependence on expert input and continuously adapting to new scenarios, NeSyC offers a scalable, efficient, and explainable solution for open-domain tasks.
>
> #### 3. Regarding Data curation complexity.
>
> Many frameworks [7,8,9] rely on large datasets to achieve robustness, requiring data collection, manual curation, and labeled examples. This reliance makes them resource-intensive and impractical for real-world scenarios where data is often limited, inconsistent, or expensive to obtain.
>
> NeSyC addresses data curation complexity by optimizing the use of interaction data, including failed experiences, and leveraging pre-trained LLMs to access a vast repository of common knowledge. Failures are treated as valuable learning opportunities, allowing the framework to refine knowledge by addressing gaps without extensive task-specific data. LLMs provide broad contextual understanding, while symbolic tools validate and refine this knowledge, ensuring logical consistency and applicability. This combined approach minimizes the need for large, curated datasets, making NeSyC both data-efficient and scalable for real-world applications.
>
> ---
>
> [1] Lin et al. CLMASP: Coupling Large Language Models with Answer Set Programming for Robotic Task Planning. arXiv 2024.
>
> [2] Liu et al. LLM+P: Empowering Large Language Models with Optimal Planning Proficiency. arXiv 2023.
>
> [3] Agarwal et al. LLM+Reasoning+Planning for supporting incomplete user queries in presence of APIs. arXiv 2024.
>
> [4] Pan et al. Logic-LM: Empowering Large Language Models with Symbolic Solvers for Faithful Logical Reasoning. EMNLP 2023.
>
> [5] Yao et al. Information Extraction over Structured Data: Question Answering with Freebase. ACL 2024.
>
> [6] Salunkhe et al. Evolution of Techniques for Question Answering over Knowledge Base: A Survey. IJCA 2020.
>
> [7] Song et al. Llm-planner: Few-shot grounded planning for embodied agents with large language models. ICCV 2023.
>
> [8] Mu et al. EmbodiedGPT: Vision-Language Pre-Training via Embodied Chain of Thought. NeurIPS 2023.
>
> [9] Driess et al. PaLM-E: An Embodied Multimodal Language Model. ICML 2023.

---

> ### Author Response · Authors · 2024-11-26
>
> ### Regarding Paper Readability.
>
> During the discussion period, we have continuously revised the main manuscript, along with the previously mentioned Appendix sections, to address the points raised by the reviewers. The following are the revisions we have made to enhance the readability of the paper:
>
> * In Section 1, Introduction, we revised the text to clearly highlight the practical challenges of generalizing actionable knowledge in open-domain environments.
> * We simplified Figure 1 to present our motivation and the problem we aim to solve in a clear and straightforward manner. Specifically, we reduced the number of text elements and boxes, which previously made the figure appear overly dense and complex.
> * We revised the caption of Figure 1 to ensure that the concept and objective of NeSyC discussed in the introduction section are clearly conveyed.
> * To clarify the objective of NeSyC, we corrected Eq. (3) in Section 2.3, Problem Formulation, and added relevant references to support the updated formulation.
> * We revised the visual representation of our framework in Figure 2, simplifying its design to make it more intuitive and help readers better understand the key components of our approach.
> * In Section 3.1, we updated the descriptions of the Rule Reformulation and Rule Application phases to clearly articulate their objectives and underlying techniques.
> * In Section 3.2, we refined the explanation of the process for inducing and validating hypotheses to improve clarity.
> * In Section 3.3, we revised the explanation of the modules handling environment interactions and the processes for managing experiences to make them clearer.
> * We updated the symbols for instructions and goal states in Section 3.3 to align with the revisions made in Section 2.3.
> * In Algorithm 2, we revised the symbols to align with Section 2.3, ensuring clarity in the inputs between modules and their interactions with the environment.
>
> Additionally, we aim to further improve the following aspects before the final revision submission deadline:
>
> * We will carefully reorganize the additional analyses and experiments conducted during the discussion period in Section 4. Evaluation and the Appendix to further enhance the clarity of our paper.
>
> We are grateful for the reviewer's feedback during the discussion period and have worked diligently to incorporate the given comments, enhancing the presentation of the paper.
> To ensure the best possible revision, we would greatly appreciate the reviewer's suggestions, particularly in areas where further improvement might be needed. As a representative of a broad readership, your insights are invaluable, and we kindly request your input on these matters.
> The deadline for uploading the revised PDF is approaching; however, if you provide additional feedback even after the deadline, we will ensure that it is reflected in the final version. Once again, we sincerely thank the reviewer for the time, effort, and invaluable feedback.

---

### Official Review · Reviewer_2Gcc · 2024-11-04

**Soundness:** 3
**Presentation:** 2
**Contribution:** 3
**Rating:** 5
**Confidence:** 4

**Summary:**

This paper introduces NeSyC, where a large language model is given access to ILP like framework, specifically Answer Set Programming (ASP), alongside an ASP solver like clingo, where the agent iteratively comes up with hypotheses, attempts to solve tasks, and builds up a knowledge base about the world, aided by a symbolic tool. The authors evaluate their approach on open-domain environments, and show performance improvements over previous results.

**Strengths:**

- The proposed method seems novel, and the combination of neural agents and symbolic tools in open-domain settings is of interest to the community at large.
- The method outperforms previous methods on open-domain environments.
- The proposed method seems to be able to handle high-dynamic settings where the world changes in unpredictable ways.

**Weaknesses:**

- The experimental results seem to have different numbers from the baselines, particularly in a comparable environment like ALFWorld (see questions about Reflexion and ReAct). The baseline methods in their own existing work did not use some of the environments used in this work. The numbers seem to be different enough to ask the question on performance, though it is possible this is not a weakness and the difference can be explained.
- There is no comparison to the proposed approach for RLBench on existing work on that environment like ARP+ or RVT.
- The writing can be improved to be clearer, as there are several complicated modules at play. It would be good to include qualitative examples of tasks, plans, and trajectories (see questions).

**Questions:**

- In Line 354, the authors mention temperature = 0, why was greedy sampling used?
- What was the observation space for the various environments examined? Were RGB images provided, like was done in e.g., LLM-Planner?
- Why are the ReAct performance numbers in the static case so different from the ReAct paper (71 SR reported in their paper for ALFWorld vs 35.8 SR). Same for Reflexion (>0.9 SR reported in their paper on ALFWorld vs 39.0 SR). Is this purely from the difference in the LM used?
- How many trials were used for Reflexion? How many trials were used with the proposed method?
- In Figure 12, the Tower of Hanoi example, why did the agent decide to first put the rings in pegs in the wrong order, only to finish it correctly later?
- In Appendix B.5, do the authors still use VoxPoser for low-level robot control?
- The robotics tasks seem to rely on VoxPoser for low-level control. The reason for the performance of the proposed approach is “constrained by factors such as actuator range, grip force, and balance”, but the plans in Table 11 don’t have any such parameters. How is NeSyC influencing things like grip force? Does the model come up with things like “fragile” in Appendix D.3.1?
- What do some qualitative planned trajectories on a task from all the baselines, and the proposed method look like? Would be great to include those.

---

> ### Author Response · Authors · 2024-11-19
>
> Dear Reviewer 2Gcc,
>
> We are grateful for the detailed and thoughtful feedback received from the reviewer and for the opportunity to improve the clarity of our paper.
> We especially appreciate the reviewer's effort and interest in thoroughly reviewing the Appendix, which is beyond the mandatory scope. We will provide detailed and comprehensive responses to the weaknesses and questions raised by the reviewer.
>
> ---
> > **Weakness 1**: The experimental results seem to have different numbers from the baselines, particularly in a comparable environment like ALFWorld (see questions about Reflexion and ReAct). The baseline methods in their own existing work did not use some of the environments used in this work. The numbers seem to be different enough to ask the question on performance, though it is possible this is not a weakness and the difference can be explained. \
> > **Question 3**: Why are the ReAct performance numbers in the static case so different from the ReAct paper (71 SR reported in their paper for ALFWorld vs 35.8 SR). Same for Reflexion (>0.9 SR reported in their paper on ALFWorld vs 39.0 SR). Is this purely from the difference in the LM used?
>
> The **performance differences** observed in the baseline methods between the existing work and our experiments, particularly in ALFWorld, are primarily due to changes in the environmental configuration to accommodate the open-domain setting.
>
> The environmental settings are designed to reflect open-domain conditions, building upon existing benchmark configurations, which are inherently complex due to their dynamic and unpredictable nature.
> These settings involve:
> (1) expanded observations, including spatial relations and object physical states, along with diverse and rephrased instructions that go beyond simple templates, requiring agents to process varied linguistic inputs, as demonstrated in [1,2], and
> (2) frequent environmental changes, such as unpredictable state transitions and variable object affordances, necessitating robust reasoning and adaptability, similar to [3,4].
>
> Specifically, for observations, we configured the environment to include a wide range of object relations and states, extending beyond a simple list of object types. This setup incorporates richer details using various predicates related to object states and relations (e.g., pickupable, sliceable, can_open, dirty, at, etc).
> For instructions, instead of relying solely on about ten types of template-based instructions (e.g., "Heat some tomato and put it in the fridge.") from the original benchmarks, we utilized a diverse set of paraphrased instructions (e.g., "Refrigerate the heated tomato."). These extensions were designed to effectively capture and utilize the changes in environmental dynamics as inputs for the agent.
>
> In terms of environment dynamics, we categorized the settings into three levels based on their dynamics, each affecting state changes. For Static settings, the environment remains consistent across episodes, with stable object states, action preconditions and effects, and goal conditions. For Low Dynamic settings, objects may change locations or conditions within episodes, but these changes require only minor adjustments to the agent's existing transition rules. For High Dynamic settings, object states and attributes can change unpredictably within episodes, affecting goal conditions and action preconditions, significantly increasing complexity. The agent must continuously re-evaluate its plans and refine its knowledge to effectively handle frequent state shifts and varying affordances.

---

> ### Author Response · Authors · 2024-11-19
>
> For this reason, baselines such as ReAct [5] and Reflexion [6] show **performance differences** even in the static setting. Additionally, to ensure fair and meaningful comparisons under these open-domain conditions, we made the following adjustments for the baselines, as demonstrated in [7].
>
> For ReAct and Reflexion, the original papers utilize **task-specific demonstrations** as few-shot input prompts.
> In our setting, however, we provide task-agnostic demonstrations as input prompts without specifying task information, which increases the complexity of grounding within the given environment.
> To quantitatively validate this performance difference, we conduct ablation studies with ReAct focusing on two key factors: expanded observations and task-specific few-shot prompts, which can affect model performance. We randomly selected 49 tasks and used GPT-4o-mini as the LLM for the experiments.
>
> The results indicate that when we adjust our Static settings to replicate the original experimental settings used for ReAct—by reducing the complexity of observations and providing task-specific prompts—the performance aligns more closely with the results reported in the original paper.
> This demonstrates that the observed performance differences stem from the additional challenges introduced by our open-domain settings.
> Additionally, the 71\% SR mentioned by the reviewer for ReAct corresponds to the average performance on the best-performing task category. However, when considering the overall average performance across all task categories, the reported SR in the original paper is 57\%.
> In our experimental results below, when comparing the average performance across all tasks, ReAct achieved approximately 59.2\% in the reproduced original setting, which aligns closely with the score originally reported in [5].
>
> We add these clarifications in Appendix B.7 to avoid any confusion regarding our experimental settings.
>
> **Performance comparison of ReAct on variants of ALFWorld environmental settings**
> | Configuration                                             | **SR** | **GC** |
> |-----------------------------------------------------------|--------|--------|
> | Static setting (without task-specific prompt and with expanded observations) | 14.3   | 33.7   |
> | Original setting (with task-specific prompt and without expanded observations) | 59.2   | 64.8   |
>
> ---
>
> [1] Zheng et al. JARVIS: A Neuro-Symbolic Commonsense Reasoning Framework for Conversational Embodied Agents. arXiv 2022.
>
> [2]  Chen et al. Language-Augmented Symbolic Planner for Open-World Task Planning. RSS 2024.
>
> [3] Yoo et al. Exploratory Retrieval-Augmented Planning For Continual Embodied Instruction Following.  NeurIPS 2024.
>
> [4] Cai et al. Open-world multi-task control through goal-aware representation learning and adaptive horizon prediction. CVPR 2023.
>
> [5] Yao et al. ReAct: Synergizing Reasoning and Acting in Language Models. ICLR 2023.
>
> [6] Shinn et al. Reflexion: Language Agents with Verbal Reinforcement Learning. NeurIPS 2023
>
> [7] Xie et al. TravelPlanner: A Benchmark for Real-World Planning with Language Agents. ICML 2024.

---

> ### Author Response · Authors · 2024-11-19
>
> > **Weakness 2**: There is no comparison to the proposed approach for RLBench on existing work on that environment like ARP+ or RVT.
>
> **ARP+** [1] and **RVT** [2], as noted by the reviewer, are not directly comparable to our framework because they are not designed to evaluate reasoning capabilities for high-level task planning.
> Instead, they can be utilized as low-level skill decoders in conjunction with our framework, potentially complementing and enhancing its functionality.
>
> We understood the reviewer’s concern to indicate a lack of baselines specifically tailored for robot manipulation tasks, such as RLBench. In response, we conduct additional experiments incorporating methods specialized for robot manipulation while generating task plans using LLM-based approaches such as SayCan [3] and VoxPoser [4]. The experiments in the table below are conducted under the same settings as those described in Table 1 of our paper.
>
> SayCan is designed to identify feasible skills based on the current context, enabling the selection of the most appropriate skill while avoiding the execution of impossible ones. However, in the RLBench environment, the presence of fine-grained discrete parameters adds significant complexity to the process of calculating LLM scores, as SayCan's approach relies on high-level feasibility scoring, which becomes less effective in such detailed and parameterized skills.
> For instance, rotating from 1 degree to 180 degrees would result in 180 distinct skill options, making it more challenging to identify the optimal skill and leading to reduced performance for SayCan in such scenarios.
>
> VoxPoser utilizes prompts designed with a deep understanding of the low-level policy’s operational mechanisms, setting it apart from existing baselines.
> By aligning the prompt design with the specific ways the low-level policy performs actions, VoxPoser achieves improved performance in static environments, resulting in higher success rates compared to other LLM-based baselines.
> However, as environmental dynamics increase, it still faces limitations in generating appropriate high-level plans.
>
> **Performance comparison in RLBench**
> | **Method**      | **SR (Static)** | **GC (Static)** | **SR (Low Dynamics)** | **GC (Low Dynamics)** | **SR (High Dynamics)** | **GC (High Dynamics)** |
> |------------------|-----------------|-----------------|------------------------|------------------------|------------------------|------------------------|
> | LLM-planner      | 16.7           | 23.3           | 16.7                  | 20.8                  | 18.3                  | 21.7                  |
> | ReAct            | 23.3           | 25.8           | 21.7                  | 23.3                  | 18.3                  | 20.0                  |
> | Reflexion        | 33.3           | 41.4           | 21.7                  | 24.4                  | 23.3                  | 23.3                  |
> | AutoGen          | 43.3           | 54.2           | 23.3                  | 28.6                  | 21.7                  | 23.3                  |
> | CLMASP           | 94.5           | 95.8           | 0.0                   | 6.0                   | 0.0                   | 3.7                   |
> | **SayCan**           | 30.0           | 37.5           | 20.0                  | 27.5                  | 5.0                   | 7.5                   |
> | **VoxPoser**         | 60.0           | 65.0           | 20.0                  | 25.0                  | 20.0                  | 21.7                  |
> | NeSyC     | 85.5       | 88.5       | 81.5              | 84.8              | 79.0              | 81.8              |
>
> ---
>
> [1]  Zhang et al. Autoregressive Action Sequence Learning for Robotic Manipulation. arXiv 2024.
>
> [2] Goyal et al. RVT: Robotic View Transformer for 3D Object Manipulation. CoRL 2023.
>
> [3] Ahn et al. Do As I Can, Not As I Say: Grounding Language in Robotic Affordances. CoRL 2022.
>
> [4] Huang et al. VoxPoser: Composable 3D Value Maps for Robotic Manipulation with Language Models. CoRL 2023.

---

> ### Author Response · Authors · 2024-11-19
>
> > **Weakness 3**: The writing can be improved to be clearer, as there are several complicated modules at play. It would be good to include qualitative examples of tasks, plans, and trajectories (see questions).
>
> Thank you for your valuable feedback regarding the clarity of our paper. We are committed to revising our paper to improve its readability and ensure that the modules and their functions are clearly explained. To address the reviewer's concerns, we respond to the specific questions raised in this valuable feedback. Please read the responses to the questions below.
>
> > **Question 1**: In Line 354, the authors mention temperature = 0, why was greedy sampling used?
>
> To enhance the reproducibility of results from the LLM and minimize the risk of generating highly variable outputs, we intentionally set the temperature to 0. This decision was made to increase consistency across trials. With a temperature of 0, the LLM is more likely, though not guaranteed, to generate similar responses for identical inputs, reducing output variation while improving reliability in the results [1,2].
>
> $[1]$ Renze et al. The Effect of Sampling Temperature on Problem Solving in Large Language Models. arXiv 2024.
>
> $[2]$ Ouyang et al. LLM is Like a Box of Chocolates: the Non-determinism of ChatGPT in Code Generation. arXiv 2023.
>
> > **Question 2**: What was the observation space for the various environments examined? Were RGB images provided, like was done in e.g., LLM-Planner?
>
> As mentioned in Appendix B.6 of our paper and Weakness 1, **observations** are provided in a complex, fully text-based format, including spatial relations and object physical states. These are accompanied by diverse and rephrased instructions that go beyond simple templates, requiring agents to process varied linguistic inputs effectively.
> Our work is in the same vein as several previous works [1,2,3,4] in embodied reasoning tasks, which assume that the observations are provided in a textual format. The textual format is assumed to be generated through object detection models combined with heuristic methods. As the reviewer's valuable question suggests, exploring the use of VLMs (Vision-Language Models) or MLLMs (Multimodal Large Language Models) to generate symbolic representations for complex reasoning directly from RGB images in embodied tasks is being considered as the next step in our research.
>
> $[1]$ Huang et al. Language Models as Zero-Shot Planners: Extracting Actionable Knowledge for Embodied Agents. ICML 2022.
>
> $[2]$ Anh et al. Do As I Can, Not As I Say: Grounding Language in Robotic Affordances. arXiv 2022.
>
> $[3]$ Zhao et al. Large Language Models as Commonsense Knowledge for Large-Scale Task Planning. NeurIPS 2023.
>
> $[4]$ Singh et al. ProgPrompt: Generating Situated Robot Task Plans using Large Language Models. CoRL 2022.
>
> > **Question 4**: How many trials were used for Reflexion? How many trials were used with the proposed method?
>
> Both Reflexion [1] and our framework (as shown in Figure 1 of our paper) use the term 'trial'. While Reflexion uses 'trial' to refer to a complete episode, our framework uses it as the execution of an action. For consistency, in our response below to the reviewer's question above, we will use the term 'trial' in the same sense as Reflexion, referring to a complete episode.
>
> To ensure a fair comparison in the open-domain setting, both Reflexion and our framework were tested under conditions where no pre-defined, task-specific data was provided beforehand.
> Following [2], we limited each model to a single trial without additional opportunities to attempt multiple episodes for the same task, ensuring consistency with other baseline models.
> As in [2]'s implementation of Reflexion, we maintained the reflection mechanism for short-term memory and introduced an additional process for storing reflections in memory whenever more than three action errors occurred during task execution, facilitating adaptation to the open-domain setting.
>
> $[1]$ Shinn et al. Reflexion: Language Agents with Verbal Reinforcement Learning, NeurIPS 2023.
>
> $[2]$ Xie et al. TravelPlanner: A Benchmark for Real-World Planning with Language Agents, ICML 2024.

---

> ### Author Response · Authors · 2024-11-19
>
> > **Question 5**: In Figure 12, the Tower of Hanoi example, why did the agent decide to first put the rings in pegs in the wrong order, only to finish it correctly later?
>
> In addressing the Hanoi Tower problem, our hypothesis generator faced challenges in deriving a complete set of rules for an optimal plan solely from prior experiences. While it successfully generated basic rules (e.g., `:- move(DiskA, DiskB), larger(DiskA, DiskB), on(DiskB, DiskA).`), which ensure that smaller disks (DiskA) can only be placed on larger disks (DiskB), it failed to identify the recursive pattern required for optimal solutions, such as moving smaller disks to the auxiliary peg before transferring the largest disk to the target peg.
>
> As a result, our framework successfully completed the task, but the process involved some unnecessary actions, falling short of achieving optimal efficiency. While the final goal was satisfied, the intermediate steps did not strictly adhere to the optimal solution for the Hanoi Tower problem.
>
> Addressing this limitation, particularly the ability to identify and execute optimal intermediate steps, will be a key focus of our future work. We aim to enhance the framework’s capacity to infer and apply such complex recursive patterns, enabling more efficient task completion.
>
> > **Question 6**: In Appendix B.5, do the authors still use VoxPoser for low-level robot control? \
> > **Question 7**: The robotics tasks seem to rely on VoxPoser for low-level control. The reason for the performance of the proposed approach is “constrained by factors such as actuator range, grip force, and balance”, but the plans in Table 11 don’t have any such parameters. How is NeSyC influencing things like grip force? Does the model come up with things like “fragile” in Appendix D.3.1?
>
> In the real-world experiments, we did not use VoxPoser for low-level robot control. Instead, we utilized an RGB-D camera and an object detection module to identify object coordinates.
> For low-level control, we employed the open-source robot motion planning framework, MoveIt [1].
> Specifically, among the various motion planning algorithms available in MoveIt, we used the Model Predictive Control (MPC) algorithm to generate and continuously optimize control commands.
> MPC excels in trajectory optimization within complex environments, effectively handling constraints and ensuring real-time execution performance, which has made it a widely adopted approach in traditional robotic motion planning, as supported by prior research [2].
> The robot's movements were executed based on the generated plan, combined with pre-defined primitive skills, ensuring precise and efficient control.
>
> In RLBench, where VoxPoser was adapted for low-level control, the action space includes parameters that represent the physical properties of the environment.
> For example, instead of selecting a high-level semantic action like 'Turn right', our framework can compute a more precise action such as `action(rotate(clockwise, 100), T)` (i.e., 'Rotate clockwise 100 degrees in timestep T'). This approach enables precise control while maintaining semantic interpretability.
>
> As the reviewer pointed out, the granularity of actions in real-world scenarios slightly differs from those in RLBench.
> While we did not directly implement continuous force control for the gripper, our tasks required reasoning based on a discretized range of physical parameters within the actuator's operational range.
> To clarify this approach, the table below illustrates how physical parameters were mapped into discrete semantic values indicating the grasping strategy in our experiments. Additionally, to ensure the LLM effectively understands and utilizes these discrete semantic values, we incorporated explanations about the Grab Region into the prompt.
>
> **Mapping of physical parameters to discrete semantic actions.**
> | **Semantic Values for Grip** | **Grab Region (Diameter Range)** |
> |------------------------------|----------------------------------|
> | Precision                   | Center Point + 0–10%            |
> | Focus                       | Center Point + 10–20%           |
> | Standard                    | Center Point + 20–30%           |
> | Balance                     | Center Point + 30–40%           |
> | Power                       | Center Point + 40–50%           |
>
> We will include these examples and provide a more detailed explanation of the experimental setup for real-world scenarios in Appendix B.5.
>
> Additionally, the example involving the term `fragile`, as mentioned in Appendix D.3.1, pertains to a scenario in ALFWorld. This example highlights NeSyC's ability to interpret contextual properties but is not directly tied to low-level physical control in real-world experiments.
>
> $[1]$ Coleman et al. Reducing the barrier to entry of complex robotic software: a moveit! case study. arxiv 2014
>
> $[2]$ Yu et al. Language to Rewards for Robotic Skill Synthesis. CoRL 2023.

---

> ### Author Response · Authors · 2024-11-19
>
> > **Question 8**: What do some qualitative planned trajectories on a task from all the baselines, and the proposed method look like? Would be great to include those.
>
> In response to the reviewer’s suggestion, we included visualization of sample trajectories from each baseline and our framework in Figure 11 of our paper for qualitative analysis and included a comprehensive explanation of the experimental results in Appendix E.1.
>
> ---
> We sincerely thank you once again for your thoughtful review and hope that our responses have adequately addressed your concerns. During the discussion period, we will actively engage to identify and resolve any areas that may require further clarification. Please do not hesitate to leave additional questions if needed!

---

> > ### Comment · Reviewer_2Gcc · 2024-11-21
> >
> > Thank you for your engaging replies and answering my questions.
> >
> > > The results indicate that when we adjust our Static settings to replicate the original experimental settings used for ReAct
> >
> > I really appreciate this additional experiment by the authors. This does address my primary concern with reproducibility and rigor.
> >
> > > In response, we conduct additional experiments incorporating methods specialized for robot manipulation while generating task plans using LLM-based approaches such as SayCan [3] and VoxPoser [4].
> >
> > Thank you for these additional experiments, which also addressed my concerns about using RLBench as a setting.
> >
> > > we utilized an RGB-D camera and an object detection module to identify object coordinates
> >
> > I was trying to find this in the current manuscript, where are the details of the object detection module mentioned in the manuscript?
> >
> > > we included visualization of sample trajectories from each baseline and our framework in Figure 11
> >
> > Thank you for including this, it does make the affordance and problem setting much clearer.
> >
> > ---
> >
> > Based on the author's responses they have adequately addressed my primary concern, and answered other questions adequately. Therefore, I will be raising my score to a 5.
> >
> > However, the presentation of the paper needs careful rework. As somebody in the neuro-symbolic field, it was still very challenging to understand and read the paper. There are a lot of moving parts, and the concepts are fairly complex. For instance, in Figure 1, while the leftmost panel clearly shows the domain shift and the task setting, the middle panel suddenly introduces "Sematic Parsing" and "Logic Deduction".
> >
> > Based on the figure, it is not clear what the role of "Logic Deduction" is, or how it helps. The current way the figure is presented looks like the authors are just giving an LLM a series of functions like `rotate`, `go`, and `move_to`. The LLM tries using these symbolic tools, collects some experience of what worked and what did not work, and then modifying the parameters of those functions. This does not seem to include any benefits of using ASP.

---

> ### Author Response · Authors · 2024-11-22
>
> Thank you for your prompt response to our reply and for acknowledging the improvement in scores! We are delighted to hear that we have addressed the reviewer’s primary concerns and adequately answered other questions. We also sincerely thank you for providing further suggestions to improve our paper through continued discussion. Following the reviewer’s suggestions, we have included a detailed explanation of the **object detection module** and carefully revised the **presentation of the paper**.
>
> We have updated the revised paper, and the following provides an explanation of the changes made based on the reviewer’s feedback.
>
> ---
>
> > **Additional Question**: I was trying to find this in the current manuscript, where are the details of the object detection module mentioned in the manuscript?
>
> We acknowledge that our explanation of the real-world experimental setups requires more detail regarding the **object detection module**.
> Specifically, we used the publicly available "google/owlv2-base-patch16-ensemble" model, OWLv2 [1] from Hugging Face [2].
> This model was utilized to obtain object detection results (i.e., bounding-box, object name) and their coordinates, which were subsequently used to calculate the low-level action sequence.
>
> The process is as follows:
>
> 1) Capture a top-view image using a camera positioned above the table.
> 2) Use the **object detection module** [1] to identify the categories and bounding box coordinates of one or more target objects from the top-view image and retrieve their height information using a depth camera.
> 3) Convert the bounding box and height coordinates of the target object(s) to the robot arm’s coordinate system.
> 4) When one or more transformed target object coordinates are provided, input them into predefined heuristic skill functions (e.g., `move A to B`, `grasp A`) to prioritize which target coordinates to approach first and assign waypoints for these target coordinates accordingly.
> 5) Feed the inferred waypoints into a motion planning algorithm (i.e., MPC) to compute the low-level action sequence.
>
> We included this additional explanation in Appendix B.5 of the updated PDF.
>
> $[1]$ Minderer et al. Scaling Open-Vocabulary Object Detection. NeurIPS 2023.
>
> $[2]$ Wolf, Thomas, et al. Transformers: State-of-the-art natural language processing. EMNLP 2020.

---

> ### Author Response · Authors · 2024-11-22
>
> > **Additional Weakness**: the presentation of the paper needs careful rework. As somebody in the neuro-symbolic field, it was still very challenging to understand and read the paper. There are a lot of moving parts, and the concepts are fairly complex. For instance, in Figure 1, while the leftmost panel clearly shows the domain shift and the task setting, the middle panel suddenly introduces "Sematic Parsing" and "Logic Deduction". \
> > Based on the figure, it is not clear what the role of "Logic Deduction" is, or how it helps. The current way the figure is presented looks like the authors are just giving an LLM a series of functions like `rotate`, `go`, and `move_to`. The LLM tries using these symbolic tools, collects some experience of what worked and what did not work, and then modifying the parameters of those functions. This does not seem to include any benefits of using ASP.
>
> As our work introduces and interprets a neuro-symbolic approach from the perspective of embodied agents, the **multiple components** including 'Task planner' and 'Action executor' that make up our framework may have been challenging to follow for some readers. To address this, we have incorporated the reviewer’s feedback and revised the introduction section as well as **Figure 1** to enhance clarity.
>
> First, we will provide a concise explanation to clarify the core ideas and address your concerns.
> Our paper introduces NeSyC, a neuro-symbolic framework designed to generalize actionable knowledge in open-domain environments by integrating LLMs with symbolic tools such as an ASP solver.
> The framework enables embodied agents to tackle diverse and dynamic tasks by leveraging accumulated experiences and refining actionable knowledge represented in symbolic form (i.e., rule).
> NeSyC operates through the iteration of two key phases: rule reformulation (hypothesis generation and contrastive validation) and rule application (computing valid plans using rules and handling action errors to trigger rule refinement).
> This process is driven by the interleaved collaboration of LLMs and symbolic tools, allowing the agent to adapt effectively across a wide range of environments. This approach aligns with prior works [1, 2, 3] that enhance LLMs by utilizing external feedback from symbolic tools and environments, achieving superior task performance compared to self-correction methods that rely solely on LLMs.
>
> We recognize that the introduction of 'Semantic Parsing' and 'Logic Deduction', as well as the role of the symbolic tool in contributing to knowledge generalization, may be challenging to understand and could potentially lead to misunderstandings.
> The original intent of Figure 1 was to provide a high-level overview of NeSyC’s process, illustrating our objectives rather than the specific methods used to achieve them. In response to the reviewer’s valuable feedback as a peer in the neuro-symbolic field, we have revised Figure 1 to more clearly convey our intent.
>
> * The leftmost panel illustrates the task setup and domain shifts, highlighting the open-domain nature of the environments with which the agent interacts.
> * The middle panels introduce key components such as the accumulated 'Experience Set' and prior 'Actionable Knowledge', where actions like `rotate`, `go` and `move_to` are represented as rules and the 'Reasoning Models' that form the basis of the neuro-symbolic approach. As shown in Case 1, the conventional neuro-symbolic framework utilizes the LLM and symbolic tool as functions for semantic parsing and logical reasoning, respectively. In Case 2, NeSyC leverages these models for a more integrated process, where LLMs and symbolic tools collaboratively refine actionable knowledge, enabling continual learning and adaptation based on new experiences.
> * The rightmost panel demonstrates that our framework, in comparison to conventional neuro-symbolic approaches, is able to compute logically valid actions, allowing the agent to transition seamlessly to the next step.
>
> This revision demonstrates our framework’s ability to generalize actionable knowledge, represented as symbolic expressions (i.e., rules) such as `rotate`, `go`, and `move_to`, enabling more robust and adaptive decision-making in dynamic environments.
> Based on this explanation, we have revised Figure 1 along with its caption and updated the related description in the introduction section of the revised paper.
>
> $[1]$ Wu et al. Large Language Models Can Self-Correct with Key Condition Verification. EMNLP 2024.
>
> $[2]$ Shinn et al. Reflexion: Language Agents with Verbal Reinforcement Learning. NeurIPS 2023.
>
> $[3]$ Pan et al. Logic-LM: Empowering Large Language Models with Symbolic Solvers for Faithful Logical Reasoning. EMNLP 2023.
>
> ---
>
> We sincerely hope that our response and efforts have adequately addressed the reviewer’s additional concerns. Should there be any remaining feedback or questions, we would be happy to address them. Thank you once again for your thoughtful review!

---

> > ### Comment · Reviewer_2Gcc · 2024-11-26
> >
> > Thank you for your enthusiastic and detailed responses. Unfortunately, I share the concern about system complexity with reviewer HoAB, and also concerns about the quality of the writing and the presentation of the paper, and so I will maintain my current score. I appreciate the author's time and work on doing additional experiments and providing answers to clarification questions.

---

> ### Author Response · Authors · 2024-11-26
>
> Thank you for your prompt and continuous feedback on the concerns regarding system complexity and readability of our paper. We would like to provide additional clarification.
>
> ---
>
> Our framework is systematically well-defined and designed to reduce system complexity across multiple dimensions: 1. minimizing **framework adaptation complexity** for diverse scenarios, 2. lowering **knowledge maintenance complexity** by reducing expert involvement, and 3. decreasing **data curation complexity** for effective knowledge generalization.
> ### 1. Regarding framework adaptation complexity.
> NeSyC introduces a well-defined modular design with clearly distinct components, each serving a specific and well-defined purpose. This modular approach enables seamless adaptability, allowing individual modules to be replaced or customized for a wide range of scenarios without increasing the complexity of restructuring the overall framework.
>
> We demonstrated the generality of our framework through various experiments.
> As shown in Table 1 and Figure 5 of Section 4, NeSyC enabled us to conduct experiments across a wide range of benchmarks—including ALFWorld, VirtualHome, Minecraft, RLBench, and real-world scenarios—by **simply replacing the executor.**
> As demonstrated in Table 3 and Figure 3, our framework achieves consistent performance improvements and broad applicability, proving its effectiveness even **when different LLMs are utilized in the hypothesis generator and interpreter.**
> Additionally, Table 6 demonstrates that our framework leverages the **error handler to seamlessly incorporate human feedback**, resulting in further performance enhancements.
> ### 2. Regarding knowledge maintenance complexity.
> Conventional neuro-symbolic systems often rely on static, predefined rules crafted by domain experts [1,2,3,4].
> While suitable for well-defined, closed domains [5,6], this reliance on rigid knowledge makes them impractical for dynamic open-domain environments, where tasks and conditions vary significantly. Open-domain scenarios demand frequent updates to domain-specific knowledge, requiring continuous expert involvement to refine or expand the rule base. This process is time-consuming, resource-intensive, and limits scalability and efficiency.
>
> NeSyC overcomes these challenges by dynamically generalizing rules through a seamless integration of inductive and deductive reasoning, significantly reducing the need for extensive manual intervention.
> Using inductive reasoning, hypotheses are automatically generated through the pre-trained knowledge of LLMs, minimizing the reliance on experts for crafting rules and enabling rapid adaptation to diverse scenarios.
> These hypotheses are then validated using deductive reasoning via symbolic tools, ensuring logical consistency and domain relevance. This combination makes the rules both reliable and actionable. By reducing dependence on expert input and continuously adapting to new scenarios, NeSyC offers a scalable, efficient, and explainable solution for open-domain tasks.
> ### 3. Regarding Data curation complexity.
> Many frameworks [7,8,9] rely on large datasets to achieve robustness, requiring data collection, manual curation, and labeled examples. This reliance makes them resource-intensive and impractical for real-world scenarios where data is often limited, inconsistent, or expensive to obtain.
>
> NeSyC addresses data curation complexity by optimizing the use of interaction data, including failed experiences, and leveraging pre-trained LLMs to access a vast repository of common knowledge. Failures are treated as valuable learning opportunities, allowing the framework to refine knowledge by addressing gaps without extensive task-specific data. LLMs provide broad contextual understanding, while symbolic tools validate and refine this knowledge, ensuring logical consistency and applicability. This combined approach minimizes the need for large, curated datasets, making NeSyC both data-efficient and scalable for real-world applications.
>
> [1] Lin et al. CLMASP: Coupling Large Language Models with Answer Set Programming for Robotic Task Planning. arXiv 2024.
>
> [2] Liu et al. LLM+P: Empowering Large Language Models with Optimal Planning Proficiency. arXiv 2023.
>
> [3] Agarwal et al. LLM+Reasoning+Planning for supporting incomplete user queries in presence of APIs. arXiv 2024.
>
> [4] Pan et al. Logic-LM: Empowering Large Language Models with Symbolic Solvers for Faithful Logical Reasoning. EMNLP 2023.
>
> [5] Yao et al. Information Extraction over Structured Data: Question Answering with Freebase. ACL 2024.
>
> [6] Salunkhe et al. Evolution of Techniques for Question Answering over Knowledge Base: A Survey. IJCA 2020.
>
> [7] Song et al. Llm-planner: Few-shot grounded planning for embodied agents with large language models. ICCV 2023.
>
> [8] Mu et al. EmbodiedGPT: Vision-Language Pre-Training via Embodied Chain of Thought. NeurIPS 2023.
>
> [9] Driess et al. PaLM-E: An Embodied Multimodal Language Model. ICML 2023.

---

> ### Author Response · Authors · 2024-11-26
>
> ### Regarding Paper Readability.
>
> During the discussion period, we have continuously revised the main manuscript, along with the previously mentioned Appendix sections, to address the points raised by the reviewers. The following are the revisions we have made to enhance the readability of the paper:
>
> * In Section 1, Introduction, we revised the text to clearly highlight the practical challenges of generalizing actionable knowledge in open-domain environments.
> * We simplified Figure 1 to present our motivation and the problem we aim to solve in a clear and straightforward manner. Specifically, we reduced the number of text elements and boxes, which previously made the figure appear overly dense and complex.
> * We revised the caption of Figure 1 to ensure that the concept and objective of NeSyC discussed in the introduction section are clearly conveyed.
> * To clarify the objective of NeSyC, we corrected Eq. (3) in Section 2.3, Problem Formulation, and added relevant references to support the updated formulation.
> * We revised the visual representation of our framework in Figure 2, simplifying its design to make it more intuitive and help readers better understand the key components of our approach.
> * In Section 3.1, we updated the descriptions of the Rule Reformulation and Rule Application phases to clearly articulate their objectives and underlying techniques.
> * In Section 3.2, we refined the explanation of the process for inducing and validating hypotheses to improve clarity.
> * In Section 3.3, we revised the explanation of the modules handling environment interactions and the processes for managing experiences to make them clearer.
> * We updated the symbols for instructions and goal states in Section 3.3 to align with the revisions made in Section 2.3.
> * In Algorithm 2, we revised the symbols to align with Section 2.3, ensuring clarity in the inputs between modules and their interactions with the environment.
>
> Additionally, we aim to further improve the following aspects before the final revision submission deadline:
>
> * We will carefully reorganize the additional analyses and experiments conducted during the discussion period in Section 4. Evaluation and the Appendix to further enhance the clarity of our paper.
>
> We sincerely appreciate the reviewer's feedback during the discussion period and have worked diligently to incorporate the provided comments, improving the overall presentation of the paper.
> To ensure the highest quality revision, we would greatly value the reviewer's suggestions, especially in areas where further enhancements may be necessary. Your insights, as a representative of a broad readership, are invaluable, and we kindly request your input on these matters.
> As the deadline for uploading the revised PDF approaches, we assure you that any additional feedback provided, even after the deadline, will be carefully incorporated into the final version. Once again, we sincerely thank the reviewer for the time, effort, and valuable insights.

---

### Official Review · Reviewer_B2XV · 2024-11-07

**Soundness:** 4
**Presentation:** 4
**Contribution:** 4
**Rating:** 10
**Confidence:** 4

**Summary:**

This paper addresses the challenge of generalizing knowledge for embodied task planning from prior experiences to novel scenarios and domains using a neural-symbolic continual learning approach. The proposed approach, NeSyC (Neural-Symbolic Continual Learner?), represents general knowledge and experience (hypotheses and grounded instances of action preconditions and effects, respectively) using Answer Set Programming and relies on existing ASP solvers to find plans that satisfy goals according to the best current hypothesis. Given a set of experiences (the effect of actions taken given the precondition state), an LLM is used to parse these experiences into positive and negative examples based on the action affordances. Given these positive and negative examples, an LLM is used to generate a set of hypotheses by first extracting background knowledge from the examples then generating hypotheses that satisfy batches of the examples. The hypotheses are further curated via an LLM-based implementation of \theta-subsumption to include the most general of the generated hypotheses. Given these hypotheses, a symbolic hypothesis interpreter validates each with respect to all of the examples, providing feedback and a score for each hypothesis. This process of hypothesis generation and interpretation repeats for a maximum number of iterations, resulting in a final, most general hypothesis to be used as the general knowledge during planning.

Given a natural language instruction of a task and the current history of execution, an LLM is used to generate both goal states associated with the instruction and the current state of the world. A symbolic task planner uses the general knowledge from the iterative hypothesis generation-interpretation phase to find plans that reach the goal state. During execution, an error handler identifies after each action whether it succeeded or failed (or whether the effects match the general knowledge) and updates a working memory accordingly. In the case of failure, the current working memory is appended to the initial set of experiences, and the system re-enters the hypothesis generation-interpretation phase to update the general knowledge used for planning.

The proposed approach is evaluated through extensive experimentation on multiple standard benchmarks and compared against existing state-of-the-art methods, as well as demonstrated on a real robot in a complex tabletop scenario.

**Strengths:**

- The problem being addressed is of high significance to the embodied intelligence community.
- The paper is well written and organized; the method is clearly and thoroughly explained and illustrated.
- The contributions are clear, novel, and significant.
- The experimental evaluation is extensive and thorough. The approach is evaluated across multiple diverse benchmarks with increasing difficulty within each benchmark (increasing dynamics). The methods against which the approach is compared are, to the best of my knowledge, the most appropriate state-of-the-art methods.
- The performance is a significant improvement over other methods for dynamic environments.
- Several relevant ablations are included that help answer questions about the relative performance of different components; an appropriate suite of state-of-the-art LLMs are included, showing that the hypothesis generation-interpretation phase is very successful across LLMs, with the exception of Llama-3-8B.

**Weaknesses:**

- The proposed approach performs worse than CLMASP (another neuro-symbolic approach) in static environments for 3 of the 4 benchmarks.

**Questions:**

I recommend strong accept at this time because of the strengths listed above.

Here are a few questions that I would appreciate having answered:
- What is the motivation for batching the examples during the hypothesis generation step? I imagine that fewer examples makes it easier to generate a good hypothesis, and batches could promote hypothesis diversity. Did you notice any interesting relationship between batch size and performance? Did you try including all examples?
- Why is the "satisfy" feedback only provided once the max iterations are met (equation 6)? Is there no early termination condition for hypotheses that satisfy the examples?
- The approach does not reach 100% on any of the metrics on any benchmarks; what are some common failures?
- Llama-3-8B performed quite poorly, as reported in Table 3. Future work is proposed to use model distillation. Have you tried more recent small models, like LLama-3.1-8B or LLama-3.2-3B?


Here are a few minor grammatical errors to fix:
- Line 421 "in Figure 3" -> "In Figure 3"
- line 422 "alignment between the H1 score" H1 should be HI
- line 514 "which constraints" -> "which constrains"

A small part of the appendix seems to have been mistakenly erased. See line 1201, which seems to start in the middle of a sentence.

---

> ### Author Response · Authors · 2024-11-19
>
> Dear Reviewer B2XV,
>
> We sincerely appreciate the time and effort you dedicated to reviewing our paper. Your insightful and constructive feedback has been invaluable in helping us identify areas for improvement and refine our work.
> Additionally, we are grateful for your thorough review, including the Appendix, which went beyond the mandatory scope. We have addressed and corrected all the grammatical errors you identified.
>
> Moving forward, we will provide detailed and thoughtful responses to the weaknesses and questions you raised.
>
> ---
>
> > **Weakness 1**:
> The proposed approach performs worse than CLMASP (another neuro-symbolic approach) in static environments for 3 of the 4 benchmarks.
>
> As CLMASP [1] leverages pre-defined expert knowledge, it outperforms NeSyC in static environments. In these environments, where object states, action conditions, and goals remain consistent across episodes, CLMASP leverages additional expert-level actionable knowledge to achieve near-optimal performance. However, this reliance on pre-defined knowledge becomes a significant limitation in Low Dynamic and High Dynamic settings, leading to a substantial performance drop in open-domain environments. In contrast, NeSyC, which learns directly from experience data, not only achieves comparable performance in static environments but also maintains robust performance across varying levels of environmental dynamics.
>
> [1] Lin et al. CLMASP: Coupling Large Language Models with Answer Set Programming for Robotic Task Planning. arXiv 2024.

---

> ### Author Response · Authors · 2024-11-19
>
> > **Question 1**: What is the motivation for batching the examples during the hypothesis generation step? I imagine that fewer examples makes it easier to generate a good hypothesis, and batches could promote hypothesis diversity. Did you notice any interesting relationship between batch size and performance? Did you try including all examples?
>
> Our motivation for batching the examples during the hypothesis generation step stems from two key considerations: (1) achieving effective generalization by simultaneously processing diverse examples and (2) addressing the inherent limitations of the context window size, which varies depending on the LLM used and becomes increasingly constrained as more examples accumulate. A batch-based approach ensures a practical balance between these factors, enabling efficient and effective hypothesis generation.
>
> In response to the reviewer’s questions about batch size and performance, we observed that batch size can impact hypothesis quality, especially when set to extreme values.
> When the batch size is too small, the generated hypotheses often include an excessive number of overly specific predicates tailored to individual cases, which undermines sample efficiency and limits adaptability.
> On the other hand, including all examples in a single batch often results in overly abstract hypotheses that fail to capture necessary task-specific details.
> Here are examples of the process for generating preconditions and related rules for **pick up** and **put down** actions from experiences, comparing cases with a small batch size and those including all experiences.
>
> **Small Batch (batch size=3)**
> ```
> :- action(pick_up(O, L), T), not object(O), not location(L).
> :- action(put_down(O, L), T), not object(O), not location(L).
> :- action(heat(O, L), T), not object(O), not location(L).
> :- action(cool(O, L), T), not object(O), not location(L).
> :- action(clean(O, L), T), not object(O), not location(L).
> :- action(use(O), T), not object(O).
> :- action(open(O), T), not openable(O).
> :- action(close(O), T), not openable(O).
> :- action(go_to(L), T), not location(L).
> :- action(pick_up(O, L), T), not openable(L), not is_open(L, T).
> :- action(put_down(O, L), T), not openable(L), not is_open(L, T).
> :- action(pick_up(O, L), T), not is_open(O, T), not openable(O).
> :- action(put_down(O, L), T), not is_open(O, T), not openable(O).
> :- action(heat(O, L), T), not is_heater(L), not heatable(O).
> :- action(cool(O, L), T), not is_cooler(L), not coolable(O).
> :- action(clean(O, L), T), not is_cleaner(L), not cleanable(O).
> :- action(pick_up(O, L), T), not is_open(L, T).
> :- action(put_down(O, L), T), not is_open(L, T).
> :- action(pick_up(O, L), T), not is_open(O, T).
> :- action(put_down(O, L), T), not is_open(O, T).
> :- action(put_down(O, L), T), not is_cleaned(O, T), not cleanable(O).
> ```
>
> **Large Batch (batch size=all examples)**
> ```
> :- action(pick_up(O, L), T), not is_open(L).
> :- action(put_down(O, L), T), not is_open(L).
> ```
>
> A balanced batch size of approximately 6 achieves sufficient results. This size enables the generation of appropriately generalized hypotheses that avoid overfitting specific cases while maintaining sufficient specificity to handle diverse scenarios effectively. Below, we provide examples of hypotheses generated with balanced batch size.
>
> **Balanced Batch(ours) (batch size=6)**
> ```
> :- action(pick_up(O, L), T), not at(O, L, T).
> :- action(pick_up(O, L), T), not is_open(L, T).
> :- action(pick_up(O, L), T), not robot_at(L, T).
> :- action(pick_up(O, L), T), holding(O, T).
> :- action(put_down(O, L), T), not is_open(L, T).
> :- action(put_down(O, L), T), not holding(O, T).
> :- action(put_down(O, L), T), not robot_at(L, T).
> :- action(put_down(O, L), T), not openable(L).
> ```
>
> We greatly appreciate the reviewer's insightful observations, and we believe that such analysis presents valuable considerations for our future work.

---

> ### Author Response · Authors · 2024-11-19
>
> > **Question 2**: Why is the "satisfy" feedback only provided once the max iterations are met (equation 6)? Is there no early termination condition for hypotheses that satisfy the examples?
>
> The reason why the "*satisfy*" feedback is only provided once the maximum iterations "*itermax*" are reached is that "*itermax*" is not intended to directly determine the quality of the rules but serves as a practical limit to the number of iterations in the rule reformulation phase.
>
> The critical factor in ensuring effective hypothesis generation and improvement lies in the framework's iterative process, which relies on repeated interactions with the environment, updates to the experience set, and scoring-based feedback mechanisms (i.e., HI scoring). These mechanisms enable the framework to refine hypotheses efficiently and adapt to diverse scenarios.
>
> The ability to generate appropriate hypotheses within one or several iterations depends on the specific action, task, and the LLM used. For this reason, we typically set "*itermax*" to 3 or more, as this value has proven sufficient for generating effective hypotheses through our LLM-based ILP process. During each iteration, the HI scoring and feedback mechanism ensures that proper feedback is provided, enabling efficient rule refinement without requiring an excessive number of iterations.
>
> Here are some example cases, we conduct an experiment in the RLBench environment using a window manipulation task with the instruction "Open the left window through the handle".
> Although one iteration appeared successful with 12 rules, we continued to refine and expand the rule set until it converged to 13 rules, which were necessary to handle both left and right window operations.
> We observed varying convergence rates across different LLMs. GPT-4o achieved convergence within two iterations, whereas GPT-4o-mini required three iterations in this case.
>
> **Impact of iteration count on performance**
> |**Model**|**itermax**|0|1|2|3|4|5|
> |-|-|-|-|-|-|-|-|
> |**GPT-4o**|**# of rule**|9|12|13|13|13|13|
> ||**Task Success/Failure**|Failure|Success|Success|Success|Success|Success|
> |**GPT-4o-mini**|**# of rule**|1|9|10|13|13|13|
> ||**Task Success/Failure**|Failure|Failure|Failure|Success|Success|Success|
>
> In summary, "*itermax*" serves as a practical parameter for limiting iterations rather than being a critical factor in rule quality. The framework's robustness stems from its ability to generalize knowledge through accumulated experiences and effective hypothesis generation process, rather than the specific value of "*itermax*".
>
> ---
> > **Question 3.** The approach does not reach 100% on any of the metrics on any benchmarks; what are some common failures?
>
> The common reasons for task failures are related to translation errors during the semantic parsing process using the LLM.
> This issue is not unique to our approach; similar errors also occur in the baselines.
> In both NeSyC and CLMASP, additional errors can arise due to the use of a symbolic solver.
> While NeSyC and CLMASP include a mechanism to correct such synthetic errors, repeated errors beyond a certain threshold or significant planning disruptions caused by these errors can lead to termination due to exceeding the maximum environment step limit, thereby reducing the success rate.
> For example, given the instruction, "Place a few remote controls on the sofa", the symbolic representation generated by the LLM might be `goal(T) :- is_remote_control(O), is_sofa(L), at(O, L, T).`.
> Whereas the ground truth is `goal(T) :- is_remotecontrol(O), is_sofa(L), at(O, L, T).`.
> In cases like this, minor discrepancies in symbols can lead to failures in achieving the goal, underscoring the sensitivity of symbolic reasoning to exact syntax.

---

> ### Author Response · Authors · 2024-11-19
>
> > **Question 4.** Llama-3-8B performed quite poorly, as reported in Table 3. Future work is proposed to use model distillation. Have you tried more recent small models, like LLama-3.1-8B or LLama-3.2-3B?
>
> Although we have not yet conducted detailed research for future work, in response to the reviewer's suggestion, we conducted experiments similar to those presented in Table 3 of our paper using Llama-3.1-8B and Llama-3.2-3B.
> For Llama-3.1-8B, while it outperformed Llama-3-8B, its performance remained below practically meaningful levels.
> For Llama-3.2-3B, issues with the semantic parsing process prevented it from functioning correctly, making it challenging to obtain meaningful experimental results.
> Specifically, Llama-3.2-3B frequently introduced syntax errors during the semantic parsing process, such as incorrectly mapping predicates (e.g., using `object1` instead of the appropriate `object` type parameter), creating non-standard predicates like `inside_room` instead of the correct `inside` predicate, and generating non-existent predicates such as `variable_type`. These errors disrupted subsequent processing and affected overall performance.
>
> |**LLM**|**SR**|**→ SR**|**GC**|**→ GC**|
> |-|-|-|-|-|
> |**Llama-3.1-8B**|44.3|→ 56.8|49.6|→ 60.0|
> |Llama-3-8B|43.9|→ 40.2|43.9|→ 40.2|
> |GPT-4o-mini|41.9|→ 78.7|44.7|→ 79.3|
> |Claude-3.0-Opus|50.7|→ 76.7|53.6|→ 78.6|
> |Claude-3.5-Sonet|51.4|→ 78.4|54.2|→ 80.2|
> |Llama-3-70B|58.8|→ 85.1|60.4|→ 86.6|
> |GPT-4o|64.2|→ 90.2|67.0|→ 90.5|
> |GPT-4|69.6|→ 89.2|73.3|→ 89.8|
>
> ---
>
> We hope our responses have adequately addressed your concerns, and we would be happy to provide further clarification if needed.
> Thank you once again for your valuable review!

---

### Author Response · Authors · 2024-11-19
**General Response to All Reviewers**

We thank all reviewers for their constructive feedback. Based on the reviewers' comments and suggestions, we have made the following revisions to our paper.

Main Manuscript Revisions:

* Fixed typos and clarified notation (Reviewer B2XV):
    * Line 421: "in Figure 3" → "In Figure 3"
    * Line 422: "H1 score" → "HI score"
    * Line 514: "which constraints" → "which constrains"
* Clarified notation distinctions between sets ($\mathcal{I}_d$, $\mathcal{G}_d$) and elements ($i_d$, $g_d$) in Eq. (3) (Reviewer CpeJ)
* Enhanced caption in Figure 1 to better reflect the process details (Reviewer CpeJ, 2Gcc)
* Modified examples in General Knowledge $R$ in Figure 2 (Reviewer CpeJ)
* Revised Figure 1 and introduction section to better illustrate our neuro-symbolic framework (Reviewer 2Gcc)


Appendix Enhancements:

* Added a comprehensive analysis of experimental results in Appendix E.1 (Reviewer 2Gcc)
* Updated Appendix B.6 with detailed dataset descriptions (Reviewer CpeJ)
* Included examples and detailed setup for real-world scenarios in Appendix B.5 (Reviewer 2Gcc)
* Updated Appendix B.7 with performance differences in open-domain settings (Reviewers 2Gcc, HoAB)
* Added Figure 11 with sample trajectories from comparative models (Reviewer 2Gcc)
* Updated explanation of real-world object detection module in Appendix B.5 (Reviewer 2Gcc)

We have carefully addressed all major concerns raised by the reviewers and made detailed revisions. We hope these changes have enhanced the clarity and quality of our paper. We will continue to make updates as necessary throughout the discussion period.

---

### Author Response · Authors · 2024-11-28
**General Response to All Reviewers**

We sincerely appreciate the constructive feedback provided by all reviewers—Reviewer B2XV, Reviewer 2Gcc, Reviewer HoAB, and Reviewer CpeJ. In response to the reviewers' comments and feedback, as well as through our own efforts, we have finalized the revisions to our paper and uploaded the final version. We hope that these updates will be carefully considered by the reviewers and effectively address any remaining concerns. The detailed list of revisions follows below:


**Main Manuscript Revisions**
 * **Correction of Typos (Reviewer B2XV)**
    * Corrected 'in Figure 3' to 'In Figure 3' in Section 4.
    * Corrected 'H1 score' to 'HI score' in Section 4.
    * Corrected 'which constraints' to 'which constrains' in Section 5.
 * **Mathematical Clarity (Reviewer CpeJ)**
    * Clarified the distinction between sets ($\mathcal{I}_d, \mathcal{G}_d$) and elements ($i_d, g_d$) in Eq. (3) of Section 2.3.
    * Revised the notation for instructions and goal states in Section 3.3 to align with Eq. (3) in Section 2.3.
    * Revised the notation for instructions and goal states in Algorithm 2 to ensure consistency with both Eq. (3) in Section 2.3 and the description in Section 3.3.
 * **Figure Improvements (Reviewers 2Gcc, CpeJ)**
    * Simplified Figure 1 to clearly present the motivation and problem addressed by our framework.
    * Revised the caption of Figure 1 to clearly convey the concept and objective of our framework as discussed in Section 1.
    * Simplified the visual representation of Figure 2 to make it more intuitive and help readers better understand the key components of our framework.
 * **Readability Improvements (Reviewer 2Gcc)**
    * Refined Section 1 by reducing paraphrasing of key terms, enhancing consistency, and clarifying ambiguous expressions for improved clarity.
    * Added citations in Section 1 to support the argument that conventional neuro-symbolic approaches face challenges in open-domain environments.
    * Revised the text in Section 1 to emphasize the practical challenges of generalizing actionable knowledge in open-domain environments.
    * Revised terminology in Section 3 for precision and clarity, replacing 'general knowledge' with 'generalized knowledge' and 'most general hypothesis' with 'optimized hypothesis'.
    * Refined the explanation of the process for inducing and validating hypotheses in Section 3.2 to improve clarity.
    * Revised the explanation of components managing environment interactions and processes for handling experiences in Section 3.3 for better clarity.
    * Adjusted experiment descriptions in Section 4 to align with the revised terminology introduced in Section 3.
 * **Simplification of Framework Explanations (Reviewers 2Gcc, HoAB)**
    * Revised the framework motivation, technical overview, and contributions in Section 1 to ensure cohesive connections.
    * Revised and Expanded the explanation of ILP and ASP in Section 2 to highlight their relevance to our framework, ensuring smoother integration with Section 3.
    * Updated the descriptions of the Rule Reformulation and Rule Application phases in Section 3.1 to clearly articulate their objectives and techniques.

**Appendix Enhancements (Reviewers B2XV, 2Gcc, HoAB, CpeJ)**
* Enhanced Appendix B.5 with detailed real-world scenario examples and a comprehensive explanation of the object detection module.
* Updated Appendix B.6 with detailed dataset documentation, including data creation procedures, preprocessing methods, and illustrative examples of how the datasets were constructed and utilized in our experiments.
* Updated Appendix B.7 with an explanation of why baseline methods show lower performance in open-domain settings compared to their reported results in the original papers, clarifying the impact of environmental differences on performance metrics.
* Added rollout trajectories in Appendix E.1 to provide qualitative comparisons between baseline methods and our framework, demonstrating the effectiveness of our approach in handling sequential tasks and decision-making processes.
* Added experiments with smaller LLMs, such as LLama-3.1-8B and LLama-3.2-3B, in Appendix E.7, showing that their performance is still not practically meaningful.
* Added experiments on continual learning scenarios in ALFWorld and RLBench in Appendix E.6, showing robust task performance superior to Reflexion and stable generalization results with respect to rule convergence.

---

We understand that the discussion period is ongoing, and we welcome any responses, additional feedback, or questions regarding our response. We will continue to provide updates and actively participate as necessary throughout the discussion period. Once again, we sincerely thank all reviewers for their effort and dedication.

---

### Public Comment · ~Wonje_Choi2 · 2025-02-28
**Minor Title Correction Request**

Dear ACs and Reviewers,

We would like to begin by expressing our sincere gratitude for your positive evaluation of our paper.

We hope this message finds you well. We are writing regarding our paper titled:

"NeSyC: A Neuro-symbolic Continual Learner For Complex Embodied Tasks in Open Domains.”

We noticed a small typographical error in the title: specifically, the word “in” should be capitalized. Would it be possible to update the title to the following?

"NeSyC: A Neuro-symbolic Continual Learner For Complex Embodied Tasks In Open Domains”

We fully understand that the title can only be changed at the request of the reviewers or the area chair, and that it must not significantly alter the scope of the paper. This proposed change is purely typographical and does not affect the paper’s content or scope in any way.

Thank you for your time and consideration. We look forward to hearing from you.

Sincerely,

The Authors

---

### Meta-Review · Area_Chair_QY4Z · 2024-12-19

**Metareview:**

**Summary**: This paper introduces NeSyC, a neuro-symbolic continual learner designed to enable embodied agents to generalize actionable knowledge across diverse open-domain tasks. Inspired by the hypothetico-deductive model, NeSyC employs a contrastive generality improvement scheme to iteratively generate and validate hypotheses using LLMs and symbolic tools. Additionally, it incorporates a memory-based monitoring scheme to detect action errors and trigger knowledge refinement. The framework is evaluated on benchmarks like ALFWorld, VirtualHome, Minecraft, RLBench, and a real-world robotic tabletop scenario. NeSyC demonstrates strong performance across static, low dynamic, and high dynamic environments, outperforming baselines like AutoGen, ReAct, and CLMASP in open-domain settings.

**Strengths**:
- High-importance problem and clear, novel, and substantial contributions: The reviewers agree that NeSyC presents an innovative neuro-symbolic approach that combines LLM-based hypothesis generation with symbolic validation. This iterative process enables the agent to continually refine its knowledge, which is particularly valuable in open-domain, dynamic environments.
- Strong empirical performance: NeSyC demonstrates significant improvements over state-of-the-art baselines, especially in environments with high dynamics where existing methods struggle. Reviewer `B2XV` highlights the thorough and extensive evaluation on multiple benchmarks, including both simulation and real-world tasks. In fact, the reviewer gives the paper a 10, which I've personally never seen before at ICLR.
- Comprehensive evaluation: The reviewers commend the extensive experiments across ALFWorld, VirtualHome, Minecraft, RLBench, and real-world tasks. Additionally, Reviewer `B2XV` notes that the ablation studies provide valuable insights into the contributions of different components.
- Solid writing: (with some exceptions, read below) Reviewers generally praise the writing and organization, and clarity, and exposition of the paper.

**Weaknesses**:
- Inconsistencies in baseline results: Reviewer `2Gcc` raises concerns about discrepancies in baseline performance for methods like ReAct and Reflexion in ALFWorld. The authors attribute these differences to variations in the LLMs and environments but could improve clarity around this point. Additionally, comparisons to other RLBench methods, such as ARP+ or RVT, are missing, although the authors explain (convincingly) why these methods are not directly comparable to NeSyC. This discussion should be included in the related work section of the paper.
- Performance in static environments: Reviewer `B2XV` points out that NeSyC underperforms compared to CLMASP in static environments. The authors clarify that this is due to CLMASP’s reliance on predefined expert knowledge, which is less applicable in dynamic tasks. While the explanation is reasonable, the comparison highlights a trade-off between generalization and static task optimization, which should be included in the discussion section of the paper.
- Additional discussion of failure cases: While NeSyC shows strong performance, Reviewer `B2XV` and Reviewer `2Gcc` suggest adding a more detailed analysis of common failure cases. The authors provide some discussion of errors related to LLM-based semantic parsing and symbolic representation, but further exploration of these limitations would strengthen the work.
- Writing clarity: Despite the overall solid writing, multiple reviewers point out things that could be clarified or that read as convoluted, particularly given the complexity of the proposed framework. The authors address this by refining several sections and figures during the rebuttal, but further qualitative examples of planned trajectories and error handling would enhance clarity.

**Recommendation**: The paper introduces an innovative neuro-symbolic approach for open-domain embodied tasks with clear technical contributions and strong experimental validation. While there are concerns about performance in static environments, baseline discrepancies, and writing clarity, the authors have addressed these points adequately. The reviewers agree on the significance of the work, particularly its robustness in dynamic and open-domain settings. Finally, the paper scored a 10, which is the only such high grade that I've ever seen as an AC. As such, I vote for Accept (spotlight).

**Additional Comments On Reviewer Discussion:**

During the rebuttal, the authors addressed most concerns effectively:

- Performance in static environments: Pointed out by Reviewer `B2XV`. The authors clarified that CLMASP's advantage in static tasks stems from its reliance on expert knowledge, which becomes a limitation in dynamic settings.
- Baseline discrepancies: Pointed out by Reviewer `2Gcc`. The authors explained the differences in reported baseline results due to variations in LLMs and environmental setups. Additional experiments with recent LLMs were provided to strengthen the comparisons.
- Writing clarity improvements: The authors revised the text and figures to address concerns about clarity and added further details in the appendix, including dataset documentation, rollout trajectories, and error analysis.
- Discussion on failure cases: The authors provided examples of common failures, such as LLM parsing errors and inconsistencies in symbolic reasoning, and discussed potential future improvements.

Reviewers appreciated the detailed and thoughtful responses, and Reviewer `B2XV` maintained their strong accept recommendation. While Reviewer `2Gcc` remained slightly cautious, they acknowledged the improvements made during the rebuttal period. Reviewer `CpeJ` increased their score to an 8. Reviewer `HoAB` has a score of 5, but their review is rather short and they didn't engage in discussion during rebuttal. Overall, because of the 8 and the 10 I am inclined to Accept with a spotlight, although I wouldn't mind bumping this down to a poster. I am certain the paper needs to be accepted though.

---

### Decision · Program_Chairs · 2025-01-22

Accept (Poster)